# A placental model of SARS-CoV-2 infection reveals ACE2-dependent susceptibility and differentiation impairment in syncytiotrophoblasts

J. Chen [1,2,3,16], J. A. Neil [4,16], J. P. Tan[1,2,3,16], R. Rudraraju[4,16], M. Mohenska[1,2,3], Y. B. Y. Sun [1,2,3], E. Walters[1,2,3,5,6], N. G. Bediaga[5,6], G. Sun[1,2,3], Y. Zhou[1,2,3], Y. Li[7], D. Drew[8], P. Pymm[8,9], W. H. Tham[8,9], Y. Wang[7], F. J. Rossello[3,10], G. Nie[7], X. Liu [11,12,13,14], K. Subbarao [4,15] ✉ & J. M. Polo [1,2,3,5,6] ✉

SARS-CoV-2 infection causes COVID-19. Several clinical reports have linked COVID-19 during pregnancy to negative birth outcomes and placentitis. However, the pathophysiological mechanisms underpinning SARS-CoV-2 infection during placentation and early pregnancy are not clear. Here, to shed light on this, we used induced trophoblast stem cells to generate an in vitro early placenta infection model. We identified that syncytiotrophoblasts could be infected through angiotensin-converting enzyme 2 (ACE2). Using a co-culture model of vertical transmission, we confirmed the ability of the virus to infect syncytiotrophoblasts through a previous endometrial cell infection. We further demonstrated transcriptional changes in infected syncytiotrophoblasts that led to impairment of cellular processes, reduced secretion of HCG hormone and morphological changes vital for syncytiotrophoblast function. Furthermore, different antibody strategies and antiviral drugs restore these impairments. In summary, we have established a scalable and tractable platform to study early placental cell types and highlighted its use in studying strategies to protect the placenta.

In 2019, a betacoronavirus called SARS-CoV-2 emerged causing the COVID-19 pandemic[1]. By April 2023, more than 763 million people were infected with SARS-CoV-2 and over 6 million people died from COVID-19 (https://covid19.who.int/). Although SARS-CoV-2 infection is mild in the majority of cases, patients can develop severe acute respiratory distress syndrome and organ failure[2,3]. As with many other viral infections, the risk of developing severe disease is more likely in pregnant women than in non-pregnant women[4]. Recent reports have shown that pregnant women with a SARS-CoV-2 infection are at an increased risk of having a stillborn or preterm infant, and these negative birth outcomes are exacerbated when maternal SARS-CoV-2 infection occurs earlier in

gestation[5,6]. Vaccination against SARS-CoV-2 during pregnancy provides significant protection against stillbirth and infant death in the first month of life compared with in unvaccinated women[7]. Treatment with the antiviral drug remdesivir during pregnancy or immediately postpartum may also improve COVID-19 recovery rates[8].

ACE2 and transmembrane serine protease 2 (TMPRSS2) have been identified as entry factors for SARS-CoV-2 infection[9]. However, infection can also occur in the absence of TMPRSS2 through endocytosis[10]. The broad expression of both ACE2 and TMPRSS2 means that SARS-CoV-2 can, in theory, infect many organs in addition to the respiratory tract, such as the heart, kidneys and intestines[11–20].

**Fig. 1 | Placental tissue, EVTs and STs express ACE2. a**, Overview of this study using iTSCs to develop an in vitro placental model of SARS-CoV-2 infection. **b,c**, Immunohistochemistry images of first-trimester placental villi (**b**) and maternal decidua (**c**) for ACE2, HCG and HLA-G. Scale bars, 1,000 μm (column 1 and 3), 200 μm (column 2 and 4). **d**, Immunofluorescence analysis of ACE2 (red) along with GATA2 (iTSCs; green), HLA-G (EVTs; green) or HCG (STs; green).

Scale bars, 25 μm. **e**, Immunofluorescence analysis of TMPRSS2 (red) along with GATA3 (iTSCs; green), MMP2 (EVTs; green) or SDC1 (STs; green). Scale bars, 25 μm. **f**, Quantitative PCR (qPCR) analysis of *ACE2* and *TMPRSS2* expression in iTSCs, EVTs, STs and lung AT2 cells (positive control) (fold change relative to iTSCs). For **f**, *n* = 3 independent experiments. Data are mean ± s.e.m., showing variance. No statistical tests were performed.

There is substantial evidence that placental tissue also expresses both ACE2 and TMPRSS2[21,22]. Several studies have detected SARS-CoV-2 virus in placental tissue from infected pregnant women, associated, in some cases, with placental inflammation and pathology[6,21,23–27]. Although vertical transmission of SARS-CoV-2 has been shown to occur, it is a rare occurrence[28–30]. Despite the varying reports of pregnancy loss, especially in the first trimester, the implications of SARS-CoV-2 in the early stages of embryonic, fetal development and placentation are still largely unclear[31–34]. Recent reports suggest that SARS-CoV-2 infection at the maternal–placental interface without full vertical transmission may be sufficient to affect pregnancy and fetal development[35,36]. Furthermore, histopathological reports indicate that villous syncytiotrophoblasts (STs) may be the primary target of infection[24,37,38]. On this note, STs produce human chorionic gonadotropin (HCG) hormone, which is vital for pregnancy[39–43].

Placental in vitro models can provide a great tool to investigate SARS-CoV-2 infection of the placenta. It was shown that SARS-CoV-2 can replicate to varying degrees in placental explants[44]. Moreover, a study confirmed infection of placental clusters and showed an association with an inflammatory response[27]. Although the use of primary placental cells is promising, these models are limited to analysis of at-term placental tissue and tissue donation. The derivation of trophoblast stem cells (TSCs) capable of differentiating into both main placental cell types in vitro—extravillous cytotrophoblasts (EVTs) and STs—facilitates the study of placental biology and pathology[45]. For example, a study used a trophoblast organoid approach, but found limited

infection with SARS-CoV-2[46]. Furthermore, using a model derived from extended pluripotent stem cells (EPSCs) and trophoblast organoids, it was shown that mononuclear STs exhibit susceptibility to SARS-CoV-2 infection with limited infection observed in mature STs, TSCs and EVTs[47]. Thus, to understand the mechanism and implications of SARS-CoV-2 infection during early placentation, further investigation using early placental models is imperative. In this Article, we use induced TSCs (iTSCs) to generate a complex in vitro model of early placental infection by SARS-CoV-2[48,49].

## Results

### STs are productively infected with SARS-CoV-2

To understand the impact of SARS-CoV-2 infection in the placenta, we used iTSCs to generate an in vitro infection model (Fig. 1a). Using first-trimester placenta, we first confirmed the expression of ACE2 and TMPRSS2 within the placental villi, especially in STs lining the villous surface (marked by HCG staining) (Fig. 1b). In the maternal decidua, multiple cells were also faintly stained for ACE2, including EVTs (HLA-G positive) (Fig. 1c), consistent with previous reports[50,51]. By contrast, very little staining of TMPRSS2 was observed in the villi and decidua (Extended Data Fig. 1a). Using trophoblast organoids, it was shown that there is minimal overlap of ACE2 and TMPRSS2 expression[46].

We next examined the expression of ACE2 using our previously reported model of iTSCs before and after differentiation into EVTs and STs using the 55F iTSC line[49,52] (Fig. 1d and Extended Data Fig. 1b). As expected, iTSCs expressed GATA2 and GATA3 (Fig. 1d,e), and EVTs

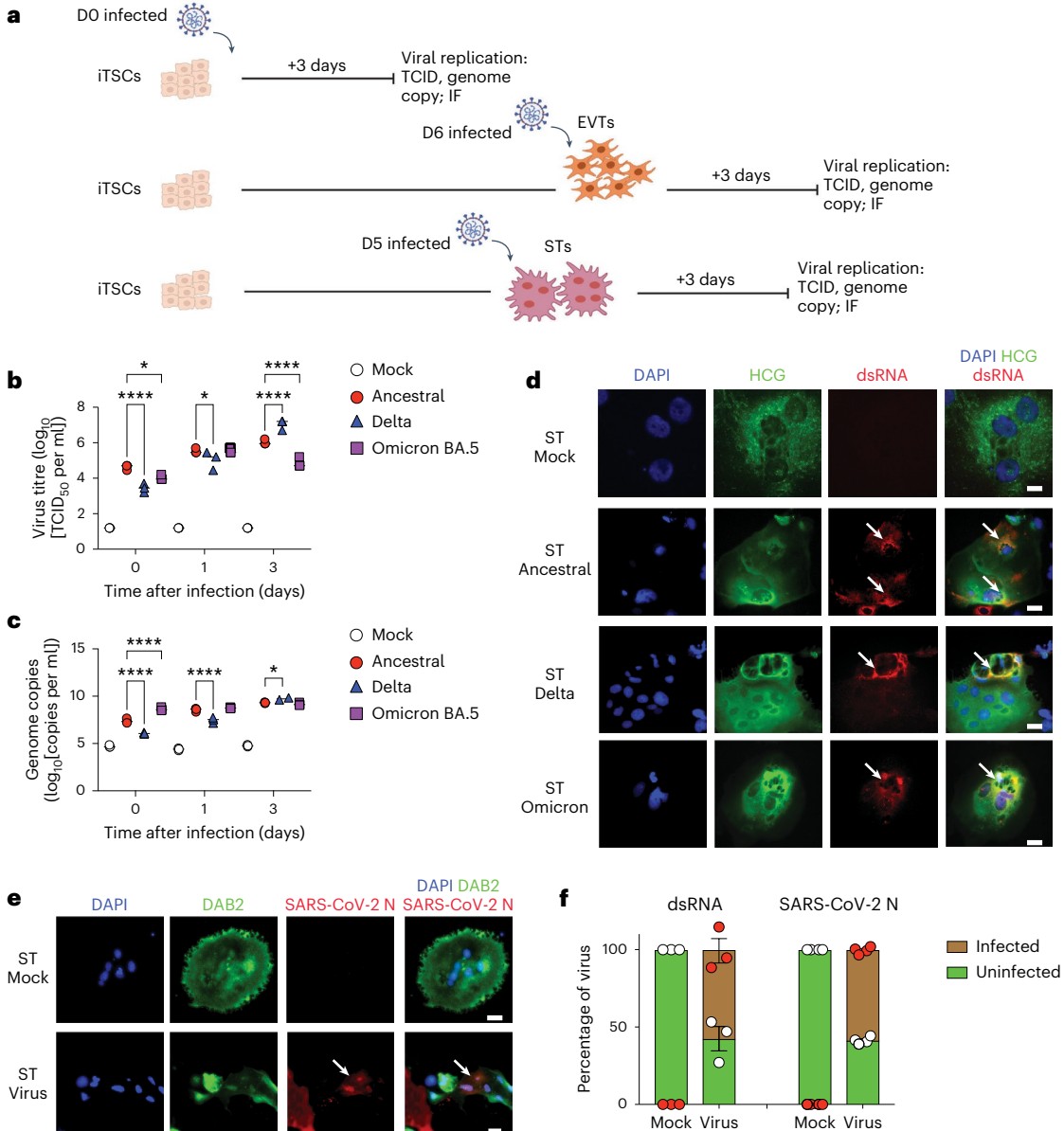

**Fig. 2 | STs are productively infected with SARS-CoV-2. a**, Schematic of differentiation and infection with SARS-CoV-2. IF, immunofluorescence. **b**, Viral titre expressed as the $\log_{10}$-transformed median tissue culture infectious dose ($TCID_{50}$) per ml of infected cultures. **c**, Genome copies expressed as $\log_{10}$-transformed copies per ml of infected cultures. **d,e**, Immunofluorescence analysis (day 3 after infection) of dsRNA (**d**) or SARS-CoV-2 N (**e**) in the 55F cell line infected with either ancestral, Delta or Omicron BA.5 SARS-CoV-2 variants ($10^4$ particles per well at an MOI of 0.26). The white arrows indicate dsRNA or SARS-CoV-2 N. Cells were counterstained with DAPI. For **d** and **e**, scale bars,

$25\,\mu m$. **f**, The percentage of infected ($dsRNA^+$ or SARS-CoV-2 $N^+$) or uninfected ($dsRNA^-$ or SARS-CoV-$2^-$) STs at day 3 after infection. The total number of cells counted was as follows: 1,228 (dsRNA) and 325 (SARS-CoV-2 N) cells. For **b, c** and **f**, representative graphs are shown from 2 independent experiments showing $n = 3$ biological replicates. Statistical analysis was performed using two-way analysis of variance (ANOVA) comparing the infected conditions with the mock control (**b** and **c**) and independent unpaired two-tailed $t$-tests comparing only between virus against mock control (**f**); *$P < 0.05$, ****$P < 0.0001$. Data are mean ± s.e.m.

and STs expressed typical cell-specific markers such as HLA-G/MMP2 and HCG/SDC1, respectively (Fig. 1d,e and Extended Data Fig. 1k). ACE2 expression was observed in both STs and EVTs but not in iTSCs (Fig. 1d). We also detected *ACE2* mRNA in EVTs and STs but not in iTSCs (Fig. 1f). EVTs and STs also expressed higher mRNA levels of *TMPRSS2* relative to iTSCs (Fig. 1f). However, we could not detect TMPRSS2 by immunofluorescence staining in our iTSC, EVT or ST in vitro cultures, in contrast to in the lung AT2 cultures that were used as a positive control (Fig. 1e and Extended Data Fig. 1c). These results mirror those observed in first-trimester placental tissue and suggest that EVTs and STs may be susceptible to SARS-CoV-2 infection.

To identify whether these cells are susceptible to infection, we infected iTSCs, as well as EVTs and STs towards their terminal differentiation at day 6 and day 5, respectively (Fig. 2a), with $10^4$ viral particles of the ancestral (wild type) SARS-CoV-2 virus (equivalent to a multiplicity of infection (MOI) of 0.26). In iTSCs and EVTs, no significant increase in viral titre was observed in the supernatant over time (Extended Data Fig. 1d,f) and no viral double-stranded RNA (dsRNA) was detected in the infected iTSCs and EVTs at day 3 after infection (Extended Data Fig. 1e,g–i). By contrast, viral titres increased in the supernatant of STs by 3 days after infection (Fig. 2b,c) and infection was also confirmed by identification of dsRNA[25] (Fig. 2d,f and

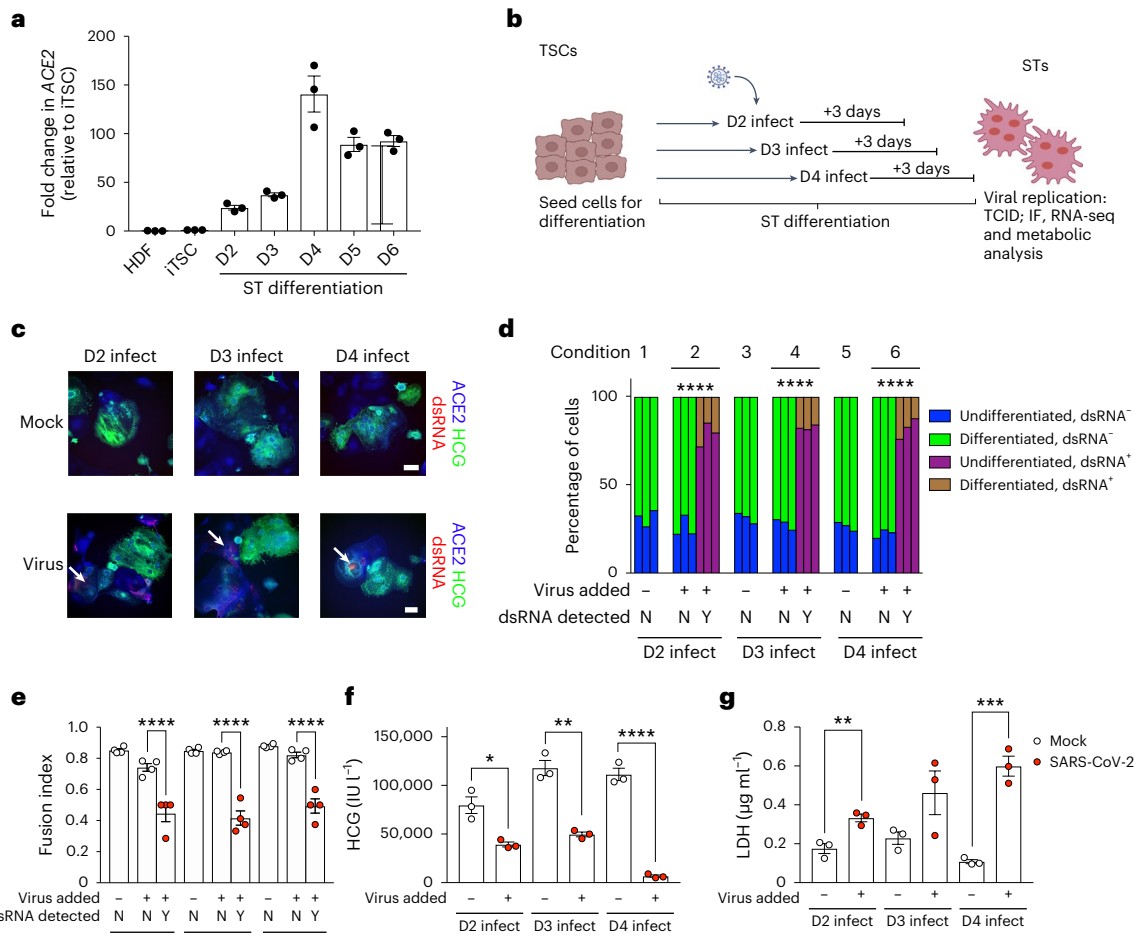

**Fig. 3 | SARS-CoV-2 can infect STs during differentiation and affects differentiation potential, cell death and HCG production. a**, qPCR analysis of *ACE2* expression during ST differentiation (days 2–6) (fold change relative to iTSC). **b**, Schematic of infection of STs during differentiation. **c**, Immunofluorescence analysis of dsRNA (red), HCG (green) and ACE2 (blue) in 55F STs of mock- and virus-infected conditions (days 2, 3 and 4). The white arrows indicate dsRNA. Cells were counterstained with DAPI. Scale bars, 25 μm. **d**, Quantification of cellular differentiation of dsRNA⁺/dsRNA⁻ cells in infected or mock conditions. Percentage comparisons of differentiated and undifferentiated populations are based on each group (with or without virus added and dsRNA⁺ or

dsRNA⁻). The total cells counted was as follows: 582 (condition 1), 506 (condition 2), 815 (condition 3), 640 (condition 4), 558 (condition 5) and 850 (condition 6) cells. N, no; Y, yes. **e**, The fusion index of 55F STs during differentiation. **f**, HCG levels in the supernatant of 55F STs during differentiation. **g**, LDH levels in the supernatant of 55F STs during differentiation. $n = 3$ independent experiments (**a**) and $n = 3$ independent experimental replicates of cells (**d** and **e**). For **f** and **g**, representative graphs are shown from 2 independent experiments showing $n = 3$ biological replicates. For **d**–**g**, independent unpaired two-tailed *t*-tests were used to compare only between virus against mock control at each timepoint; *$P < 0.05$, **$P < 0.01$, ***$P < 0.001$, ****$P < 0.0001$. Data are mean ± s.e.m.

Extended Data Fig. 1j,k). We also confirmed the presence of intracellular SARS-CoV-2 nucleocapsid in STs by immunofluorescence (Fig. 2e). Although STs are productivity infected, the maximal virus titre produced by STs reached approximately 10⁴ to 10⁶ TCID₅ per ml, which was around 10 times lower than that produced by lung AT2 cells[53]. Growth of the Delta and Omicron BA.5 variants in STs was also confirmed (Fig. 2b–d), indicating that newer variants have maintained the ability to infect STs. Positive/productive infection in STs by ancestral, Delta and Omicron BA.5 was also confirmed in two additional donor cell lines (32F and FT008) (Extended Data Fig. 2). Overall, we observed similar growth kinetics between the viral variants (Fig. 2b,c and Extended Data Fig. 2b,d). Taken together, we show that STs can be infected by all major SARS-CoV-2 variants.

### SARS-CoV-2 affects differentiation, cell death and HCG production

As STs were the only cell type in our model that were productively infected by SARS-CoV-2, we focused on this cell type for further study. The placenta during development is a dynamic tissue with constant

turnover[54]. This is especially relevant for STs[55], which are present as a spectrum of undifferentiated, differentiating and differentiated cells within an active syncytial tissue. To determine how early STs could be infected during differentiation from iTSCs, we first analysed the expression of *ACE2* and found that cells begin to express *ACE2* mRNA as early as at day 2 of differentiation (Fig. 3a), which we verified using immunofluorescence staining of both the 32F and 55F cell lines (Extended Data Fig. 3a,b). Moreover, we verified protein expression by western blot analysis of the 55F cell line during differentiation and observed expression of ACE2 from day 2 to day 6 (Extended Data Fig. 3c). In contrast to *ACE2*, no substantial increase in *TMPRSS2* mRNA was observed during ST differentiation (Extended Data Fig. 3d). To test whether the expression of ACE2 was sufficient to confer susceptibility to infection by SARS-CoV-2, we infected 55F STs at days 2, 3 and 4 of differentiation (Fig. 3b) and found that cells could be infected as early as day 2 (Fig. 3c). However, we did not find any correlation between higher viral replication and higher ACE2 expression (Extended Data Fig. 3e).

Infected (dsRNA⁺) cells appeared to be morphologically more immature than non-infected (dsRNA⁻) cells and had lower expression

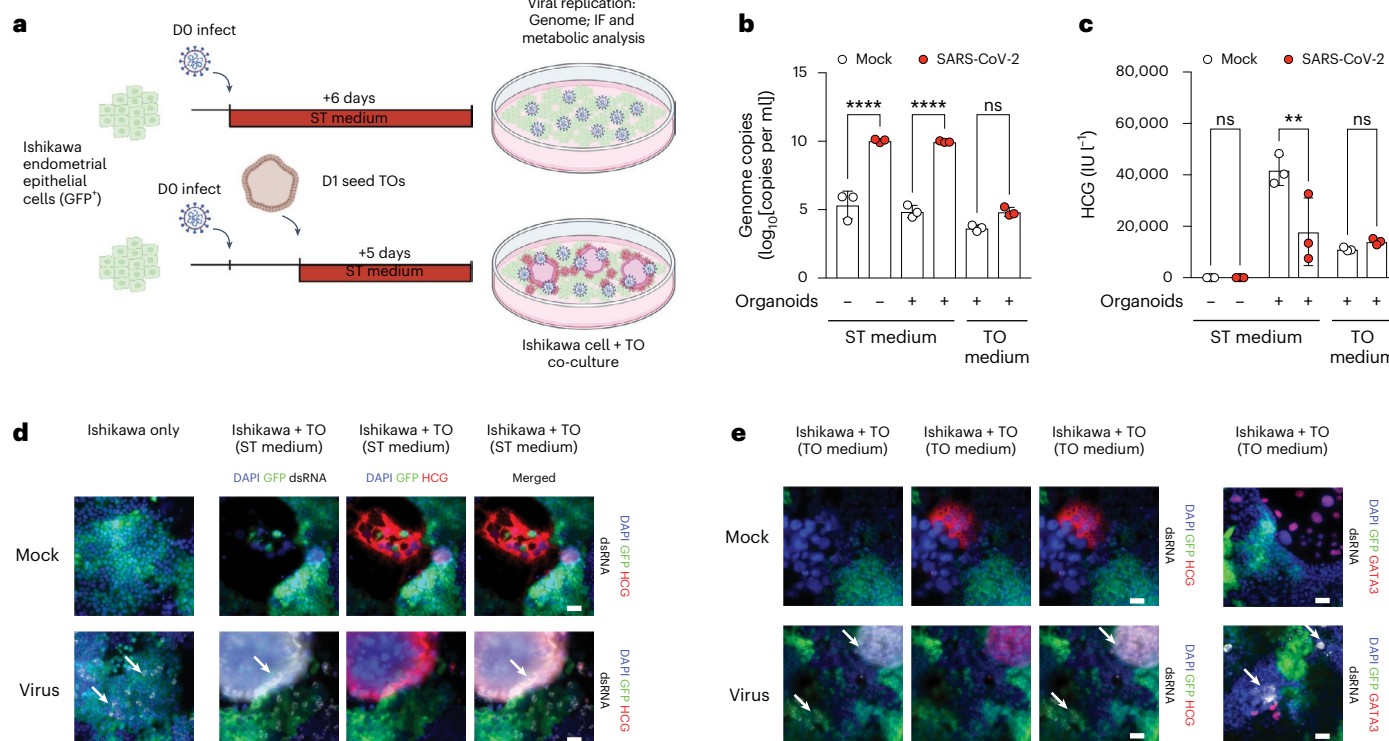

**Fig. 4 | Secondary infection of STs in trophoblast organoids using a co-culture model. a**, Schematic of the co-culture experiment with GFP-positive Ishikawa cells and trophoblast organoids (TOs). **b**, Genome copy analysis of mock- and virus-infected co-culture conditions. **c**, HCG levels in the supernatant of mock- and virus-infected co-culture conditions. **d,e**, Immunofluorescence analysis of dsRNA (white), HCG (red) and GFP (green) in Ishikawa cells and 55F trophoblast organoids cultured in ST medium (**d**) or trophoblast organoid medium (**e**). The white arrows indicate dsRNA. Cells were counterstained with DAPI. For **d** and **e**, scale bars, 25 μm. For **b** and **c**, representative graphs are shown from 2 independent experiments showing n = 3 biological replicates. Statistical analysis was performed using one-way ANOVA comparing infected conditions with the mock control (**b**); **P < 0.01, ****P < 0.0001. Data are mean ± s.e.m.

of HCG despite being multinucleated (Extended Data Fig. 3f). We therefore assessed the levels of HCG and the cellular morphology to quantify the proportions of differentiated and undifferentiated cells (Methods) that were either dsRNA positive or dsRNA negative. We found that the dsRNA⁺ ST cells within a virus-infected culture (cells exposed to the virus and infected) were significantly less differentiated than dsRNA⁻ cells (cells exposed to the virus but not infected) within the same culture (Fig. 3d). Importantly, dsRNA⁻ cells within the virus-infected cultures had a similar differentiation potential to the mock-infected cells. The dsRNA⁺ cells within the infected culture also appeared to demonstrate a significantly lower fusion index compared with the dsRNA⁻ cells, corroborating the observation that virus-infected cells have impaired differentiation (Fig. 3e). Taken together, these results suggest that SARS-CoV-2 infection hinders the differentiation of STs and impaired cell fusion.

As HCG levels increase as differentiation progresses, we next assessed HCG levels in response to infection. We observed that HCG levels were significantly lower in the infected cells throughout differentiation (days 2, 3 and 4) compared with in the mock controls (Fig. 3f). This reinforces differentiation and function impairment of differentiating STs in SARs-CoV-2-infected cultures. We next analysed cytotoxic stress using a lactate dehydrogenase (LDH) assay and found that infected cells released higher levels of LDH compared with the mock controls, indicating that there were significantly higher levels of cytotoxic stress in the SARS-CoV-2 infected cultures (Fig. 3g). Although LDH levels were increased, the overall loss of cell viability by day 3 after infection was modest (Extended Data Fig. 3g). However, we did observe a trend of increased caspase 3/7 activity after infection (Extended Data Fig. 3h). Although these observations were not statistically significant, they

suggest a role for apoptosis-mediated cell death in STs after SARS-CoV-2 infection, albeit only in a small proportion of cells.

As differentiating STs are susceptible to SARS-CoV-2 infection, we established an in vitro vertical or secondary infection model to determine whether infected endometrial cells could subsequently infect bystander STs (Fig. 4a and Extended Data Fig. 3i). After infection of the endometrial epithelial cells (Ishikawa cells), we co-cultured trophoblast organoids generated from our 55F iTSC line in either ST medium (facilitating ST differentiation of trophoblast organoids) or in trophoblast organoid medium (to maintain trophoblast organoids). We observed a significant increase in viral genomes in ST medium compared with in trophoblast organoid medium (Fig. 4b and Extended Data Fig. 3i). Importantly, HCG production was significantly impaired in infected differentiated trophoblast organoids, mirroring our findings in the monoculture system (Fig. 4c). These results were further confirmed by the presence of dsRNA (Fig. 4d,e). Taken together, SARS-CoV-2 infection of STs generated in 2D or through 3D trophoblast organoids could occur directly or through vertical transmission, leading to an impairment of differentiation potential, a lack of HCG production and a modest increase in cell death during differentiation.

### Infected STs show an increase in viral responses
To understand the effects of infection at a greater depth, we analysed the transcriptome of ST cells infected at day 3 of differentiation and analysed the cells at day 3 after infection (3 days differentiation + 3 days after infection) to reach the theoretical full differentiation time course (day 6). Correspondence analysis (CoA) indicated that SARS-CoV-2-infected cells were transcriptionally divergent from mock-infected cells (Fig. 5a). Differential gene expression (DGE) analysis identified

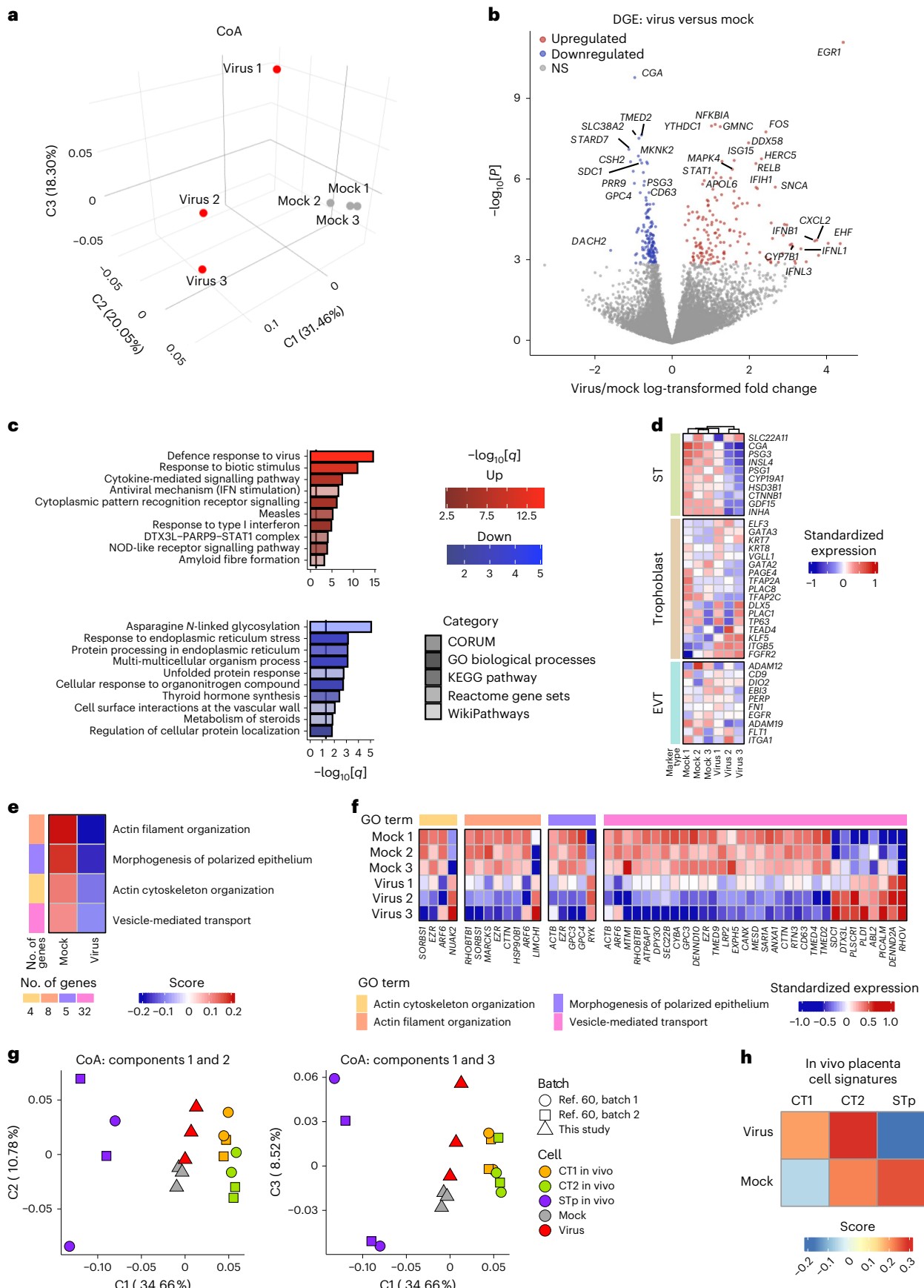

**Fig. 5 | The transcriptome profiles of infected STs show an increase in viral responses. a**, Transcriptome CoA of virus and mock conditions of STs at day 3 of infection. **b**, DGE analysis between the virus and mock conditions. **c**, Functional enrichment analysis of significantly upregulated or downregulated gene sets (false-discovery rate < 0.05). GO, Gene Ontology. **d**, Hierarchically clustered heat map of cell-identity genes expressed (genes of EVT, trophoblast and ST) in STs under virus (virus 1, virus 2, virus 3) and mock (mock 1, mock 2, mock 3)

conditions. **e**, Gene signature scores for DGEs (virus versus mock) associated with cytoskeleton-related GO terms. **f**, The DGEs between the virus and mock conditions within the cytoskeleton-related GO terms. **g**, Components 1, 2 and 3 of the transcriptome-wide CoA of STs at day 3 of infection under the virus and mock conditions, along with in vivo CT and STp cells. **h**, Gene set variation analysis gene signature scores of the virus and mock samples for the in vivo CT and STp signatures.

155 genes that were upregulated and 140 genes that were downregulated in infected cells (Supplementary Table 1). Importantly, we identified that, among these differentially expressed genes, ST-specific genes such as *CGA* and *PSG3* were significantly downregulated, consistent with an impairment in differentiation (Fig. 5b). Upregulated genes were related to interferon signalling (*IFNL1*, *IFNB1*, *IFIH1*) (Fig. 5b) and TNF signalling through NF-κB, such as *MAPK4*, *STAT1*, *RELB and NFKBIA*, indicating an innate response to viral infection[56]. Furthermore, Gene Ontology analysis showed an enrichment of viral response, along with antiviral mechanism (IFN stimulation) and response to type I interferon in the significantly upregulated genes in infected cells. By contrast, genes downregulated after infection were enriched in cellular and metabolic processes (Fig. 5c). Although not strongly enriched, we did identify 14 significantly upregulated genes that are involved in positive regulation of cell death, such as *EGR1*, *FOS*, *SNCA* and *PHLDA1*. Moreover, expression levels of TSC-identity-specific (trophoblast: *ELF3*, *GATA3*, *KRT7*, *KRT8*), EVT-identity-specific (EVT: *ADAM12*, *CD9*, *DIO2*, *EBI3*) and ST-identity-specific (ST: *SLC22A11*, *CGA*, *PSG3*, *INSL4*, *PDG1*) genes showed that, although mock-infected cells have robust expression of ST-related differentiation genes, SARS-CoV-2-infected cultures did not express these genes and had higher expression levels of TSC-related genes. This suggested that cells were less differentiated, consistent with our previous observations (Fig. 5d). Taken together, we show that SARS-CoV-2 infection of ST cultures elicits an NF-κB-mediated inflammatory response and has a negative effect on the differentiation pathway of cells. We also observed downregulation of genes associated with 'actin cytoskeleton organization', 'actin filament organization', 'morphogenesis of polarized epithelium' and 'vesicle mediated transport' (Fig. 5e). Particularly, the genes *ACTN2*, *PKD2*, *GSN* and *CDH1* (Supplementary Table 1) were all downregulated in infected samples, consistent with reports of impairment of ST formation and cytoskeleton regulation[57–59] (Fig. 5f). This is consistent with the poor and undifferentiated ST morphology as described in Fig. 2d.

To further examine the impairment of differentiation potential, we next compared our RNA-sequencing (RNA-seq) data with single-cell RNA-seq (scRNA-seq) data of in vivo first-trimester placental cells[60] containing placenta developmental cytotrophoblasts (CTs) at several developmental stages, EVTs, EVT progenitors and ST progenitor (STp) cells (Extended Data Fig. 4a–c), with the starting population

determined as CT1 (Methods). Single-cell populations within the CT to STp trajectory were then pseudobulked and integrated with our RNA-seq data. The integration showed that SARS-CoV-2-infected cells clustered closer to the CTs (less differentiated cells) compared with the mock-infected cells (which cluster nearer to the STp cells) (Fig. 5g). Furthermore, cell score analysis revealed an enrichment for an undifferentiated CT signature in the infected samples, in contrast to in the mock samples, which were enriched for STp (Fig. 5h).

## Inhibition of ACE2 prevents viral infection of STs

Finally, we investigated whether ACE2 could be targeted to inhibit SARS-CoV-2 entry into STs using anti-ACE2 antibodies. We generated and characterized antibodies against recombinant human ACE2 from a phage library (Extended Data Fig. 5a). We then validated the binding affinity of these clones and selected WCSL141 and WCSL148 for the blocking experiments (Extended Data Fig. 5b,c). We showed that ancestral SARS-CoV-2 infection of STs at both day 3 and 5 of differentiation was blocked by these two anti-ACE2 antibodies (virus^anti-ACE2) (Fig. 6a). We did not detect infectious virus or viral genome copies in the culture supernatants, whereas both were detected in virus-infected conditions treated with a control antibody (virus^Ctrl), demonstrating that ACE2 antibody inhibits virus entry (Fig. 6b). Similar inhibition with our anti-ACE2 antibodies was observed for infection with Omicron BA.5 (Fig. 6c,d). Inhibition of infection was superior when using a combination of these anti-ACE2 antibodies compared with using either of the antibodies individually (Fig. 6c,d). Ancestral SARS-CoV-2 infection could also be inhibited with anti-spike antibodies from Regeneron (imdevimab and casirivimab; also known as REGN10987 and REGN10933, respectively) and CR3022 (Fig. 6c,d). By contrast, Omicron BA.5 escaped inhibition by CR3022 and REGN10987 (Fig. 6c,d). This is consistent with other published studies showing escape from monoclonal antibody neutralization by Omicron variants[61–63]. Taken together, we demonstrated that SARS-CoV-2 infection of ST cells could be prevented using anti-ACE2 antibodies, anti-spike antibodies or treatment with antiviral compounds.

We next analysed the transcriptomes of cells treated with our anti-ACE2 antibodies after infection. Hierarchical clustering and CoA of samples showed that cultures that were blocked with anti-ACE2 antibodies (virus^anti-ACE2) clustered closely with mock-infected

**Fig. 6 | Inhibition of ACE2 using anti-ACE2 antibodies restores normal differentiation and function of STs. a**, Immunofluorescence analysis of dsRNA (red), ACE2 (blue) and HCG (green) in STs treated with anti-ACE2 antibodies (or isotype control (Ctrl)) at day 3 after infection, at day 3 or day 5 of ST differentiation. The white arrows indicate dsRNA. Cells were counterstained with DAPI. Scale bars, 25 μm. **b**, Virus titre at day 3 after infection expressed as log_{10}[TCID_{50} per ml] (left) and genome copies per ml of antibody-treated and/or virus-infected STs at day 3 and day 5 of differentiation (right). **c,d**, The virus titre at day 3 post infection of antibody-treated STs at day 3 of differentiation expressed as log_{10}[TCID_{50} per ml] (**c**) and genome copies in the supernatant (log_{10}[copies per ml]) (**d**). WCSCL141 and WCSL148 are anti-ACE2 antibodies; REGN10987, REGN10933 and CR3022 are anti-SARS-CoV-2 spike antibodies. **e**, Transcriptome-wide hierarchical clustering analysis at day 3 post infection in STs under the virus-infected (virus^Ctrl), mock-infected (mock^Ctrl) and treated (mock^anti-ACE2, virus^anti-ACE2) conditions. The y-axis represents Canberra distance. **f**, Transcriptome-wide *k*-means clustering analysis at day 3 in STs under the indicated conditions (clusters 1–5 (C1–C5)). **g**, Hierarchically clustered heat map

of cell-identity genes (genes of ST, EVT and trophoblast) expressed in STs under the indicated conditions. **h**, Gene set variation analysis gene signature scores of day 3 post infection ST samples (under the indicated conditions) for the in vivo CT and STp signatures. **i**, Quantification of cellular differentiation of dsRNA^+/dsRNA^- STs under the indicated conditions. The total cells counted was as follows: 335 (condition 1), 378 (condition 2), 278 (condition 3) and 312 (condition 4) cells. **j**, HCG levels in the supernatant of STs under the indicated conditions. **k**, LDH levels in the supernatant of STs under the indicated conditions. For **b–d**, **j** and **k**, representative graphs are shown from 2 independent experiments showing *n* = 3 biological replicates. For **b**, **j** and **k**, *n* = 2 for the day 3 mock^Ctrl group. For **i**, *n* = 3 independent experimental replicates of cells. Statistical analysis was performed using one-way ANOVA comparing the infected conditions with the mock control (**c**, **d** and **i**), independent unpaired two-tailed *t*-tests comparing virus^Ctrl against virus^anti-ACE2 (*n* = 3 samples) (all graphs in **b**) and unpaired two-tailed independent *t*-tests comparing only virus^Ctrl against mock^anti-ACE2 and virus^anti-ACE2 (*n* = 3 samples) (**j** and **k**); *$P$ < 0.05, **$P$ < 0.01, ***$P$ < 0.001, ****$P$ < 0.0001. Data are mean ± s.e.m.

samples and were separated from infected cells treated with a control antibody (virus^Ctrl) (Fig. 6e and Extended Data Fig. 6a). As expected, we observed high expression of SARS-CoV-2 RNA under the virus^Ctrl condition, but not under the mock or in virus^anti-ACE2

conditions (Extended Data Fig. 6b,c). Unsupervised *k*-means clustering identified unique clusters of genes that were upregulated and downregulated in the virus^Ctrl condition in contrast to in the other samples (Fig. 6f and Supplementary Table 1). Functional enrichment analysis

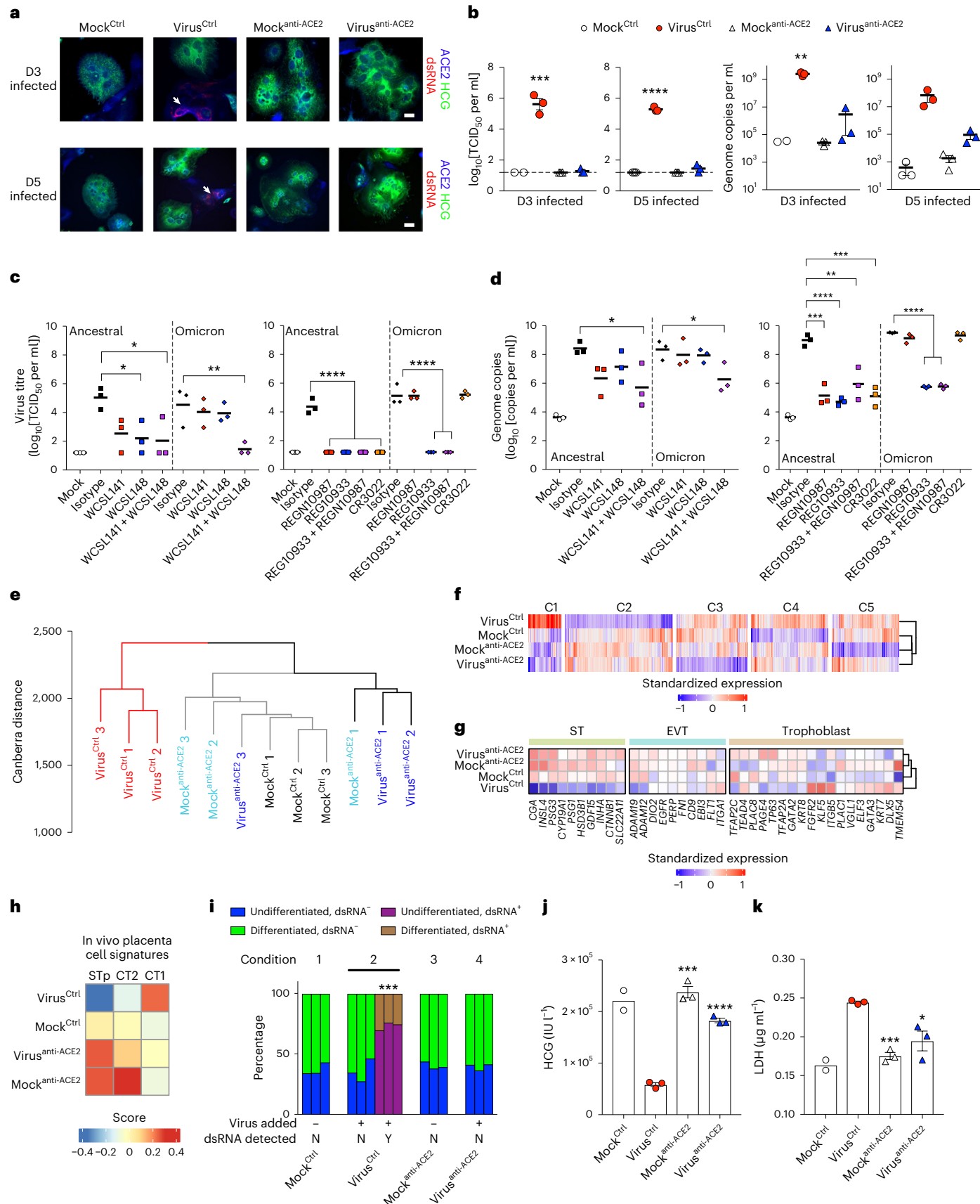

of clusters 1 and 2 showed upregulation of host-defence response to virus and IFN signalling pathways (cluster 1), and downregulation of cellular and metabolic processes (cluster 2) (Extended Data Fig. 6d). We next examined placental-identity genes of TSCs, EVTs and STs and found that virus[anti-ACE2] cultures exhibited an increase in expression of ST-identity genes, similar to in the mock-infected controls (Fig. 6g). Integration analysis with publicly available scRNA-seq data from in vivo placenta (as described in the previous section) showed that the virus[Ctrl] cells cluster closer to the CTs, whereas infected cells treated with anti-ACE2 antibodies clustered along with the mock[Ctrl] samples and further away from the CTs (Extended Data Fig. 6e,f). Furthermore, cell score analysis revealed that only the infected cells (virus[Ctrl]) samples were stalled during differentiation, exhibiting strong CT1 enrichment, whereas all of the other samples showed enrichment of the ST signature (Fig. 6h). We also identified that a positive enrichment of the mock[anti-ACE2] samples for the CT2 signature is due to subtle differences in a small fraction of the genes within that signature. Meanwhile, the virus[Ctrl] sample showed a robust upregulation and downregulation of the CT1 and STp signature scores, respectively, with respect to the virus[anti-ACE2], mock[anti-ACE2] and mock[Ctrl] samples. (Extended Data Fig. 7a). Furthermore, hierarchical clustering of the samples for each signature showed that the mock[anti-ACE2], mock[Ctrl] and virus[anti-ACE2] samples are always clustered together, which is consistent with our findings and, overall, we did not notice an impact on the cellular differentiation. We also confirmed that anti-ACE2 antibody blocking rescued the differentiation potential and HCG and LDH levels of infected ST cells (Fig. 6i–k).

Finally, we determined the effectiveness of the antiviral compounds remdesivir or molnupiravir[64] against SARS-CoV-2 viral replication in ST cells. We observed that both of the antiviral compounds were effective against SARS-CoV-2 infection in ST cells at 1 μM, with minimal drug toxicity (Extended Data Fig. 6g,h), and restored the levels of HCG production compared with in the control infected cultures (DMSO) (Extended Data Fig. 6h). These data indicate that antiviral compounds that inhibit viral replication can also restore ST function after SARS-CoV-2 infection (Extended Data Fig. 6g).

## Discussion

In vitro models are a great tool to understand inherent cellular and molecular mechanisms of SARS-CoV-2 infection and test specific drug treatments[65–67]. Here we found that SARS-CoV-2 can infect STs but not TSCs or EVTs. Similar low infection rates in TSCs (2–3% naive TSCs) and EVTs (1–2%) were found previously[47]. Although EVTs were not infected by SARS-CoV-2 in our model, others have reported that these cells are susceptible to viruses like adenovirus[68]. The reason for the lack of SARS-CoV-2 infection in EVTs despite ACE2 expression is unclear and will require further investigation. Interestingly, Ruan et al.[47] reported an infection rate of around 10% at day 1 of differentiating STs in 2D, and Karvas et al.[46] reported that SARS-CoV-2 was detected only in a few CTBs and a fraction of multinucleated ST-like cells in their undifferentiated trophoblast organoid system and 2D ST models. In contrast to these two previous studies, in addition to demonstrating a robust infection of STs in 2D models, we also demonstrated that trophoblast organoids can be differentiated into STs and infected by SARS-CoV-2 after differentiation. We found that SARS-CoV-2 infection in STs was around 57% at day 3 (measured by dsRNA and SARS-CoV-2 N). This proportion discrepancy between studies probably reflects differences in models, differentiation, the differentiation stage of the STs and the timing of analysis after infection[47]. Our findings are consistent with histopathological studies using clinical samples reporting that STs in the intervillous space are the typical cells that harbour SARS-CoV-2 in infected placentas[24,37,38,69,70]. Our co-culture vertical-transmission model suggests that STs can also be secondarily infected from maternal cells. This provides a possible explanation for recent clinical reports of patients with vertical transmission of COVID-19 to liveborn and stillborn infants showing placental necrosis specifically in the intervillous space and in STs of placentas[69,71].

Similar to that shown previously[47], we found that STs are susceptible to infection by several virus variants early into differentiation (2 days of differentiation, at the mononuclear stage). However, our study expands on this, showing that infection can occur also late in ST differentiation (polynuclear stage), which indicates that infection could occur at different stages of ST development. Growth in STs was lower in comparison to in previous studies of lung AT2 cells[53]. Viral replication in lung AT2 cells, in contrast to in STs, is dependent on TMPRSS2 and induces minimal interferon signalling. It is possible that these factors contribute to the degree of replication observed in these cell types. Infection, in both our study and a previous study[47], was associated with a blockade of differentiation and upregulation of genes associated with response to viral infection. However, we also observed upregulated genes associated with cellular structure/function, providing a possible molecular explanation for the observed impairment in ST differentiation and morphology after SARS-CoV-2 infection. We found that anti-viral treatments, such as remdesivir and anti-ACE2 antibodies, prevent infection, which is in agreement with *ACE2* knockout being refractory to SARS-CoV-2 infection[47]. Importantly, we found that anti-ACE2 antibody therapy and antivirals restore proper HCG levels and lower cell death. We also showed that the combination of different anti-ACE2 and anti-spike antibodies is vital for the prevention of infection with Omicron variants, in contrast to ancestral SARS-CoV-2.

The impaired differentiation potential of infected STs was manifested by the reduction in fusion index and HCG production. As iTSC-derived STs are a model of early placentation, these results support and provide an explanation for clinical evidence that the SARS-CoV-2 may affect the placenta in early development[72]. Specifically, reduced HCG production may be associated with complications in pregnancy, including early miscarriage[15]. An advantage of our in vitro infection system is the ability to identify differences in morphological phenotypes between dsRNA+ and dsRNA− cells within the same culture system. The acquisition of typical differentiation features such as cell fusion is still observed in dsRNA+ cells; however, further morphological aspects of differentiation, such the foot processes/microvilli, and the aforementioned progressive increase in HCG levels are not observed within these cells. These are vital for the function of the anchoring villi between the fetal–maternal interface and their disruption can lead to complications[73–75]. Impaired ST morphology and function in vivo has been reported to lead to pre-eclampsia—a complication that is observed at a higher incidence in pregnant women with COVID-19[76–79].

Through comparison with in vivo placental scRNA-seq datasets, we confirmed that infected STs were less differentiated than mock-infected STs and more similar to CTs. Importantly, changes in cellular-identity genes in the placentas of patients with COVID-19 have been observed previously[27]. Consistent with other reports of SARS-CoV-2 infection in a variety of cell types, we identified upregulation of viral-response and innate-immunity genes. A robust inflammatory response was also observed after SARS-CoV-2 infection of placental explants and in placentas of patients with COVID-19[27]. Importantly, SARS-CoV-2 infection elicited an IFN response in our model, similar to responses in other cell types such as lung and cardiac cells[65,80,81]. For other viruses such as Zika virus that infect the placenta, IFN production, particularly by STs, is critical to protect against viral replication[82,83]. Whether there is any role for IFNs in restricting SARS-CoV-2 infection or whether SARS-CoV-2 can subvert the IFN response in these cells requires further investigation.

As demonstrated by our results, iTSCs can be of great use to established placental models of infection; however, they are still reductionist approaches and they do not completely address infection in the context of the high complex placental biology and pathophysiology. Our co-culture system of iTSC-derived organoids and endometrial epithelial-like cells made progress towards establishing a multicellular complex model for placental infection during implantation, modelling

the potential of vertical transmission and highlighting the importance of different methods and models to describe and understand these processes. In the future, we envision that these models will serve as a platform for further improvements and include, for example, immune cells, as we and others have done in other models derived from induced pluripotent stem cells[84]. Finally, we anticipate that in vitro models of the placenta, such as the one used here, will be used to facilitate a deeper understanding of COVID-19 pathogenesis and provide a platform for drug discovery and potential treatments.

## Online content

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

[1]Department of Anatomy and Developmental Biology, Monash University, Clayton, Victoria, Australia. [2]Development and Stem Cells Program, Monash Biomedicine Discovery Institute, Clayton, Victoria, Australia. [3]Australian Regenerative Medicine Institute, Monash University, Clayton, Victoria, Australia. [4]Department of Microbiology and Immunology, The University of Melbourne at the Peter Doherty Institute for Infection and Immunity, Melbourne, Victoria, Australia. [5]Adelaide Centre for Epigenetics, Faculty of Health and Medical Sciences, The University of Adelaide, Adelaide, South Australia, Australia. [6]South Australian Immunogenomics Cancer Institute, Faculty of Health and Medical Sciences, The University of Adelaide, Adelaide, South Australia, Australia. [7]Implantation and Pregnancy Research Laboratory, School of Health and Biomedical Sciences, RMIT University, Melbourne, Victoria, Australia. [8]Infectious Diseases and Immune Defences Division, The Walter and Eliza Hall Institute of Medical Research, Parkville, Victoria, Australia. [9]Department of Medical Biology, University of Melbourne, Melbourne, Victoria, Australia. [10]University of Melbourne Centre for Cancer Research, The University of Melbourne, Melbourne, Victoria, Australia. [11]School of Life Sciences, Westlake University, Hangzhou, China. [12]Research Center for Industries of the Future, Westlake University, Hangzhou, China. [13]Westlake Laboratory of Life Sciences and Biomedicine, Hangzhou, China. [14]Westlake Institute for Advanced Study, Hangzhou, China. [15]WHO Collaborating Centre for Reference and Research on Influenza, Melbourne, Victoria, Australia. [16]These authors contributed equally: J. Chen, J. A. Neil, J. P. Tan, R. Rudraraju. ✉e-mail: kanta.subbarao@influenzacentre.org; jose.polo@monash.edu

## Methods

### Ethics statement

Ethics approval (RES-19-0000-399A) for the use of first trimester human placental tissues for research was obtained from the Human Ethics Committee at Monash Health, in accordance with the guidelines of the National Health and Medical Research Council of Australia. Women undergoing elective termination of pregnancy were recruited, with exclusion criteria of known fetal abnormalities or participants younger than 18 years. All of the participants provided informed written consent on placental tissue donation, and there was no self-selection bias or other biases that may affect the result.

### Placental tissue preparation and Immunohistochemistry

First-trimester placental tissues were collected after elective pregnancy termination with full ethics approval and written consent. Tissues were then de-identified, fixed in buffered formalin and embedded in paraffin. Tissues were sectioned (5 μm) onto SuperFrost Plus slides (Thermo Fisher Scientific), dried overnight at 37 °C, deparaffinized in histolene, then rehydrated in graded solutions of ethanol to Milli-Q water. After antigen retrieval by microwaving for 10 min in 0.01 M citrate buffer (pH 6.0), endogenous peroxidase was quenched with 3% $H_2O_2$, and tissues were next incubated with a blocking buffer containing high-salt TBS (0.3 M NaCl in 50 mM Tris, pH 7.6), 0.1% Tween-20 (Sigma-Aldrich), and 15% goat or horse serum for 20 min at room temperature. The sections were then incubated with primary antibodies (ACE2, ab15438, Abcam, 1:200; HCG beta, ab9582, Abcam, 1:200; HLA-G, ab7759, Abcam, 1:50) for 1 h at 37 °C; rabbit or mouse IgG (X0936 and X0931 respectively, Dako, both 1:10,000) was used for the negative controls. The sections were next incubated with a biotinylated goat anti-rabbit IgG or horse anti-mouse IgG (BA-1000 and BA-2000 respectively, Vector Laboratories, both 1:1,000), then with an avidin–biotin-complex conjugated to horseradish peroxidase (Vector Laboratories), each for 30 min at room temperature. All antibodies were incubated in blocking buffer containing TBS, 0.1% Tween-20 and 10% FBS (Thermo Fisher Scientific). Colour was developed with peroxidase substrate 3,3′-diaminobenzidine (DAB) (Dako). The sections were counterstained with Harris haematoxylin (Sigma-Aldrich), mounted in DPX new mounting medium (Sigma-Aldrich) and imaged using the Olympus microscope fitted with a Fujix HC-2000 high-resolution digital camera (Fujix).

### Cell culture and differentiation

iTSCs (32F, 55F) and trophoblast stem cell (FT008) lines were maintained as described previously[45,49]. In brief, iTSCs were cultured on 5 μg ml⁻¹ collagen-IV-coated (Sigma-Aldrich) plates in TSC medium (DMEM/F-12 GlutaMAX (Thermo Fisher Scientific) supplemented with 0.3% BSA (Sigma-Aldrich), 0.2% FBS, 1% ITS-X supplement (Thermo Fisher Scientific), 0.1 mM 2-mercaptoethanol (Thermo Fisher Scientific), 0.5% penicillin–streptomycin (Thermo Fisher Scientific), 1.5 μg ml⁻¹ L-ascorbic acid (Sigma-Aldrich), 5 μM Y27632 (Selleckchem), 2 μM CHIR99021 (Miltenyi Biotec), 0.5 μM A83-01 (Sigma-Aldrich), 1 μM SB431542, 50 ng ml⁻¹ EGF (Peprotech) and 0.8 mM VPA (Sigma-Aldrich)). iTSCs were passaged every 4–5 days with medium replacement every other day.

Differentiation of iTSCs into STs and EVTs was performed and modified as previously described[45]. For the differentiation of iTSCs into STs, iTSCs were seeded at a density of $3.75 \times 10^4$ cells per well onto a 24-well plate precoated with 2.5 μg ml⁻¹ collagen IV and cultured in ST differentiation medium (DMEM/F-12, GlutaMAX supplemented with 0.3% BSA, 4% KSR (Thermo Fisher Scientific), 1% ITS-X supplement, 0.1 mM 2-mercaptoethanol, 0.5% penicillin–streptomycin, 2.5 μM Y27632 and 2 μM forskolin (Selleckchem)). The medium was replaced every 3 days. For the differentiation of iTSCs into EVTs, iTSCs were seeded at a density of $3.4 \times 10^4$ cells per well onto a 24-well plate precoated with 1 μg ml⁻¹ collagen IV and cultured in EVT differentiation medium (DMEM/F-12, GlutaMAX supplemented with 0.3% BSA, 4% KSR, 1% ITS-X supplement, 0.1 mM 2-mercaptoethanol, 0.5% penicillin–streptomycin, 2.5 μM Y27632, 100 ng ml⁻¹ hNRG1 (Cell Signaling Technology), 7.5 μM A83-01 and 2% Matrigel (Corning). On day 3 of differentiation, the medium was replaced with EVT differentiation medium without hNRG1, and Matrigel was added to a final concentration of 0.5%. On day 6 of differentiation, EVT differentiation medium was replaced without hNRG1 and KSR, and Matrigel was added to a final concentration of 0.5%.

### Generation of trophoblast organoids and secondary infection model

To generate trophoblast organoids (55F cell line), $1 \times 10^4$ iTSCs were seeded per well of a 12-well plate in Matrigel and cultured in trophoblast organoid medium for 7–10 days. The trophoblast organoids were passaged every 7 days with mechanical pipetting. Trophoblast organoid medium contained Advanced DMEM/F12 supplemented with 1% N2 supplement (Thermo Fisher Scientific), 2% B27 supplement minus vitamin A (Thermo Fisher Scientific), 1% penicillin–streptomycin, 1.25 μM N-acetyl-L-cysteine (Sigma-Aldrich), 2 mM L-glutamine (Thermo Fisher Scientific), 0.5 μM A83-01, 1.5 μM CHIR99021, 50 ng ml⁻¹ EGF, 80 ng ml⁻¹ R-spondin-1 (Peprotech), 100 ng ml⁻¹ FGF2 (Miltenyi Biotec), 50 ng ml⁻¹ HGF (Peprotech), 2.5 μM PGE2 (Sigma-Aldrich) and 2 μM Y-27632 (ref. [85]).

To establish the secondary infection model, GFP-positive (lentivirus transduction with GFP) Ishikawa cells (Sigma-Aldrich, 99040201) were first seeded at a density of $1 \times 10^4$ cells per well onto a 24-well plate in MEM-alpha (Thermo Fisher Scientific) supplemented with 2 mM L-glutamine, 1% MEM Non-Essential Amino Acids Solution (Thermo Fisher Scientific), 1% penicillin–streptomycin and 5% FBS. Then, 24 h later, the Ishikawa cells were infected with SARS-CoV-2 accordingly as described in the 'SARS-CoV-2 infection' section below. Another 24 h later, 20–30 organoids were seeded onto the infected Ishikawa cells in trophoblast organoid medium or ST medium and further cultured for 5 days before being collected for analysis. All plasmids are available on request (GFP lentiviral plasmid).

### SARS-CoV-2 infection

iTSCs were infected in their undifferentiated state at day 0, EVTs were infected at day 6 of differentiation and STs were infected at day 2 to day 5 of differentiation. Vero (ATCC, CCL-81), Vero E6-TMPRSS2 (CellBank Australia, JCRB1819) and Calu-3 (ATCC, HTB-55) cells were used to propagate the SARS-CoV-2 viruses. Titration of virus stocks to estimate their $TCID_{50}$ was done either in Vero or Vero E6-TMPRSS2 cells. Placental cells in 24-well plates were infected in duplicate or triplicate with $10^4$ $TCID_{50}$ (as determined by titration in Vero cells) of ancestral SARS-CoV-2 (Australia/VIC01/2020, WT) for 1 h. In some experiments, STs were infected in triplicate with $10^5$ $TCID_{50}$ (as determined by titration in Vero E6-TMPRSS2 cells) of ancestral, Delta (Australia/VIC/18440/2021, B.1.617.2) or Omicron (Australia/VIC/61194/2022, BA.5) variants for 1 h. Virus inoculum was removed, and cells were cultured in cell-type-specific medium for up to 3 days. The supernatants were collected and the medium was replaced daily. $TCID_{50}$ in the supernatants was determined by tenfold serial dilution in Vero cells (experiments using ancestral virus only) or Vero E6-TMPRSS2 cells (experiments that included Omicron) and calculated using the Reed and Muench method. RNA was extracted from the supernatants using the QIAamp Viral RNA mini kit (Qiagen) and E gene expression assessed using the SensiFAST Probe No-Rox One Step Kit (Bioline) and the following primers/probes: Fwd, 5′-ACAGGTACGTTAATAGTTAATAGCGT-3′; Rev, ATATTGCAGCAGTACG-CACACA; and Probe, FAM-ACACTAGCCATCCTTACTGCGCTTCG-BBQ. Viral genomes were interpolated using a standard curve generated by a plasmid containing the E gene. Where indicated, cells were infected with the icSARS-CoV-2-nLuc reporter virus (donated by Ralph S. Baric). Each experiment was repeated independently at least twice.

The presence of SARS-CoV-2 dsRNA or SARS-CoV-2 nucleocapsid in infected cells is evidence of active viral replication within the cells[72].

## Phage library isolation of anti-ACE2 monoclonal antibodies

Biopanning for anti-ACE2 human antibodies using the CSL human antibody phage library was performed as previously described[86]. Phages displaying human Fabs were enriched after three rounds of biopanning on biotinylated recombinant human ACE2 immobilized to streptavidin Dynabeads (Dynal M-280,112.06D, Invitrogen). After the third round of panning, individual clones were selected for further analyses by enzyme-linked immunosorbent assay (ELISA) for the presence of human ACE2-binding phage. Positive clones were sequenced and annotated using the International ImMunoGeneTics database and aligned in Geneious Prime. Fabs from positive phage were reformatted into IgG1 expression plasmids and used for transient expression in Expi293 cells (Thermo Fisher Scientific). Human IgG1 antibodies were purified using protein A affinity chromatography. All of the plasmids are available on request. A list of the sequences of the anti-ACE2 antibodies is provided in Supplementary Table 1.

## Assessment of human antibody binding specificity by ELISA

MaxiSorp 96-well flat-bottomed plates were coated with 50 µl of 125 nM recombinant human or mouse ACE2 protein in PBS at room temperature for 1 h. All washes were performed three times using PBS and 0.1% Tween-20 and all incubations were performed for 1 h at room temperature. Coated plates were washed and blocked by incubation with 4% skim milk solution. Plates were washed and then incubated with 50 µl of 125 nM of anti-ACE2 monoclonal antibodies. The plates were washed and incubated with horseradish peroxidase (HRP)-conjugated goat anti-human IgG secondary antibodies (ab6858, Abcam, 1:5,000). After a final wash, 50 µl of azino-bis-3-ethylbenthiazoline-6-sulfonic acid (ABTS liquid substrate; Sigma-Aldrich) was added and incubated in the dark at room temperature for 20 min and 50 µl of 1% SDS was used to stop the reaction. Absorbance was read at 405 nm and all samples were done in duplicate.

## Affinity measurements using bio-layer interferometry

Affinity determination measurements were performed on the Octet RED96e (FortéBio) system. Assays were performed at 25 °C in solid black 96-well plates agitated at 1,000 rpm. Kinetic buffer was composed of PBS pH 7.4 supplemented with 0.1% (w/v) BSA and 0.05% (v/v) Tween-20. All assays were performed using anti-human IgG Fc capture sensor tip (AHC) sensors (FortéBio). A 60 s biosensor baseline step was applied before anti-ACE2 monoclonal antibodies (5 mg ml$^{-1}$) were loaded onto AHC sensors. For affinity measurements against human ACE2, antibodies were loaded by submerging sensor tips for 200 s and then washed in kinetic buffer for 60 s. Association measurements were performed by dipping into a twofold dilution series of human ACE2 from 6–200 nM for 180 s and dissociation was measured in kinetic buffer for 180 s. Sensor tips were regenerated using a cycle of 5 seconds in 10 mM glycine pH 1.5 and 5 s in kinetic buffer repeated five times. Baseline drift was corrected by subtracting the average shift of an antibody loaded sensor not incubated with protein and an unloaded sensor incubated with protein. Curve fitting analysis was performed with Octet Data Analysis 10.0 software using a global fit 1:1 model to determine $K_D$ values and kinetic parameters. Curves that could not be fitted were excluded from the analyses. The mean kinetic constants and s.e.m. reported are the result of three independent experiments.

## ACE2 and spike blockade

For ACE2 blockade, STs were treated with 20 µg ml$^{-1}$ of one or both of the WCSL141 and WCSL148 antibodies or 40 µg ml$^{-1}$ of human IgG1 isotype control for 1 h before SARS-CoV-2 infection as above. The anti-spike antibodies REGN10987, REGN10933[87] and CR3022[88] were produced using a previously described method[89]. For spike blockade, $10^5$ TCID$_{50}$ (as determined by titration in Vero E6-TMPRSS2 cells) of SARS-CoV-2 virus was incubated with 20 µg ml$^{-1}$ of one or both of REGN10987 and REGN10933, 20 µg ml$^{-1}$ of CR3022 or 40 µg ml$^{-1}$ of human IgG1 isotype control for 1 h before infection of STs as above. After virus removal, cells were cultured in medium containing 20 µg ml$^{-1}$ of each ACE2-blocking antibody, 20 µg ml$^{-1}$ of each spike-blocking antibody or 40 µg ml$^{-1}$ of isotype control until the end of the experiment. The medium was changed daily and infectious virus titres and genome copies on day 3 after infection were determined as described above.

## Antiviral drug treatment

STs were treated with 1 µM of remdesivir (HY-104077, MedChem Express), 1 µM of β-D-$N^4$-hydroxycytidine (NHC, HY-125033, MedChem Express) or an equivalent volume of DMSO for 3 h before SARS-CoV-2 (icSARS-CoV-2-nLuc reporter virus) infection as described above. After virus removal, cells were cultured in medium containing the drug until the end of the experiment. Virus genome copies in cell supernatant were determined as above. Reporter-virus-expressed luciferase levels in the cell lysate were assessed using the Nano-Glo Luciferase Assay Kit according to the manufacturer's instructions (Promega). Drug toxicity in uninfected cells was measured using the Cell-Titer Glo 2.0 assay according to the manufacturer's instructions (Promega). The luciferase and cytotoxicity assays were read using the FLUOstar Omega (BMG Labtech) and reported as relative luminescence units.

## HCG, LDH and cell death detection

Supernatants collected on day 3 after infection (or indicated otherwise) were analysed for HCG and LDH levels. HCG was measured using the Abnova HCG ELISA Kit (KA4005) according to the manufacturer's instructions. All of the supernatants were diluted between 1/1,000 to 1/2,000 before analysis. LDH was measured using the Abcam LDH cytotoxicity kit II (ab65393) according to the manufacturer's instructions in undiluted supernatants using an LDH standard curve. Cell viability was measured using the Promega Cell-Titer Glo 2.0 assay kit. Caspase 3/7 activity was measured using the Caspase Glo 3/7 Assay System (Promega) according to the manufacturer's instructions.

## Immunofluorescence

Cultured cells were fixed in 4% PFA (Sigma-Aldrich) in PBS for 10 min and then permeabilized in PBS containing 0.3% Triton X-100 (Sigma-Aldrich). The cultures were then incubated with primary antibodies followed by secondary antibodies (see the dilutions below). 4′,6-diamidino-2-phenylindole (DAPI) (1:1,000) (Thermo Fisher Scientific) was added to visualize cell nuclei. Images were taken using the DMi8 inverted microscope (Leica). The primary antibodies used in the study were as follows: anti-HCG (ab9582, Abcam, 1:200), anti-dsRNA (MABE1134, Merck, 1:200), anti-ACE2 (ab15348, Abcam, 1:200), anti-GATA2 (WH0002624M1, Sigma-Aldrich, 1:100), anti-GATA3 (MA1-028, Invitrogen, 1:100), anti-SDC1 (12922, Cell Signaling Technology, 1:100), anti-DAB2 (ab76253, Abcam, 1:100), anti-MMP2 (40994, Cell Signaling Technology, 1:100), anti-SARS-CoV-2 nucleocapsid (MBS154642, MyBioSource, 1:300) and anti-HLA-G (ab7759, Abcam, 1:50). Secondary antibodies used in the study (all 1:400) were Alexa Fluor 488 goat anti-mouse IgG1 (A21121, Thermo Fisher Scientific), Alexa Fluor 555 goat anti-mouse IgG (A31570, Thermo Fisher Scientific), Alexa Fluor 555 goat anti-rabbit IgG (A21428, Thermo Fisher Scientific), Alexa Fluor 555 goat anti-mouse IgG2a (A21137, Thermo Fisher Scientific), Alexa Fluor 647 donkey anti-rabbit IgG (A31573, Thermo Fisher Scientific).

## Western blotting

Cell lysates were electrophoresed through a 10% SDS–PAGE gel before transferring to a PVDF membrane. After blocking for 30 min at 4 °C in the blocking buffer (LI-COR), the membrane was incubated

overnight with anti-ACE2 and anti-GAPDH (MAB374, Merck, 1:5,000). The membrane was washed and incubated for 30 min at room temperature with a goat anti-mouse (926-68020, LI-COR, 1:50,000) and goat anti-rabbit (925-32211, LI-COR, 1:50,000) IRDye secondary antibody. After further washing, the membrane was detected with blot membranes and was scanned in the Odyssey Infrared Imaging System (LI-COR).

## Image analysis and cell morphology and quantification

Immunostained cells were imaged using the DMi8 inverted live-cell microscope (Leica). All images in this study were acquired using Motic Image Plus, Leica application suite X and image analysis was performed using ImageJ. Images were taken at ×4, ×10 or ×20 magnification depending on the type of analysis performed. Cell quantification was performed using the particle analysis option of the ImageJ (https://imagej.net/ij/index.html). Four fields of view taken at ×10 magnification were scored first for DAPI-positive nuclei, followed by quantification of HCG- and dsRNA-positive cell bodies. Cells were quantified on the basis of the morphology of foot processes/microvilli and the level of HCG of STs to determine whether the cells were differentiated and undifferentiated. For undifferentiated cells, a lack of foot processes/microvilli and low expression of cytoplasmic HCG were the typical criteria for counting. For differentiated cells, defined foot processed/microvilli and high expression of HCG were considered as criteria for counting. Next, cells were deemed to be dsRNA⁺ or dsRNA⁻ on the basis of the presence of dsRNA-positive staining within the cell (Extended Data Fig. 2d). Finally, these cells were counted and attributed to four different categories: undifferentiated dsRNA⁻, undifferentiated dsRNA⁺, differentiated dsRNA⁻ and differentiated dsRNA⁺. For the quantification of cell fusion, cells were first deemed dsRNA positive or negative as described above and assessed for fusion index, which was calculated by using the number of nuclei counted in the syncytia minus the number of syncytia, then divided by the total number of nuclei counted. Microscopy images were processed using Adobe Photoshop for merging separate colour images.

## RNA extraction and qPCR

RNA was extracted from cells using the RNeasy micro kit (Qiagen) and QIAcube (Qiagen); or the miRNeasy micro kit (Qiagen) according to the manufacturer's instructions. Reverse transcription was then performed using the QuantiTect reverse transcription kit (Qiagen). qPCR reactions were set up in triplicates using the QuantiFast SYBR Green PCR Kit (Qiagen) and then carried out on the 7500 Real-Time PCR system (Thermo Fisher Scientific) using the LightCycler 480 software. The primers used were as follows: *ACE2* F, CAGAGCAACGGTGCACCACGG; *ACE2* R, CCAGAGCCTCTCATTGTAGTCT; *TMPRSS2* F, GTCCCCACTGTCTACGAGGT; *TMPRSS2* R, CAGACGACGGGGGTTGGAAG; *GAPDH* F, CTGGGCTACACTGAGCACC; *GAPDH* R, AAGTGGTCGTTGAGGGCAATG.

## Gene expression analyses

**Pre-processing RNA-seq.** Raw next-generation RNA-seq reads were obtained in the FASTQ format and, before demultiplexing the forward read, the FASTQ reads were trimmed using trimmomatic to 18 nucleotides (the targeted read length as described above) using the following parameters: SE -phred33 CROP:18 MINLEN:18 (ref. 90). FASTQ files were then demultiplexed using sabre[91] with the parameters pe -c -u -m 1 -l10 -n for the barcode indexes as stated above. Next, demultiplexed sample reads were filter-trimmed using trimmomatic to the targeted read length of 101 nucleotides with the parameters SE -phred33 CROP:101 MINLEN:10 (ref. 90). Sequencing reads were then mapped to a customized genome, composed of both GENCODE's GRCh38.p13 and human SARS-CoV-2 (RefSeq: NC_045512.2; see the 'Custom genome for mapping' section below for further details), using STAR (v.2.5.2b)[92] and the following parameters: --outSAMattributes All --alignIntronMax 1000000 --alignEndsType Local. Aligned BAM files were then sorted

and indexed with sambamba[93] using the default parameters, followed by deduplication by unique molecular identifiers using Je's (v.1.2) je markdupes function, with the following parameters: MM = 0 REMOVE_DUPLICATES=true ASSUME_SORTED=true[94]. Read counts were then generated with Subread's (v.1.5.2) featureCount function[95], using the default parameters.

**Gene expression analyses of the human genome.** For each set of analyses (STs infected with virus, STs infected with virus and treated with anti-ACE2), genes mapped to the hSARS-CoV-2 were first removed, and genes with low counts were then filtered out. Specifically, genes with less than five raw read counts across all of the samples were removed, and genes with at least one count per million (CPM) in a minimum of two samples were retained. Before library size normalization, normalization factors were calculated using EdgeR's (v.3.32.1) calcNormFactors function[96,97]. For DGE analysis, normalization and transformation were performed using Limma's (v.3.46.0) voom function[98,99]. Differential gene testing was performed with Limma's lmFit, makeContrasts, contrasts.fit and eBayes functions. For visualization purposes, these data were log₂[CPM]-transformed using EdgeR's cpm function and the following parameters: prior = 1, log = TRUE, normalized.lib.sizes = TRUE. CoA is a dimension reduction technique that can, similar to Principal component analysis (PCA), display a low-dimensional projection of data. However, one of the key differences between CoA and PCA is that, with CoA, two variables of the data may be analysed and visualized to observe the relationship between them (for example, samples and genes)[100,101]. Correspondence analyses were performed using MADE4 (v.1.64.0)[102]. For all heat map visualizations and, where required, sample standardization was performed by normalization to the mean expression of each gene. *k*-means clustering was performed with R's (v.4.0.2) base function kmeans with parameters: centers = 6, nstart = 25. *k*-means clustering was performed on the standardized log₂[CPM] data (which was averaged between replicates before standardization). Hierarchical clustering was performed using the base R package stats (functions: dist and hclust), with the distance measure canberra and linkage method Ward.D. A set seed of 123 was used. Dendrogram visualization was performed using dendexted v.1.15.1 (parameter: k = 3)[103]; 3D visualizations were performed using plotly (v.4.9.4.1)[104]; heat map visualizations were performed using ComplexHeatmap (v.2.6.2)[105]; all other visualizations were performed using ggplot2 (v.3.3.5)[106] and, where required, ggrepel (v.0.9.1)[107]. Gene Ontology and pathway analyses were performed using Metascape (http://metascape.org)[108].

**Gene expression analysis of the human SARS-CoV-2 genome.** To quantify the amount of expression of hSARS-CoV-2 across all of the samples, the raw counts data were used, which included genes from both the human and hSARS-CoV-2 genes. The raw counts data were processed and visualized using the same procedures as described above in the 'Gene expression analyses of the human genome' section. Specifically, data were filtered, normalization factors were calculated, log₂[CPM] counts and CPM (parameter: log = FALSE) counts were generated, and standardized expressions were calculated. For visualization purposes, the expression of hSARS-CoV-2 genes across the respective genome was ordered by the genomic feature's starting base pair position.

## Custom genome for mapping

As the libraries were generated with p(A) enrichment, to avoid multi-mapping of other genes with *ACE2*, we generated a custom GENCODE GRCh38.p13 genomic reference file, in which we removed the gene *BMX*. Moreover, we generated a custom hSARS-CoV-2 (NC_045512v2) genomic reference file based on the SwissProt precursor sequence (before cleavage) and UniProt protein product (after cleavage) annotations. A custom genome combining these human and hSARS-CoV-2 genomes was generated. The protein products for annotation included:

nsp1, nsp2, nsp3, nsp4, 3CL-PRO, nsp6, nsp7, nsp8, nsp9, nsp10, Pol, Hel, ExoN, nsp15, nsp16, spike protein S1, spike protein S2, ORF3a, E, M, ORF6. ORF7a, ORF7b, ORF8, N and ORF10.

## Single-cell analysis and integration

Publicly available droplet-based scRNA data from first trimester placentas were obtained from a previous study[60] (GSM5315569, GSM5315570, GSM5315571, GSM5315572, GSM5315573, GSM5315574 and GSM5315575) for the characterization of the placental cell subtypes. Data were preprocessed and analysed as described previously[60] using Seurat (v.4.2.1)[109–112]. Placental cells were selected and further reclustered using Monocle3 (v.1.3.1)[113–116] with the function cluster_cells and parameter k = 4. Clusters were annotated using marker genes for CTs, STp cells, EVT progenitors and EVTs as specified previously[60]. Pseudotime analysis was performed using Monocle3[113–116] v.1.3.1 (with SeuratWrapper) using the default parameters except with the function learn_graph, which had close_loop = T. The scRNA data were pseudobulked by each placental cell type and placenta patient ID using the AggregateExpression function in Seurat[109–112] v.4.2.1. Only early first trimester (week 6) samples, and only cell types that were identified along the CT to STp trajectory (CT1, CT2, STp) were considered for the integration analysis (Extended Data Fig. 4).

In vivo placenta pseudobulked data were integrated with the RNA-seq data generated in this study using the left_join function in dplyr (https://dplyr.tidyverse.org, https://github.com/tidyverse/dplyr; v.1.0.10)[117]. Low-abundance genes were filtered out using the filterByExpr function in edgeR[96] (v.3.40.0) using the default parameters. Compositional differences between samples were then normalized with the trimmed mean of $M$ (TMM) value method using the calcNormFactors in edgeR[96,97] (v.3.40.0). The data were then $\log_2$[CPM]-transformed using EdgeR's cpm function and the following parameters: prior = 2, log = TRUE, normalized.lib.sizes = TRUE. Batch correction was performed using the ComBat function in the sva package[118] v.3.46 using the following parameters: sva::ComBat(dat=logCPM, batch=batch, mod=NULL). CoA and hierarchical clustering analysis were then performed on the batch-corrected data, similarly to as described in the 'Gene expression analysis of the human SARS-CoV-2 genome' section, with the exception of using Euclidean distances and Ward.D2 for hierarchical clustering.

## Statistics and reproducibility

No statistical method was used to predetermine the sample size for our experiments, but our sample sizes are similar to those reported in previous publications in the field. No data were excluded from the analyses. Data distribution was assumed to be normal, but this was not formally tested. The experiments and samples were not randomized. The investigators were not blinded to allocation during experiments and outcome assessment. The data were statistically analysed as described in the figure legends; specific statistical tests applied are indicated in the respective figure legends. Statistical analyses were performed using GraphPad Prism. The number of experiments and replicates is described in the figure legends.

## Reporting summary

Further information on research design is available in the Nature Portfolio Reporting Summary linked to this article.

## Data availability

The data are available at the Gene Expression Omnibus (GEO) repository (GSE185471). Placenta single-cell data have been previously published[60] that were reanalysed here are available at the GEO (GSM5315569, GSM5315570, GSM5315571, GSM5315572, GSM5315573, GSM5315574 and GSM5315575). All other data supporting the findings of this study are available from the corresponding author on reasonable request. Source data are provided with this paper.

## Code availability

The code is available at GitHub (https://github.com/pololab/COVID_and_Placenta).

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

## Acknowledgements

We thank the staff at the ACRF Centre for Cancer Genomic Medicine at the MHTP Medical Genomics Facility for assistance with next-generation library preparation and Illumina sequencing. The schematics (Figs. 1a, 2a, 3b, 4a and Extended Data Fig 3i) were created using BioRender. J.M.P. and K.S. were supported by an MRFF grant (MRF9200007) and a DHHS Victorian Government Grant; J.M.P. by an ARC Future Fellowship and a NHMRC Ideas Grant (APP2004774); and K.S. by an NHMRC Investigator grant (APP1177174). The Melbourne WHO Collaborating Centre for Reference and Research on Influenza is supported by the Australian Government Department of Health. W.H.T. is a Howard Hughes Medical Institute–Wellcome Trust International Research Scholar (208693/Z/17/Z). Anti-ACE2 antibody generation was supported by the Victorian Government and the Medical Research Future Fund (MRFF) GNT2002073. The South Australian immunoGENomics Cancer Institute (SAiGENCI) received grant funding from the Australian Government.

## Author contributions

J.M.P. and K.S. conceptualized and supervised the study. J.C. and J.P.T. performed all the cellular work with the support of Y.B.Y.S., E.W., G.S., X.L. and Y.Z.; J.A.N. and R.R. performed all of the viral work. J.C., J.P.T., J.A.N. and R.R. performed all of the molecular and microscopy analysis in the infected cells. M.M. performed the bioinformatics analysis under the supervision of N.G.B., F.J.R. and J.M.P.; Y.L. performed the placenta staining under the supervision of G.N.; Y.W. and G.N. provided the FT008 TSC line; D.D., P.P. and W.H.T. generated, characterized and provided the anti-ACE2 antibodies. J.M.P., K.S., J.C., J.P.T., J.A.N. and R.R. wrote the manuscript with contributions from all of the authors.

## Competing interests

J.M.P. and X.L. are listed as inventors on a patent related to the generation of iTSCs filed by Monash University. The other authors declare no competing interests.

## Additional information

**Extended data** is available for this paper at https://doi.org/10.1038/s41556-023-01182-0.

**Correspondence and requests for materials** should be addressed to K. Subbarao or J. M. Polo.

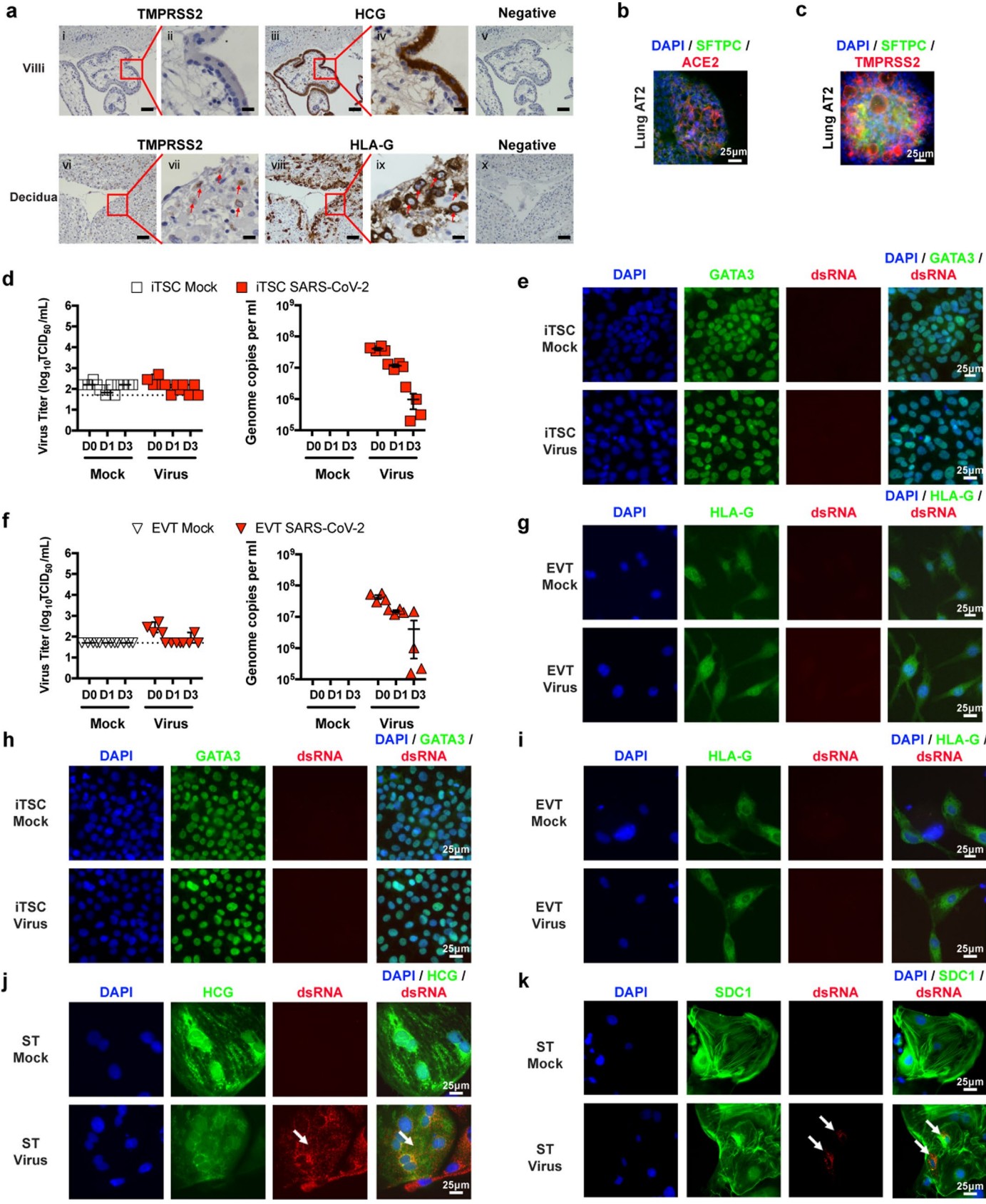

**Extended Data Fig. 1 | See next page for caption.**

**Extended Data Fig. 1 | Only STs are infected with Ancestral SARS-CoV-2.**
**a**, Immunohistochemistry images of first trimester placental villi and decidua for TMPRSS2, HCG, and HLA-G. Scale bar = 1000 μm (i, iii, v, vi, viii, x), 200 μm (ii, iv, vii, ix). **b**, Immunofluorescence images for Surfactant protein C (SFTPC; Green) and ACE2 (Red) of type 2 alveolar epithelial cell positive control. **c**, Immunofluorescence images for SFTPC (Lung AT2; Green), and TMPRSS2 (Red). **d**, Virus titre expressed in $\log_{10}$ TCID$_{50}$/ml and genome copies/ml in 55F iTSCs. **e**, Immunofluorescence images for dsRNA (Red) and GATA3 (Green) in 55F iTSCs. **f**, Virus titre expressed in $\log_{10}$ TCID$_{50}$/ml and genome copies/ml in 55F EVTs.

**g**, Immunofluorescence images for dsRNA (Red) and HLA-G (Green) in 55F EVTs. **h**, Immunofluorescence images for dsRNA (Red) and GATA3 (Green) in 32F iTSCs. **i**, Immunofluorescence images for dsRNA (Red) and HLA-G (EVTs; Green) in 32F EVTs. **j**, Immunofluorescence images for dsRNA (Red) and HCG (STs; Green) in 32F STs. **k**, Immunofluorescence images for dsRNA (Red) and Syndecan-1 (SDC1; Green) in 55F STs. White arrows indicate dsRNA. Cells counterstained with DAPI. **d,f:** Representative graphs from 2 independent experiments showing n = 4 biological replicates. Data are presented with ± SEM however, no statistical tests were performed. Source numerical data are provided.

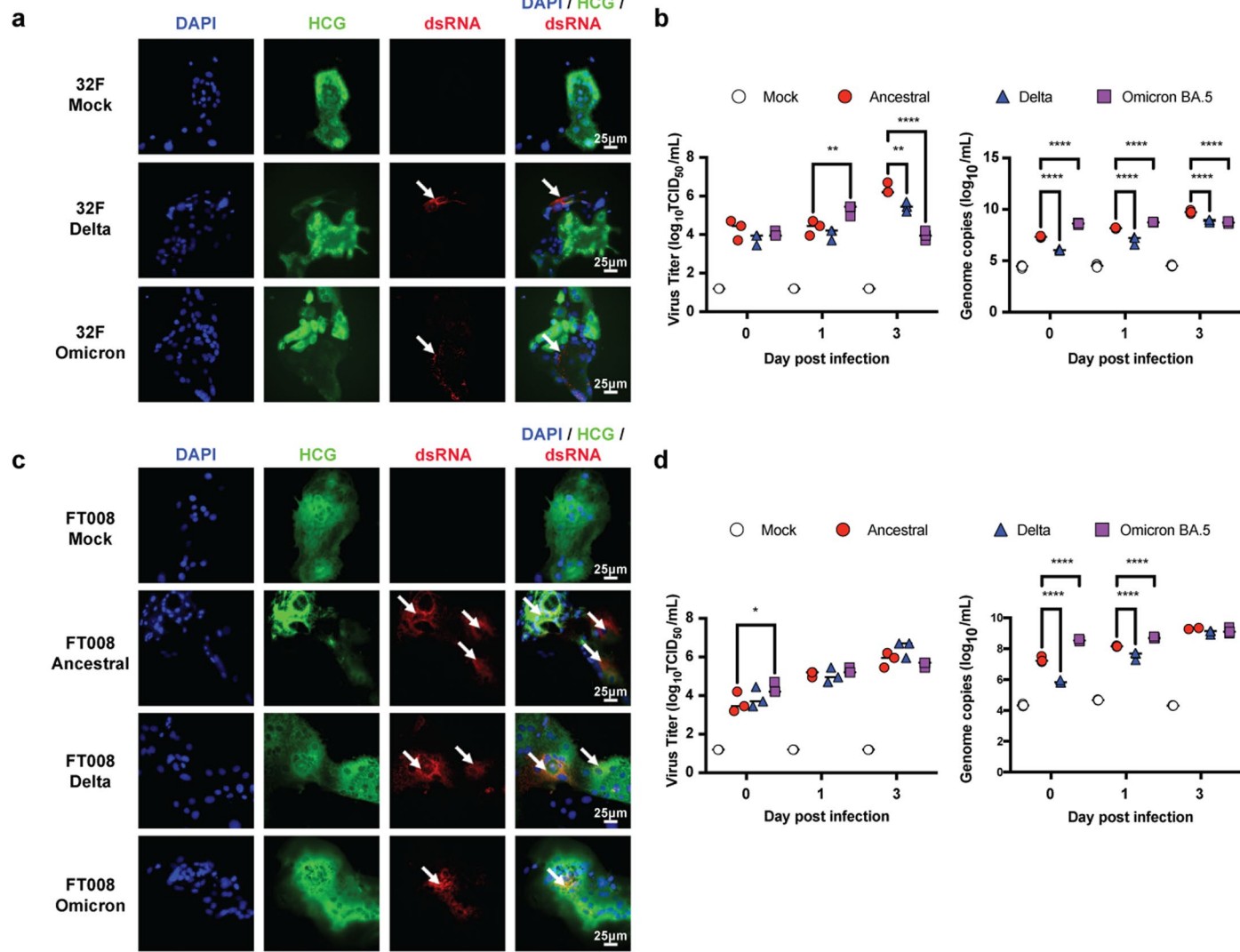

**Extended Data Fig. 2 | Infection of SARS-CoV-2 variants in 32 F and FT008 STs. a**, Immunofluorescence images for dsRNA (Red) and HCG (Green) in 32F STs (Ancestral infection in Extended Data Fig. 1j). **b**, Virus titre expressed in $\log_{10}$TCID$_{50}$/ml and genome copies in $\log_{10}$/mL in 32F STs. **c**, Immunofluorescence images for dsRNA (Red) and HCG (Green) in FT008 STs. **d**, Virus titre expressed in $\log_{10}$TCID$_{50}$/ml and genome copies in $\log_{10}$/mL in FT008 STs. White arrows indicate dsRNA. Cells counterstained with DAPI. **b,d:** Representative graphs from 2 independent experiments showing n = 3 biological replicates. **b,d:** Two-way ANOVA analysis was used to compare infected conditions with mock control. *p < 0.05, **p < 0.01, ****p < 0.0001. Data are presented with ± SEM. Source numerical data are provided.

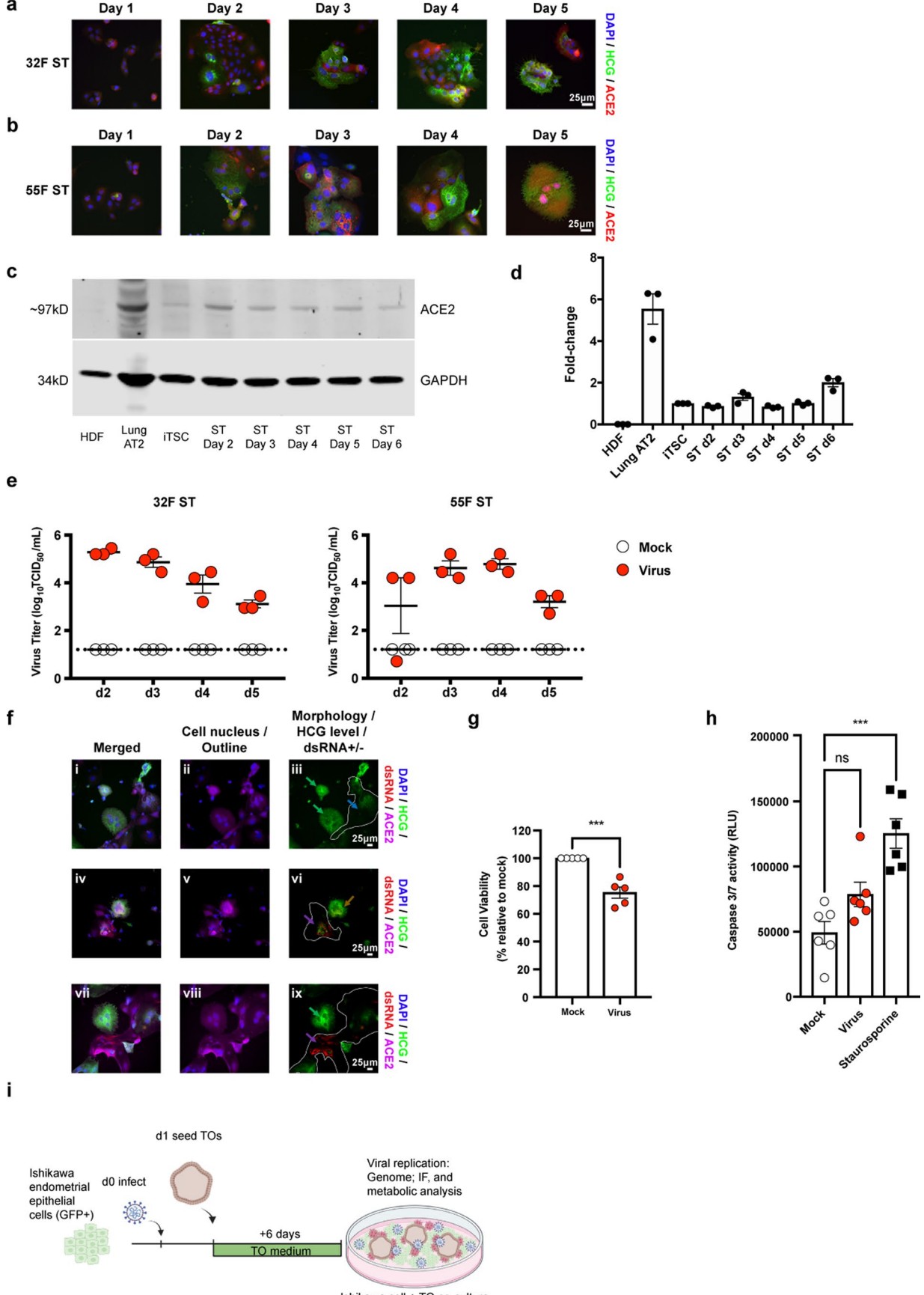

**Extended Data Fig. 3 | See next page for caption.**

**Extended Data Fig. 3 | Infection of SARS-CoV-2 during differentiation of STs.**
Immunofluorescence images for ACE2 (Red) along with HCG (STs; Green) in d1 to d5 STs in 32F (**a**) and 55F (**b**) cell lines. **c**, Western blot analysis for ACE2 (~97kD) and GAPDH (~34kD) in HDF (Fibroblast), Lung AT2, iTSC, and differentiating STs. **d**, qPCR analysis for *TMPRSS2* in HDF (Fibroblast), Lung AT2, iTSC, and differentiating STs (fold-change relative to iTSCs). **e**, Virus titre expressed in $\log_{10} TCID_{50}$/ml of 32F and 55F STs on d2 to d5. **f**, Image illustrating how the scoring of morphology of STs was performed. i, iv, vii: Merged images of DAPI (Blue), HCG (Green), dsRNA (Red), ACE2 (Magenta). ii, v, viii: Cell nucleus (DAPI) and outline of cells (ACE2) to mark multinucleated STs. iii: Merged HCG and dsRNA images for differentiated dsRNA- (green arrows) and undifferentiated dsRNA- (blue arrow, white outlined cell) cells. vi: Merged HCG and dsRNA images for differentiated dsRNA+ (brown arrow) and undifferentiated dsRNA+ (purple arrow, white

outlined cell) cells. ix: Merged HCG and dsRNA images for differentiated dsRNA-(green arrow) and undifferentiated dsRNA+ (purple arrow, white outlined cell) cells. **g**, Cell viability assay (CTG) at day 3 post infection in mock and virus cultures in 55F STs. **h**, Caspase 3/7 activity at day 3 post infection in mock, virus, and staurosporine (control) treated cultures in 55F STs. **i**, Schematic of co-culture experiment with GFP-positive Ishikawa cells and TOs cultured with TO medium. **d:** n = 3 independent experiments. **e:** Representative graphs from 2 independent experiments showing n = 3 biological replicates. **g:** n = 5 experimental replicates. **h:** n = 6 experimental replicates. **g:** Independent T-test (unpaired, two-tailed) was used to compare only between Virus against Mock control. **h:** One-way ANOVA analysis was used to compare Virus, Staurosporine, and Mock control. ***p < 0.001. Data are presented with ± SEM, however no statistical tests were performed for **d** and **e**. Source numerical data and raw blots are provided.

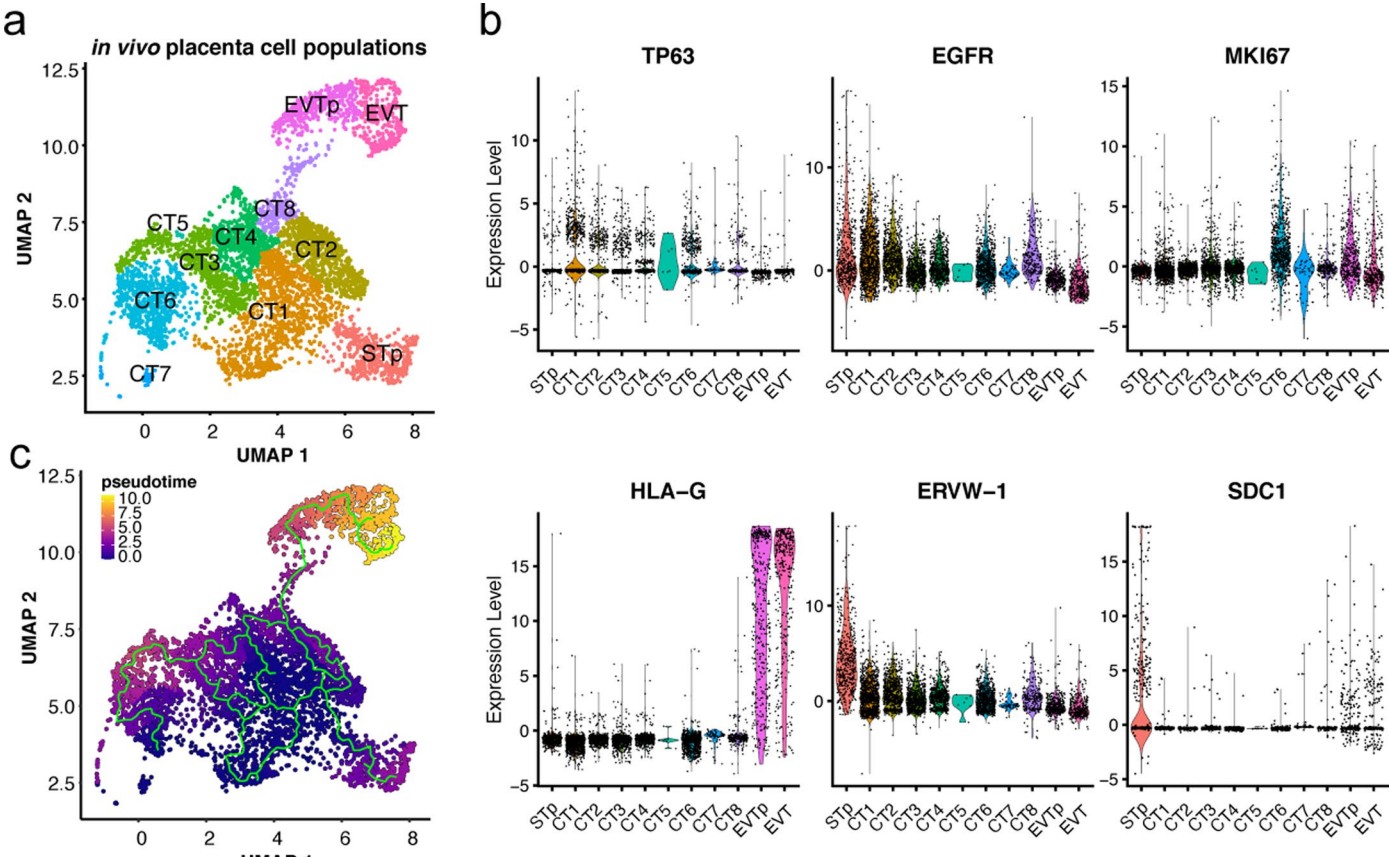

**Extended Data Fig. 4 | Single cell transcriptomic characterization of *in vivo* placental cell subpopulations. a**, *Uniform manifold approximation and projection (UMAP) projection of 5391 cells clustered by gene expression, and coloured cluster.* **b**, Expression of several placental markers across cell populations: *TP63* (likely starting CT population), *EGFR* (ST progenitor, CT), *MKI67* (differentiating CTs), *HLA-G* (EVT), *ERVW-1* and *SDC1* (ST progenitor). **c**, pseudotime analysis of *in vivo* placental populations. Data was analysed from[60].

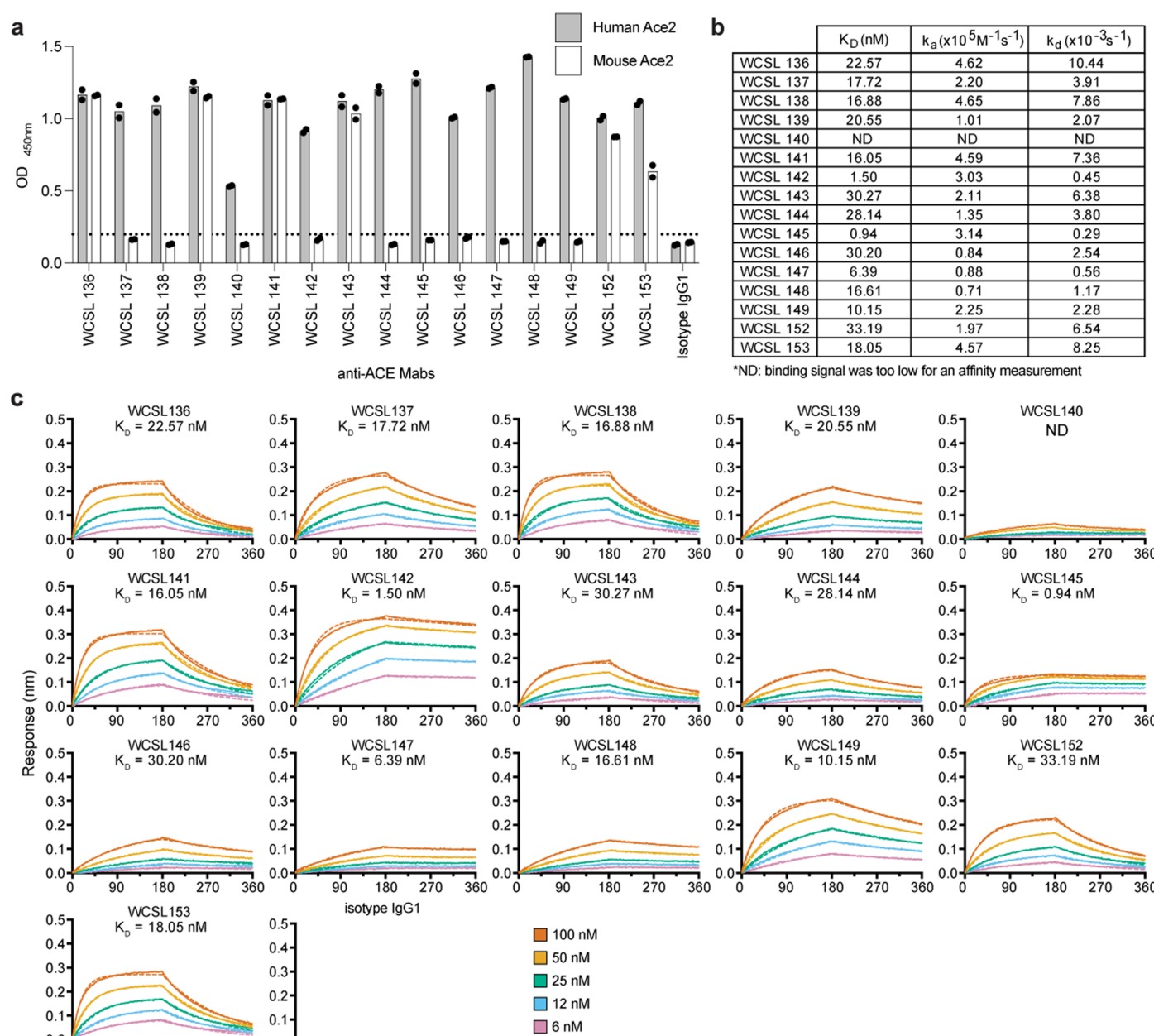

**Extended Data Fig. 5 | Recognition specificity of recombinant ACE2 and binding kinetics of anti-ACE2 monoclonal antibodies. a**, ELISA $OD_{450\,nm}$ signal of anti-ACE2 mAbs binding to human and mouse ACE2 (black and white bars respectively). Error bars represent mean ± standard deviation of technical duplicates. **b**, Measured kinetic rate constants and affinity data for human ACE2 binding to immobilized anti-ACE2 mAbs. **c**, Representative binding curves of five different human ACE2 concentrations from 6–100 nM binding to immobilized anti-ACE2 mAb. Responses measured from the experiment are represented by the solid line, and curves globally fit with a 1:1 binding model are represented by the dotted line. The isotype IgG1 control does not recognise ACE2.

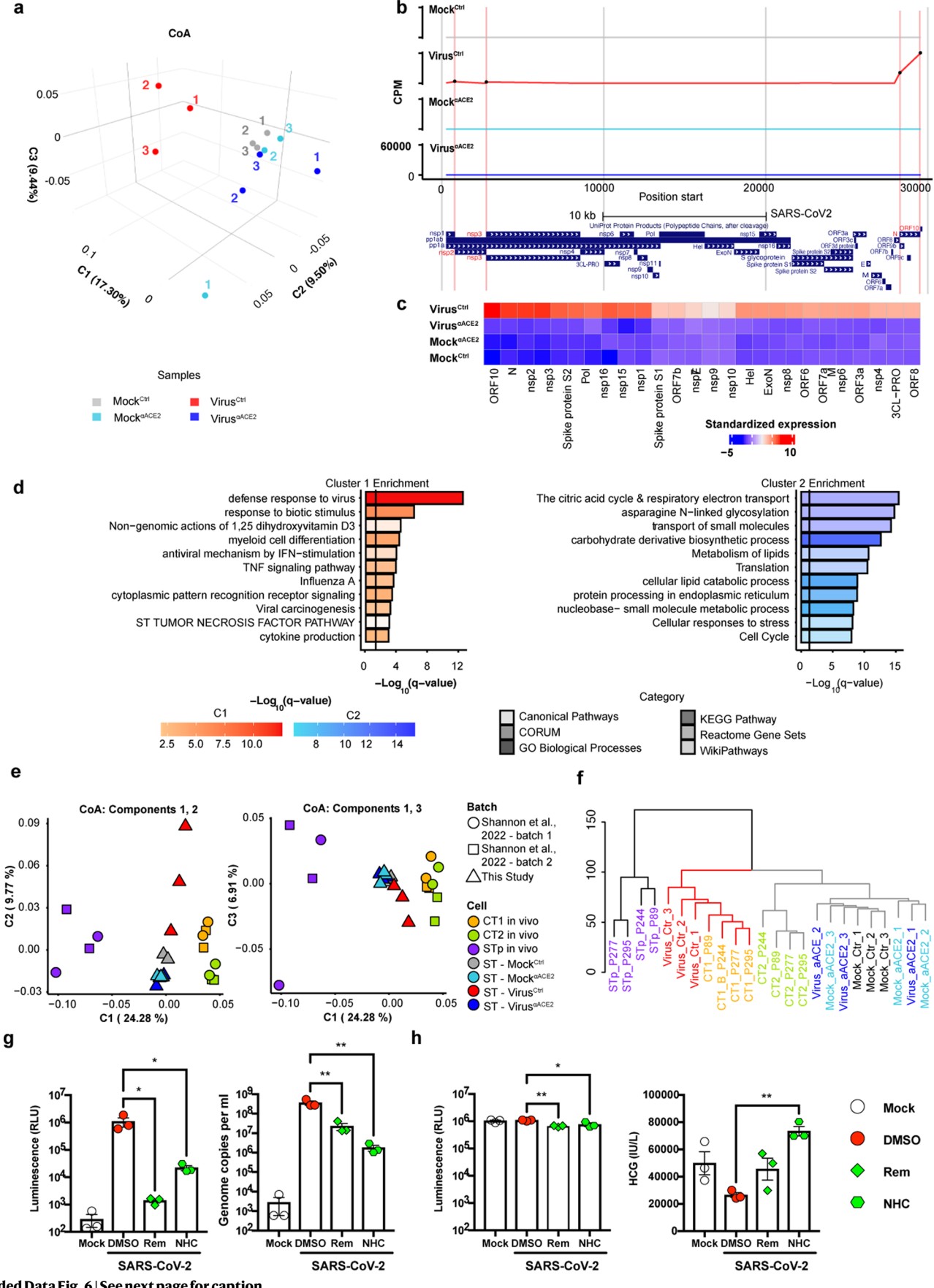

**Extended Data Fig. 6 | See next page for caption.**

**Extended Data Fig. 6 | Transcriptome differences between STs under virus, mock and treated conditions, as well as SARS-CoV-2 expression in the ST samples. a**, Transcriptome CoA of virus, mock and treated conditions of d3 infected STs. **b**, RNA expression levels of hSARS-CoV-2 genomic elements from the RNA-seq data in d3 STs under virus, mock, and treated conditions. Red vertical lines indicate regions with most transcript abundance (ORF10, N, nsp2, nsp3). **c**, Hierarchically clustered heatmap of expression levels across the SARS-CoV2 genomic elements for d3 STs under virus, mock and treated conditions. **d**, Functional enrichment analysis for genes upregulated (k-means cluster 1) and downregulated (k-means cluster 2) in STs infected with virus. **e**, Principal components 1, 2 and 3 of the CoA of virus, mock, treated conditions of d3 infected STs transcriptomics data, as well as of vivo cytotrophoblast and ST from publicly available data. **f**, Hierarchical clustering of the same samples as in **e**. **g**, Virus titre expressed in 'Luminescence relative light units (RLU)' and genome copy analysis of d5 infected STs ± antiviral drug treatment (Rem = Remdesivir; NHC = Molnupiravir). **h**, Drug toxicity (RLU) and HCG secretion by infected STs ± antiviral drugs. **g**,**h**: Cells infected with icSARS-CoV-2-nLuc reporter virus. **g**,**h**: Representative graphs from 2 independent experiments showing n = 3 biological replicates. **g**,**h**: One-way ANOVA analysis with Dunnett's multiple comparison test was used to compare antiviral drug treated conditions with DMSO control. *p < 0.05, **p < 0.01. Data are presented with ± SEM. Source numerical data are provided.

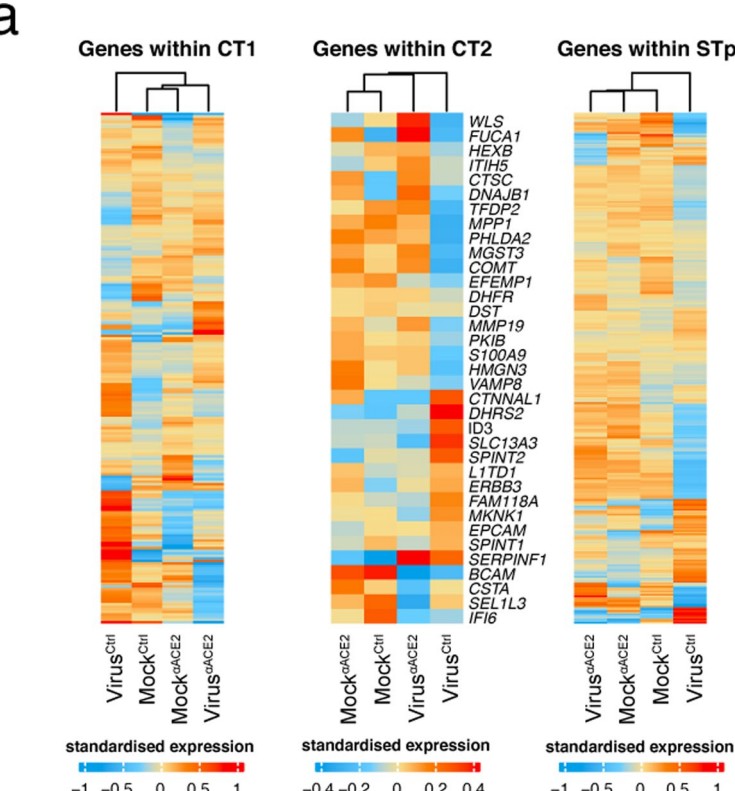

**Extended Data Fig. 7 | Heatmaps of placental gene signatures. a**, Heatmaps of placental gene signatures CT1, CT2 and STp, with hierarchical clustering of the samples based on the selected gene expression profile.

# Reporting Summary

## Statistics

For all statistical analyses, confirm that the following items are present in the figure legend, table legend, main text, or Methods section.

| n/a | Confirmed | |
|---|---|---|
| ☐ | ☒ | The exact sample size (*n*) for each experimental group/condition, given as a discrete number and unit of measurement |
| ☐ | ☒ | A statement on whether measurements were taken from distinct samples or whether the same sample was measured repeatedly |
| ☐ | ☒ | The statistical test(s) used AND whether they are one- or two-sided *Only common tests should be described solely by name; describe more complex techniques in the Methods section.* |
| ☐ | ☒ | A description of all covariates tested |
| ☒ | ☐ | A description of any assumptions or corrections, such as tests of normality and adjustment for multiple comparisons |
| ☐ | ☒ | A full description of the statistical parameters including central tendency (e.g. means) or other basic estimates (e.g. regression coefficient) AND variation (e.g. standard deviation) or associated estimates of uncertainty (e.g. confidence intervals) |
| ☐ | ☒ | For null hypothesis testing, the test statistic (e.g. *F*, *t*, *r*) with confidence intervals, effect sizes, degrees of freedom and *P* value noted *Give P values as exact values whenever suitable.* |
| ☒ | ☐ | For Bayesian analysis, information on the choice of priors and Markov chain Monte Carlo settings |
| ☒ | ☐ | For hierarchical and complex designs, identification of the appropriate level for tests and full reporting of outcomes |
| ☒ | ☐ | Estimates of effect sizes (e.g. Cohen's *d*, Pearson's *r*), indicating how they were calculated |

*Our web collection on statistics for biologists contains articles on many of the points above.*

## Software and code

Policy information about availability of computer code

| Data collection | LightCycler 480 software (Roche, version 1.5.1.62), Leica application suite X version 3.7.1.21655 for DMi8, FLUOstar® Omega (BMG Labtech) |
|---|---|
| Data analysis | GraphPad Prism (v7), ImageJ (v1.8.0_112), Adobe Photoshop (version 21.2.3), base R(3.6.3), STAR aligner (v2.5.2), limma 3.42.2, tidyverse 1.3.0, trimmomatic 0.36, sabre 1.000, Subread v1.5.2, EdgeR v3.32.1, MADE4 v1.64.0, ComplexHeatmap v2.6.2, ggplot2 v3.3.5, ggrepel v0.9.1, Seurat v4.2.1, Monocle3 v1.3.1, dplyr v1.0.10, EdgeR v3.40.0, sva packagae v3.46 |

For manuscripts utilizing custom algorithms or software that are central to the research but not yet described in published literature, software must be made available to editors and reviewers. We strongly encourage code deposition in a community repository (e.g. GitHub). See the Nature Portfolio guidelines for submitting code & software for further information.

## Data

Policy information about availability of data

All manuscripts must include a data availability statement. This statement should provide the following information, where applicable:

- Accession codes, unique identifiers, or web links for publicly available datasets
- A description of any restrictions on data availability
- For clinical datasets or third party data, please ensure that the statement adheres to our policy

Raw and processed next generation sequencing datasets were deposited at the Gene Expression Omnibus (GEO) repository with the following accession number: GSE185471

## Human research participants

Policy information about studies involving human research participants and Sex and Gender in Research.

| | |
|---|---|
| Reporting on sex and gender | This study utilised placental tissues obtained from pregnant women, so participants were females only, and sex and gender analysis does not apply to this study. |
| Population characteristics | Nothing to add here |
| Recruitment | Women undergoing elective termination of pregnancy were recruited, with exclusion criteria of known fetal abnormalities or participants <18 years of age. All participants provided informed written consent on placental tissue donation, and there is no self-selection bias or other biases that may affect the result. |
| Ethics oversight | Ethics approval for the use of first trimester human placental tissues for research was obtained from the Human Ethics Committee at Monash Health, Melbourne, Australia (RES-19-0000-399A). |

Note that full information on the approval of the study protocol must also be provided in the manuscript.

# Field-specific reporting

Please select the one below that is the best fit for your research. If you are not sure, read the appropriate sections before making your selection.

☒ Life sciences          ☐ Behavioural & social sciences          ☐ Ecological, evolutionary & environmental sciences

For a reference copy of the document with all sections, see nature.com/documents/nr-reporting-summary-flat.pdf

# Life sciences study design

All studies must disclose on these points even when the disclosure is negative.

| | |
|---|---|
| Sample size | We did not involve statistical methods to pre-determine the sample size, this was determined based on previous experience (Liu et al., Nature 2020) and other similar studies (Karvas et al., Cell Stem Cell 2022; Ruan et al., Cell Reports Medicine 2022). We generated data using 2 iTSC cell lines in multiple runs of differentiation and infection (Representative graphs from 2 independent experiments showing n=3 separate infection replicates). |
| Data exclusions | No data were excluded. |
| Replication | Each experiment was repeated independently at least twice successfully, with each experiment containing a minimum of 3 separate infection replicates. |
| Randomization | Experiments were not randomized. Randomization of samples were not necessary to our study as we needed to determine virus vs control in most cases and our sample sizes were typically low. |
| Blinding | The investigators were not blinded during data collection and analysis, as neither human/animal studies or specific grouping were involved in this manuscript |

# Reporting for specific materials, systems and methods

We require information from authors about some types of materials, experimental systems and methods used in many studies. Here, indicate whether each material, system or method listed is relevant to your study. If you are not sure if a list item applies to your research, read the appropriate section before selecting a response.

## Materials & experimental systems

| n/a | Involved in the study |
|-----|----------------------|
| ☐ | ☒ Antibodies |
| ☐ | ☒ Eukaryotic cell lines |
| ☒ | ☐ Palaeontology and archaeology |
| ☒ | ☐ Animals and other organisms |
| ☒ | ☐ Clinical data |
| ☒ | ☐ Dual use research of concern |

## Methods

| n/a | Involved in the study |
|-----|----------------------|
| ☒ | ☐ ChIP-seq |
| ☒ | ☐ Flow cytometry |
| ☒ | ☐ MRI-based neuroimaging |

# Antibodies

| | |
|---|---|
| Antibodies used | For Immunostaining:<br>Antibody, Company, Catalogue Number, Lot Number, Dilution:<br>Mouse anti-GATA2 IgG2a clone 2D11, Sigma-Aldrich Cat# WH0002624M1 Lot G2151-2D11, 1:100<br>Negative Control Rabbit Immunoglobulin Fraction, X0903, Dako, 1:10000<br>Mouse IgG1 Culture supernatant, X0931, Agilent Technologies, 1:10000<br>Goat Anti-Rabbit IgG Antibody (H+L), Biotinylated, BA-1000, Vector Laboratories, 1:1000<br>Horse Anti-Mouse IgG Antibody (H+L), Biotinylated, BA-2000, Vector Laboratories, 1:1000<br>Mouse anti-HCG IgG1 clone 5H4-E2, abcam , ab9582 Lot GR3285169-1, 1:200<br>Mouse anti-dsRNA IgG2a clone rJ2, Merck, MABE1134 Lot 3543801, 1:200<br>Rabbit anti-ACE2 IgG, abcam, ab15348 Lot GR3333640-15, 1:200<br>Mouse anti-GATA3 IgG1 clone 1A12-1D9, Invitrogen, MA1-028, 1:100<br>anti-HLA G IgG1 clone MEM-G/1, abcam, ab7759 Lot GR3262011-5, 1:50<br>Rabbit anti-SDC1 IgG, Cell Signaling Technology, Cat# 12922 Lot 1, 1:400<br>Rabbit anti-anti-DAB2 IgG, ab76253, abcam, 1:100<br>Rabbit anti-MMP2 IgG, Cell Signaling Technology, Cat# 40994, 1:100<br>Mouse anti-SARS-CoV-2 Nucleocapsid IgG, MyBioSource, MBS154642, 1:300<br>Goat anti-human IgG H&L (HRP), abcam, ab6858, 1:5000<br>Goat anti-mouse IgG1 Alexa Fluor 488, Invitrogen, Thermo Fisher Scientific, A21121 Lot 1964382, 1:400<br>Goat anti-rabbit IgG Alexa Fluor 555, Invitrogen, Thermo Fisher Scientific, A21428 Lot 1786491, 1:400<br>Goat anti-mouse IgG2a Alexa Fluor 555, Invitrogen, Thermo Fisher Scientific, A21137 Lot 1899521, 1:400<br>Donkey anti-rabbit IgG Alexa Fluor 647, Invitrogen, Thermo Fisher Scientific, A31573 Lot 1903516, 1:400<br>Donkey anti-mouse IgG Alexa Fluor 555, Invitrogen, Thermo Fisher Scientific, A31570, Lot 1850121, 1:400<br><br>For Western Blot:<br>Mouse Anti-GAPDH IgG1 clone 6C5, Merck, MAB374 Lot 3018865,  1:5000<br>Goat anti-Mouse IgG 680LT, LI-COR, 926-68020 Lot C20531-05, 1:50000<br>Goat anti-Rabbit IgG 800CW, LI-COR, 925-32211 Lot C80925-01, 1:50000 |
| Validation | Antibodies obtained from th e commercial source were validated by the suppliers, detailed validation analysis relevant literatures are provided on the company website for the products used in this study. Some antibodies were validated in a previously published study as indicated in methods or relevant literature was cited.<br><br>GATA2 (WH0002624M1) https://www.sigmaaldrich.com/catalog/product/sigma/wh0002624m1<br>Negative Control Rabbit Immunoglobulin Fraction (X0903) https://www.agilent.com/cs/library/packageinsert/public/SSX0903RUO_01.pdf<br>Mouse IgG1 Culture supernatant (X0931) https://www.agilent.com/cs/library/packageinsert/public/102432002.PDF<br>Anti-Rabbit IgG Antibody (H+L), Biotinylated (BA-1000) https://vectorlabs.com/products/antibodies/biotinylated-goat-anti-rabbit-igg<br>Anti-Mouse IgG Antibody (H+L), Biotinylated (BA-2000) https://vectorlabs.com/products/antibodies/biotinylated-horse-anti-mouse-igg<br>HCG (ab9582) https://www.abcam.com/hcg-beta-antibody-5h4-e2-ab9582.html<br>dsRNA (MABE1134) https://www.merckmillipore.com/AU/en/product/Anti-dsRNA-Antibody-clone-rJ2,MM_NF-MABE1134-25UL<br>ACE2 (ab15348) https://www.abcam.com/ace2-antibody-ab15348.html<br>GATA3 (MA1-028) https://www.thermofisher.com/antibody/product/GATA3-Antibody-clone-1A12-1D9-Monoclonal/MA1-028<br>HLA G (ab7759) https://www.abcam.com/hla-g-antibody-mem-g1-ab7759.html<br>SDC1 (Cat# 12922) https://www.cellsignal.com/products/primary-antibodies/syndecan-1-d4y7h-rabbit-mab/12922<br>MMP2 (Cat# 40994) https://www.cellsignal.com/products/primary-antibodies/mmp-2-d4m2n-rabbit-mab/40994<br>SARS-CoV-2 Nucleocapsid (MBS154642) https://www.mybiosource.com/monoclonal-covid-19-antibody/covid-19-nucleocapsid-np-coronavirus/154642<br>Goat anti-Human IgG H&L (HRP) https://www.abcam.com/products/secondary-antibodies/goat-human-igg-hl-hrp-ab6858.html<br>Goat anti-mouse IgG1 Alexa Fluor 488 (A21121) https://www.thermofisher.com/antibody/product/Goat-anti-Mouse-IgG1-Cross-Adsorbed-Secondary-Antibody-Polyclonal/A-21121<br>Goat anti-rabbit IgG Alexa Fluor 555 (A21428) https://www.thermofisher.com/antibody/product/Goat-anti-Rabbit-IgG-H-L-Cross-Adsorbed-Secondary-Antibody-Polyclonal/A-21428<br>Goat anti-mouse IgG2a Alexa Fluor 555 (A21137) https://www.thermofisher.com/antibody/product/Goat-anti-Mouse-IgG2a-Cross-Adsorbed-Secondary-Antibody-Polyclonal/A-21137<br>Donkey anti-rabbit IgG Alexa Fluor 647 (A31573) https://www.thermofisher.com/antibody/product/Donkey-anti-Rabbit-IgG-H-L-Highly-Cross-Adsorbed-Secondary-Antibody-Polyclonal/A-31573<br>Donkey anti-mouse IgG Alexa Fluor 555 (A31570)<br>https://www.thermofisher.com/antibody/product/Donkey-anti-Mouse-IgG-H-L-Highly-Cross-Adsorbed-Secondary-Antibody- |

Polyclonal/A-31570
GAPDH (MAB374) https://www.merckmillipore.com/AU/en/product/Anti-Glyceraldehyde-3-Phosphate-Dehydrogenase-Antibody-clone-6C5,MM_NF-MAB374
Goat anti-mouse IgG 680LT (926-68020) https://www.licor.com/bio/reagents/irdye-680lt-goat-anti-mouse-igg-secondary-antibody
Goat anti-rabbit IgG 800CW (925-32211) https://www.licor.com/bio/reagents/irdye-800cw-goat-anti-rabbit-igg-secondary-antibody

# Eukaryotic cell lines

Policy information about cell lines and Sex and Gender in Research

| | |
|---|---|
| Cell line source(s) | Human iTSC lines were previously generated as described in Liu et al., Nature 2020. (https://doi.org/10.1038/s41586-020-2734-6). iTSCs were generated using human fibroblasts sourced from ThermoFisher (Catalogue number, C-013-5C and lot#1528526 for 55F and lot#1569390 for 32F. Vero cells (Cat no. CCL-81) and Calu-3 cells (Cat no. HTB-55) were purchased from ATCC. Vero E6-TMPRSS2 (Cat no. JCRB1819) were purchased from CellBank Australia. Expi293 cells (Cat no. 14527) were purchased from Thermo Fisher Scientific. Ishikawa cells were sourced from Sigma-Aldrich (Catalogue number, 99040201, lot #14B013). FT008 primary cell lines were derived from the tissue consented by patients. However, as agreed in the initial ethics to receive the cells, the sex of the donor was not disclosed. However, we have analyzed sequencing data and found their genotype to be male. |
| Authentication | The human fibroblasts, Vero cells, Calu-3 cells, Vero E6-TMPRSS2, Expi293 and Ishikawa cells have been authenticated by the manufacturer's company via assays such as cellular morphology, STR profilling, mycoplasma testing, sterility testing or growth profile as stated in the certificate of analysis. The iTSCs and FT008 have been also authenticated in-house by immunostaining, qPCR, RNA-seq. |
| Mycoplasma contamination | Furthermore, all cell lines were regularly tested and were mycoplasma negative. |
| Commonly misidentified lines (See ICLAC register) | No commonly misidentified cell lines were used in this study. |

