## [Peer Review File · Nature Cell Biology]

Peer Review Information

Journal: Nature Cell Biology

Manuscript Title: A placental model of SARS-CoV-2 infection reveals ACE2-dependent susceptibility and differentiation impairment in syncytiotrophoblasts

Corresponding author name(s): Professor Jose Polo

Editorial Notes:

Reviewer Comments & Decisions:

Decision Letter, initial version:

*Please delete the link to your author homepage if you wish to forward this email to co-authors.

Dear Jose,

Your manuscript, "An iTSC-derived placental model of SARS-CoV-2 infection reveals ACE2-dependent susceptibility in syncytiotrophoblasts", has now been seen by 3 referees, who are experts in stem cell-derived placenta models, RNA-seq, SARS-CoV-2 (referee 1); SARS-CoV-2, SARS-CoV-2 infection during human development (referee 2); and SARS-CoV-2, stem cells (referee 3). As you will see from their comments (attached below) they find this work of potential interest, but have raised substantial concerns, which in our view would need to be addressed with considerable revisions before we can

consider publication in Nature Cell Biology.

Nature Cell Biology editors discuss the referee reports in detail within the editorial team, including the chief editor, to identify key referee points that should be addressed with priority, and requests that are overruled as being beyond the scope of the current study. To guide the scope of the revisions, I have listed these points below. We are committed to providing a fair and constructive peer-review process, so please feel free to contact me if you would like to discuss any of the referee comments further.

In particular, it would be essential to:

(A) Strengthen the rigor of your approaches and further validate your findings as indicated by:

Referee 1:

"Fig. 1a shows ACE2 expression in 1st trimester placental villi and decidua, but TMPRSS2 is also required for SARS-CoV-2 infection. Did the authors check expression of TMPRSS2 in these same placental tissues by immunohistochemistry?"

"Alveolar lung epithelial (AT2) cells are used as a positive control for the ACE2 antibody in Extended Data Fig. 1. How do the transcript levels of ACE2 and TMPRSS2 in iTSC, EVT and ST in Fig. 1d compare with AT2 cells? The upregulation of TMPRSS2 in specialized trophoblast cell types is much lower than ACE2. I wonder how significant it is relative to a known receptive cell type".

"Along similar lines, it would be helpful to compare the levels of viral titer following SARS-CoV-2 infection in trophoblast and alveolar cells side-by-side in Fig. 1f".

"The authors use morphological criteria to assess the extent of ST differentiation based on foot-processes/microvilli. I could not find a detailed explanation of this analysis in the Methods. Is it possible to use a more quantitative criterion like fusion index?"

Referee 2:

"In figure 1 the authors show that in the syncytiotrophoblast culture there are dsRNA + cells that also are HCG negative. It is not clear if this is due to infection or if these cells were not HCG+ to be begin with. What is the efficiency of EVT and ST differentiation from iTSC and what is the purity of these cultures at the time of infection. In addition to dsRNA staining, evaluation with additional markers of SARS-CoV-2 infection would add additional markers of infection (e.g. SARS-N/SARS-S etc). Why was 3 DPI used as a time point for infection? What happens longitudinally over time?"

"It is not clear from the methods whether the TCID50 were calculated using a qRT-PCR method or via a plaque assay method. If only completed by qRT-PCR a plaque assay method should be used for validation".

"The authors show that ACE2 expression increases significantly soon after initiation of ST differentiation. While this reviewer can see parts of this increase on IF (due to figure obstruction in extended figure 2), western blot do quantitatively validate this would more clearly show these changes over time".

"The authors show increased levels of LDH during SARS-CoV-2 infection of ST cultures and suggest

that there are increased levels of cell death but do not demonstrate increased levels of cell loss. Deeper evaluation of these cultures including evidence of cell loss and/or death using markers for apoptosis, necroptosis or necrosis should be completed”.

“It is not clear if the described studies were completed with one donor TSC cell line with repeated experiments (for the transcriptome) and are generalizable or unique to one line or two lines as referenced in figure 1? This needs to be validated with several different donor lines to demonstrate this generalizability and findings”.

(B) Experimentally expand on the impact and implications of your findings, as indicated by referee 3:

“The ongoing pandemic raises an imperative question whether SARS-CoV-2 variants may be more likely to infect placental tissues than the original strain (Argueta LB et al. iScience 2022). The susceptibility of iTSC-derived placental cells to the original SARS-CoV-2 strain and its major variants (including Delta and Omicron variants) has not been studied in this study”.

“The reviewer appreciates the authors’ efforts to develop new monoclonal antibodies and feel like that the analysis should include the current major SARS-CoV-2 variants. The authors also need to compare the potency of their antibodies with existing ones such as S2E12, REGN10933, AZD1061, CR3022, and 47D11 (Tortorici et al., Science 2020; Baum A et al., Science 2020; Zost SJ et al., Nature 2020; Yuan M et al., Science 2021; Wang C et al., Nat Commun 2020)”.

“With respect to placental cellular models, physiologically relevant ones should be highly encouraged, which may address the complexity of the placental biology and pathophysiology. Recently, Theunissen and colleagues described such a 3D trophoblast organoid model, derived from naive human pluripotent stem cells and primary TSCs, for studying placental susceptibility to infection of SARS-CoV-2 (Karvas RM et al. Cell Stem Cell 2022). The current 3D model still has many limitations (e.g., an inside-out-villous architecture with an inner syncytial compartment). Nevertheless, the current efforts should continue establishing such organoid type of 3D culture system, possibly including immune cells (as discussed by the authors) and the vasculature, which would faithfully recapitulate the placental architecture (e.g., the blood-placenta barrier). This would provide insights into clinically relevant problems such as stillbirth and reliable platforms for drug discovery”.

(C) All other referee concerns pertaining to strengthening existing data, providing controls, methodological details, clarifications and textual changes and ensuring that all figures are properly displayed (see relevant point 1 by referee 2) should also be addressed.

(D) Finally, please pay close attention to our guidelines on statistical and methodological reporting (listed below) as failure to do so may delay the reconsideration of the revised manuscript. In particular please provide:

- a Supplementary Table including all numerical source data in Excel format, with data for different figures provided as different sheets within a single Excel file. The file should include source data giving rise to graphical representations and statistical descriptions in the paper and for all instances where the figures present representative experiments of multiple independent repeats, the source data of all

repeats should be provided.

We would be happy to consider a revised manuscript that would satisfactorily address these points, unless a similar paper is published elsewhere, or is accepted for publication in Nature Cell Biology in the meantime.

- ensure that it conforms to our format instructions and publication policies (see below and www.nature.com/nature/authors/).
- provide a point-by-point rebuttal to the full referee reports verbatim, as provided at the end of this letter.
- provide the completed Editorial Policy Checklist (found here <https://www.nature.com/authors/policies/Policy.pdf>), and Reporting Summary (found here <https://www.nature.com/authors/policies/ReportingSummary.pdf>). This is essential for reconsideration of the manuscript and these documents will be available to editors and referees in the event of peer review. For more information see <http://www.nature.com/authors/policies/availability.html> or contact me.

Nature Cell Biology is committed to improving transparency in authorship. As part of our efforts in this direction, we are now requesting that all authors identified as 'corresponding author' on published papers create and link their Open Researcher and Contributor Identifier (ORCID) with their account on the Manuscript Tracking System (MTS), prior to acceptance. ORCID helps the scientific community achieve unambiguous attribution of all scholarly contributions. You can create and link your ORCID from the home page of the MTS by clicking on 'Modify my Springer Nature account'. For more information please visit www.springernature.com/orcid.

[Redacted]

We would like to receive a revised submission within six months. We would be happy to consider a revision even after this timeframe, however if the resubmission deadline is missed and the paper is eventually published, the submission date will be the date when the revised manuscript was received.

We hope that you will find our referees' comments, and editorial guidance helpful. Please do not hesitate to contact me if there is anything you would like to discuss.

Best wishes,

Stelios

Stylianos Lefkopoulos, PhD
He/him/his

Associate Editor
Nature Cell Biology
Springer Nature
Heidelberger Platz 3, 14197 Berlin, Germany

E-mail: stylianos.lefkopoulos@springernature.com
Twitter: @s_lefkopoulos

Reviewers' Comments:

Reviewer #1:

Remarks to the Author:

This manuscript from Jose Polo and colleagues examines SARS-CoV-2 infection in placental cell types generated from induced trophoblast stem cells (iTSCs). Clinical reports have linked COVID-19 infection with adverse pregnancy outcomes, but the potential of SARS-CoV-2 to infect early placental cells remains poorly understood. The authors show that the entry factors ACE2 and TMPRSS2 are expressed in two differentiated trophoblast cell types (EVTs and STs), but only STs are productively infected. They also report that infected ST cultures have reduced expression of differentiation genes and that an anti-ACE2 antibody or antiviral drugs can prevent SARS-CoV-2 infection.

In vitro models of placental infection by SARS-CoV-2 are of considerable current interest. However, the authors must clarify several points (please see below) and some of the figures appeared disorganized in the PDF (especially Fig. 4, Extended Data Figs. 1 and 2), which made it difficult to assess the data.

1. Fig. 1a shows ACE2 expression in 1st trimester placental villi and decidua, but TMPRSS2 is also required for SARS-CoV-2 infection. Did the authors check expression of TMPRSS2 in these same placental tissues by immunohistochemistry?
2. Alveolar lung epithelial (AT2) cells are used as a positive control for the ACE2 antibody in Extended Data Fig. 1. How do the transcript levels of ACE2 and TMPRSS2 in iTSC, EVT and ST in Fig. 1d compare with AT2 cells? The upregulation of TMPRSS2 in specialized trophoblast cell types is much lower than ACE2. I wonder how significant it is relative to a known receptive cell type.
3. Along similar lines, it would be helpful to compare the levels of viral titer following SARS-CoV-2 infection in trophoblast and alveolar cells side-by-side in Fig. 1f.
4. The dsRNA staining in Fig. 1k suggests that only a fraction of STs are infected by SARS-CoV-2, which is consistent with a recent report (Karvas et al., *Cell Stem Cell*, 2022). Can the authors determine the percentage of STs that shows dsRNA expression following SARS-CoV-2 infection?
5. The authors use morphological criteria to assess the extent of ST differentiation based on foot-processes/microvilli. I could not find a detailed explanation of this analysis in the Methods. Is it possible to use a more quantitative criterion like fusion index?
6. Transcriptional profiling shows an interesting difference between virus- and mock-infected STs (Fig. 3a), which is reversed by anti-ACE2 antibody treatment (Fig. 4c). Can the authors identify the developmental stage at which ST differentiation is stalled using single cell data from early placental tissues (e.g. Chen et al., *Development*, 2022; Shannon et al., *Development*, 2022)?

7. Extended Data Fig. 4b: RNA expression across SARS-CoV-2 genomic elements does not look very significant in the virus-infected sample except for the elements towards the very edge of the track. Why do those elements (e.g. ORF6/ORF7a) show lower expression compared to nsp2/3 etc. in the heatmap in the next panel (Extended Data Fig. 4c)?

Reviewer #2:

Remarks to the Author:

The manuscript "An iTSC-derived placental model of SARS-CoV-2 infection reveals ACE2-dependent susceptibility in syncytiotrophoblasts" by Chen et al examined induced fibroblast derived trophoblast stem cell cultures during SARS-CoV-2 infection. Several groups have previously shown that COVID-19 is associated with adverse fetal outcomes including miscarriages and placental alterations. The authors identify that the placental-specific cell types express ACE2 and TMPRSS2 but only the syncytiotrophoblasts show evidence of SARS-CoV-2 infection. Follow-on experiments showed that SARS-CoV-2 infected syncytiotrophoblasts have altered transcriptomes marked by diminished hallmarks of syncytiotrophoblast differentiation and function. Despite these findings there are several areas of concern that dampen the enthusiasm for this manuscript and as a consequence this manuscript should be rejected with hope for resubmission.

Major Comments

- 1) Figure 4, and extended figures 1 and 2 are not able to be displayed properly which greatly impacts the ability to accurately interpret the figure findings. This needs to be addressed to properly interpret the data presented in these figures.
- 2) In figure 1 the authors show that in the syncytiotrophoblast culture there are dsRNA + cells that also are HCG negative. It is not clear if this is due to infection or if these cells were not HCG+ to be begin with. What is the efficiency of EVT and ST differentiation from iTSC and what is the purity of these cultures at the time of infection. In addition to dsRNA staining, evaluation with additional markers of SARS-CoV-2 infection would add additional markers of infection (e.g. SARS-N/SARS-S etc). Why was 3 DPI used as a time point for infection? What happens longitudinally over time?
- 3) It is not clear from the methods whether the TCID50 were calculated using a qRT-PCR method or via a plaque assay method. If only completed by qRT-PCR a plaque assay method should be used for validation.
- 4) The authors show that ACE2 expression increases significantly soon after initiation of ST differentiation. While this reviewer can see parts of this increase on IF (due to figure obstruction in extended figure 2), western blot do quantitatively validate this would more clearly show these changes over time.
- 5) It is surprising that the authors see such a prominent impact of SARS-CoV-2 infection of bHCG production into the supernatant given that the uninfected cells are differentiating properly. What percentage of the cells are infected (as the IF images suggest that it is not a high percentage)? How do the authors explain this discrepancy?
- 6) The authors show increased levels of LDH during SARS-CoV-2 infection of ST cultures and suggest that there are increased levels of cell death but do not demonstrate increased levels of cell loss. Deeper evaluation of these cultures including evidence of cell loss and/or death using markers for apoptosis, necroptosis or necrosis should be completed.
- 7) It is not clear if the described studies were completed with one donor TSC cell line with repeated experiments (for the transcriptome) and are generalizable or unique to one line or two lines as referenced in figure 1? This needs to be validated with several different donor lines to demonstrate this generalizability and findings.
- 8) The authors need to extend their findings in better context of the field and prior reports of placental infection in vitro and in vivo. Only minimal discussion is present and it primarily is in the context of a prior report that claims there was no infection of trophoblast organoids.

Minor comments

1) Several spelling errors are noted in the figures and text (e.g. Figure 1g Virus).

Reviewer #3:

Remarks to the Author:

RE: Chen et al. An iTSC-derived placental model of SARS-CoV-2 infection reveals ACE2-dependent susceptibility in syncytiotrophoblasts

Major Comments

It is a relatively complete analysis of SARS-CoV-2 infection of an iTSC-derived placental model in 2D cell culture. It is well-written. My major concern with this manuscript is that the study lacks significant novelty as well as updated mechanistic insights into SARS-CoV-2 variant infection of placental cells.

Concerning the SARS-CoV-2 cell tropism and multiorgan infection, this topic has been extensively studied and reviewed in the past three years since the onset of the pandemic. It is well known that almost any types of cells with the expression of SARS-CoV-2 receptors (such as ACE2 and TMPRSS2) can be infected by the viruses. As far as the placenta is concerned, there were numerous reports using placental cell culture (e.g., Fahmi A et al. *Cell Reports Medicine*, 2021; Argueta LB et al. *iScience* 2022; Karvas RM et al. *Cell Stem Cell* 2022), which confirmed that syncytiotrophoblasts can be efficiently infected by SARS-CoV-2, one major finding in this study.

With respect to placental cellular models, physiologically relevant ones should be highly encouraged, which may address the complexity of the placental biology and pathophysiology. Recently, Theunissen and colleagues described such a 3D trophoblast organoid model, derived from naive human pluripotent stem cells and primary TSCs, for studying placental susceptibility to infection of SARS-CoV-2 (Karvas RM et al. *Cell Stem Cell* 2022). The current 3D model still has many limitations (e.g., an inside-out-villous architecture with an inner syncytial compartment). Nevertheless, the current efforts should continue establishing such organoid type of 3D culture system, possibly including immune cells (as discussed by the authors) and the vasculature, which would faithfully recapitulate the placental architecture (e.g., the blood-placenta barrier). This would provide insights into clinically relevant problems such as stillbirth and reliable platforms for drug discovery.

For example, the first FDA-approved anti-SARS-CoV-2 drug remdesivir (GS-5734), as indicated as a drug candidate that inhibit SARS-CoV-2 infection in 2D cell culture, failed to show clinical benefits such as shortening the recovery time and reducing mortality in the World Health Organization's solidarity trial (Pan H et al. *N Engl J Med*. 2021). Thus, the remdesivir's inhibitory role in SARS-CoV-2 infection of placental cells observed from 2D cell culture, another major finding, might also have very limited clinical implications.

Other Comments

The ongoing pandemic raises an imperative question whether SARS-CoV-2 variants may be more likely to infect placental tissues than the original strain (Argueta LB et al. *iScience* 2022). The susceptibility of iTSC-derived placental cells to the original SARS-CoV-2 strain and its major variants (including Delta and Omicron variants) has not been studied in this study.

The reviewer appreciates the authors' efforts to develop new monoclonal antibodies and feel like that the analysis should include the current major SARS-CoV-2 variants. The authors also need to compare the potency of their antibodies with existing ones such as S2E12, REGN10933, AZD1061, CR3022, and

47D11 (Tortorici et al., Science 2020; Baum A et al., Science 2020; Zost SJ et al., Nature 2020; Yuan M et al., Science 2021; Wang C et al., Nat Commun 2020).

FINANCIAL AND NON-FINANCIAL COMPETING INTERESTS – the authors must include one of three declarations: (1) that they have no financial and non-financial competing interests; (2) that they have financial and non-financial competing interests; or (3) that they decline to respond, after the Author Contributions section. This statement will be published with the article, and in cases where financial

and non-financial competing interests are declared, these will be itemized in a web supplement to the article. For further details please see <https://www.nature.com/licenceforms/nrg/competing-interests.pdf>.

Methods should be written concisely, but should contain all elements necessary to allow interpretation and replication of the results. As a guideline, Methods sections typically do not exceed 3,000 words. The Methods should be divided into subsections listing reagents and techniques. When citing previous methods, accurate references should be provided and any alterations should be noted. Information must be provided about: antibody dilutions, company names, catalogue numbers and clone numbers for monoclonal antibodies; sequences of RNAi and cDNA probes/primers or company names and catalogue numbers if reagents are commercial; cell line names, sources and information on cell line identity and authentication. Animal studies and experiments involving human subjects must be reported in detail, identifying the committees approving the protocols. For studies involving human subjects/samples, a statement must be included confirming that informed consent was obtained. Statistical analyses and information on the reproducibility of experimental results should be provided in a section titled "Statistics and Reproducibility".

All Nature Cell Biology manuscripts submitted on or after March 21 2016 must include a Data availability statement at the end of the Methods section. For Springer Nature policies on data availability see <http://www.nature.com/authors/policies/availability.html>; for more information on this particular policy see <http://www.nature.com/authors/policies/data/data-availability-statements-data-citations.pdf>. The Data availability statement should include:

- Accession codes for primary datasets (generated during the study under consideration and designated as "primary accessions") and secondary datasets (published datasets reanalysed during the study under consideration, designated as "referenced accessions"). For primary accessions data should be made public to coincide with publication of the manuscript. A list of data types for which submission to community-endorsed public repositories is mandated (including sequence, structure, microarray, deep sequencing data) can be found here <http://www.nature.com/authors/policies/availability.html#data>.
- Unique identifiers (accession codes, DOIs or other unique persistent identifier) and hyperlinks for datasets deposited in an approved repository, but for which data deposition is not mandated (see here for details <http://www.nature.com/sdata/data-policies/repositories>).
- At a minimum, please include a statement confirming that all relevant data are available from the authors, and/or are included with the manuscript (e.g. as source data or supplementary information), listing which data are included (e.g. by figure panels and data types) and mentioning any restrictions

on availability.

- If a dataset has a Digital Object Identifier (DOI) as its unique identifier, we strongly encourage including this in the Reference list and citing the dataset in the Methods.

We recommend that you upload the step-by-step protocols used in this manuscript to the Protocol Exchange. More details can found at www.nature.com/protocolexchange/about.

All imaging data should be accompanied by scale bars, which should be defined in the legend. Cropped images of gels/blots are acceptable, but need to be accompanied by size markers, and to retain visible background signal within the linear range (i.e. should not be saturated). The boundaries of panels with low background have to be demarked with black lines. Splicing of panels should only be considered if unavoidable, and must be clearly marked on the figure, and noted in the legend with a statement on whether the samples were obtained and processed simultaneously. Quantitative comparisons between samples on different gels/blots are discouraged; if this is unavoidable, it should only be performed for samples derived from the same experiment with gels/blots were processed in parallel, which needs to be stated in the legend.

- We do not recommend using Adobe Photoshop for designing figures, but we can accept Photoshop generated (.PSD or .TIFF) files only if each element included in the figure (text, labels, pictures, graphs, arrows and scale bars) are on separate layers. All text should be editable in 'type layers' and

line-art such as graphs and other simple schematics should be preserved and embedded within 'vector smart objects' - not flattened raster/bitmap graphics.

The total number of Supplementary Figures (not including the "unprocessed scans" Supplementary Figure) should not exceed the number of main display items (figures and/or tables (see our Guide to Authors and March 2012 editorial <http://www.nature.com/ncb/authors/submit/index.html#suppinfo>;

<http://www.nature.com/nbc/journal/v14/n3/index.html#ed>). No restrictions apply to Supplementary Tables or Videos, but we advise authors to be selective in including supplemental data.

GUIDELINES FOR EXPERIMENTAL AND STATISTICAL REPORTING

REPORTING REQUIREMENTS – To improve the quality of methods and statistics reporting in our papers we have recently revised the reporting checklist we introduced in 2013. We are now asking all life sciences authors to complete two items: an Editorial Policy Checklist (found here <https://www.nature.com/authors/policies/Policy.pdf>) that verifies compliance with all required editorial policies and a reporting summary (found here <https://www.nature.com/authors/policies/ReportingSummary.pdf>) that collects information on experimental design and reagents. These documents are available to referees to aid the evaluation of the manuscript. Please note that these forms are dynamic 'smart pdfs' and must therefore be downloaded and completed in Adobe Reader. We will then flatten them for ease of use by the reviewers. If you would like to reference the guidance text as you complete the template, please access these flattened versions at <http://www.nature.com/authors/policies/availability.html>.

Author Rebuttal to Initial comments

Point by point response to the reviewers comments

We extend our sincere thanks for the constructive feedback and time and effort in considering our manuscript. We believe we have addressed all comments and requested changes, which has helped us improve our manuscript considerably. Below we provide a point-by-point response to all comments and suggestions.

Reviewers' Comments:

Reviewer #1:

Remarks to the Author:

This manuscript from Jose Polo and colleagues examines SARS-CoV-2 infection in placental cell types generated from induced trophoblast stem cells (iTSCs). Clinical reports have linked COVID-19 infection with adverse pregnancy outcomes, but the potential of SARS-CoV-2 to infect early placental cells remains poorly understood. The authors show that the entry factors ACE2 and TMPRSS2 are expressed in two differentiated trophoblast cell types (EVTs and STs), but only STs are productively infected. They also report that infected ST cultures have reduced expression of differentiation genes and that an anti-ACE2 antibody or antiviral drugs can prevent SARS-CoV-2 infection.

In vitro models of placental infection by SARS-CoV-2 are of considerable current interest. However, the authors must clarify several points (please see below) and some of the figures appeared disorganized in the PDF (especially Fig. 4, Extended Data Figs. 1 and 2), which made it difficult to assess the data.

Response to Reviewer: We thank the reviewer for taking the time to review our manuscript and for the encouraging and very constructive comments. We apologise if some of our figures look disorganised, we have now tried to address this. It appears that when we submitted the figures through the manuscript submission portal the quality decreased upon uploading. Again, we apologise for the inconvenience.

1. Fig. 1a shows ACE2 expression in 1st trimester placental villi and decidua, but TMPRSS2 is also required for SARS-CoV-2 infection. Did the authors check expression of TMPRSS2 in these same placental tissues by immunohistochemistry?

Response to Reviewer: We thank the reviewer for highlighting TMPRSS2 as another marker to identify susceptibility. We have now performed TMPRSS2 staining on placental tissues (villi and decidua sections) and our cell lines (iTSC, EVT and ST). This new data is shown in Extended Data Fig. 1a-c. We observed minimal expression of TMPRSS2 in the villi and decidua. In iTSC, EVT and ST, we observed no TMPRSS2 protein expression by immunofluorescence. These data are in agreement with Karvas et al. who have recently shown that placental organoids have very little overlap of ACE2 and TMPRSS2 expression (Karvas et al., *Cell stem cell* 2022).

We would like to note that even though TMPRSS2 is a co-factor that is involved in ACE2 receptor-mediated entry, cells without TMPRSS2 are still capable of SARS-CoV-2 viral entry via the cathepsin B/L-mediated endocytosis (Gartner and Subbarao, *Microbiology Australia* 2021). Therefore, infection of STs in our model is ACE2-dependent but likely occurs independently of TMPRSS2. We have discussed these points in the results section of our manuscript (STs are productively infected with SARS-CoV-2).

2. Alveolar lung epithelial (AT2) cells are used as a positive control for the ACE2 antibody in Extended Data Fig. 1. How do the transcript levels of ACE2 and TMPRSS2

in iTSC, EVT and ST in Fig. 1d compare with AT2 cells? The upregulation of TMPRSS2 in specialized trophoblast cell types is much lower than ACE2. I wonder how significant it is relative to a known receptive cell type.

Response to Reviewer: As requested, in addition to the aforementioned query of TMPRSS2 protein expression, we have now also included qPCR analysis of TMPRSS2 mRNA expression in lung AT2 cells in comparison to the cell types (iTSC, EVT and ST) in this study. This new data is shown in Fig. 1d, and Extended Data in Fig. 3d. The data show that while EVTs and STs have upregulation of TMPRSS2 compared to iTSCs, it is less than half of the expression observed in lung AT2 cells. We also show minimal change in TMPRSS2 expression through ST differentiation. As mentioned above and also in the text (with relevant references), this is not entirely surprising based on previous studies and suggests that infection of STs is likely independent of TMPRSS2.

3. Along similar lines, it would be helpful to compare the levels of viral titer following SARS-CoV-2 infection in trophoblast and alveolar cells side-by-side in Fig. 1f.

Response to Reviewer: We thank the reviewer for the suggestion. We have analysed the growth of SARS-CoV-2 in lung AT2 cells in an independent manuscript (Rudraraju *et al.*, *bioRxiv* 2022). Therefore, we cannot include the data in this manuscript for direct side-by-side comparison. However, as that manuscript is now in bioRxiv we have referred to it in the text of our results section. While the culture conditions differ between these two cell types (i.e number of cells per well and multiplicity of infection), we note that the viral titre at days 3-4 in lung AT2 cells was approximately 10^6 to 10^7 TCID₅₀/ml in comparison to STs where virus titres reached approximately 10^4 to 10^6 TCID₅₀/ml. Therefore, the maximal viral titre in STs was at least 10 times lower than in lung AT2 cells. We have included a sentence discussing this comparison in the results section (**STs are productively infected with SARS-CoV-2**).

Although the reasons for the difference in viral growth is unclear, we know that viral replication in lung AT2 cells is dependent on TMPRSS2 and induces minimal interferon signalling (Rudraraju *et al.*, *bioRxiv* 2022) in contrast with STs where viral entry is likely TMPRSS2-independent and induces a robust inflammatory signature. The possibility of these factors contributing to viral replication is now addressed in the discussion.

4. The dsRNA staining in Fig. 1k suggests that only a fraction of STs are infected by SARS-CoV-2, which is consistent with a recent report (Karvas et al., Cell Stem Cell, 2022). Can the authors determine the percentage of STs that shows dsRNA expression following SARS-CoV-2 infection?

Response to Reviewer: We thank the reviewer for the suggestion, we have now counted the number of dsRNA+ cells to determine the percentage infected. These new data are shown in Extended Data Fig. 1m and Figure 1h (originally Fig. 1k). We found that approximately 57% of STs are dsRNA+ at day 3 post-infection. This is significantly higher than the percent reported by Karvas et al. where only approximately 1.6% of 2D STs cells were infected. Therefore, our iTSC-derived STs are a more robust model of SARS-CoV-2 infection and the comparison is now given in the results section. Other than differences in our model systems explaining this discrepancy, Karvas et al. also only analysed 24 hours post-infection which,

according to our data, may be too early to detect robust virus replication in ST cells. This point is now addressed in the discussion.

5. The authors use morphological criteria to assess the extent of ST differentiation based on foot-processes/microvilli. I could not find a detailed explanation of this analysis in the Methods. Is it possible to use a more quantitative criterion like fusion index?

Response to Reviewer: We thank the reviewer for highlighting this and apologise that we did not make it clear enough in the methods. We have now included a more comprehensive description of our morphology analysis in the methods section (Image Analysis/Cell Morphology and Quantification). Additionally, as suggested we have now added a fusion index analysis. The new data is shown in Fig. 2e. The data show that infected cells (dsRNA+) have a lower fusion index as compared to dsRNA- cells. This supports the morphological and HCG expression analysis and further confirms that both cell fusion and differentiation are impaired in infected STs.

6. Transcriptional profiling shows an interesting difference between virus- and mock-infected STs (Fig. 3a), which is reversed by anti-ACE2 antibody treatment (Fig. 4c). Can the authors identify the developmental stage at which ST differentiation is stalled using single cell data from early placental tissues (e.g. Chen et al., Development, 2022; Shannon et al., Development, 2022)?

Response to Reviewer: Great suggestion, we have now compared our RNA-seq data to single cell RNA-seq (scRNA-seq) of early placental tissues (Shannon *et al.*, Development 2022). The data are shown in the new Fig. 3g-h, Fig. 4f and Extended Data Fig. 6e-f. The data indicate that SARS-CoV-2 infected STs in our study clustered closer to placental cytotrophoblast cells (less differentiated) than the mock-infected cells, and were enriched for an undifferentiated cytotrophoblast signature, in contrast to the mock samples which were enriched for an ST progenitor signature. Furthermore, both the CoA and hierarchical clustering analysis showed that treatment with anti-ACE2 antibody led to clustering of Virus^{aACE2} along with the mock infected/treated samples away from the CTs and towards the ST progenitors.

7. Extended Data Fig. 4b: RNA expression across SARS-CoV-2 genomic elements does not look very significant in the virus-infected sample except for the elements

towards the very edge of the track. Why do those elements (e.g. ORF6/ORF7a) show lower expression compared to nsp2/3 etc. in the heatmap in the next panel (Extended Data Fig. 4c)?

Response to reviewer: We thank the reviewer for the question and have tried to improve our explanation and visualisation.

Our library preparation was performed with a poly-A capture, hence we would only expect to detect the 3' regions of genes in our transcripts. The SARS-CoV-2 genome is expected to produce one transcript so we would expect to observe the majority of the reads at the 3' end of the gene (ORF10 or N). For this reason we checked the absolute expression (in counts per million) to see where the most abundance of transcripts were along the genome (now Extended Data Fig. 6b).

We observed a few reads mapped outside of the 3' end of the transcript, this could have been due to alternative splicing, or incorrect mapping, or because non poly-A reads were captured. Interestingly, even reads outside the 3' end of the viral genome were still higher in Virus^{Ctrl} than all other samples (now Extended Data Fig. 6c). ORF6/ORF7a shows a relatively lower expression to nsp2/nsp3 because these regions contain less transcripts mapped to them.

Reviewer #2:

Remarks to the Author:

The manuscript “An iTSC-derived placental model of SARS-CoV-2 infection reveals ACE2-dependent susceptibility in syncytiotrophoblasts” by Chen et al examined induced fibroblast derived trophoblast stem cell cultures during SARS-CoV-2 infection. Several groups have previously shown that COVID-19 is associated with adverse fetal outcomes including miscarriages and placental alterations. The authors identify that the placental-specific cell types express ACE2 and TMPRSS2 but only the syncytiotrophoblasts show evidence of SARS-CoV-2 infection. Follow-on experiments showed that SARS-CoV-2 infected syncytiotrophoblasts have altered transcriptomes marked by diminished hallmarks of syncytiotrophoblast differentiation and function. Despite these findings there are several areas of concern that dampen the enthusiasm for this manuscript and as a consequence this manuscript should be rejected with hope for resubmission.

Major Comments

1) Figure 4, and extended figures 1 and 2 are not able to be displayed properly which greatly impacts the ability to accurately interpret the figure findings. This needs to be addressed to properly interpret the data presented in these figures.

Response to reviewer: We apologise to the reviewer for this problem and understand how frustrating this issue would have been during the review process. We appreciate the effort to provide constructive comments despite these issues. It appears that when we submitted the figures through the manuscript submission portal the quality decreased upon uploading. We have now made sure that the quality of the merged figures is conserved and we have also asked the editors to make the original figures available to the reviewers if necessary. Again, we apologise for the inconvenience.

2) In figure 1 the authors show that in the syncytiotrophoblast culture there are dsRNA + cells that also are HCG negative. It is not clear if this is due to infection or if these cells were not HCG+ to begin with. What is the efficiency of EVT and ST differentiation from iTSC and what is the purity of these cultures at the time of infection. In addition to dsRNA staining, evaluation with additional markers of SARS-CoV-2 infection would add additional markers of infection (e.g. SARS-N/SARS-S etc). Why was 3 DPI used as a time point for infection? What happens longitudinally over time?

Response to reviewer: We thank the reviewer for these queries and have answered each individually below.

For the query about efficiency of EVT and ST differentiation, we and others have previously reported that these cultures are reasonably homogenous (Okae *et al.*, *Cell stem cell* 2018; Liu *et al.*, *Nature* 2020; Tan, Liu and Polo, *Nature protocols* 2022). For ST differentiation specifically, the majority of cells express HCG as observed in mock-infected controls shown in Fig. 1 and Extended Data Fig. 1. Additionally, cells in our cultures start to express HCG from day 2 (Extended Data Fig. 3a,b) and this is consistent throughout the differentiation process.

In response to the reviewer’s request for an additional marker to detect SARS-CoV-2 virus, we have now included staining of nucleocapsid to validate the presence of intracellular virus (Extended Data Fig. 1I).

As for the timing of analysis, we selected 3 days post-infection because virus titres peaked on day 3 allowing us to confidently assess the impact of virus infection on the cells. In support of this approach, our data show that at day 1 post-infection, virus growth is modest (Fig. 1). Also, when we infect cells at day 3 of differentiation, we can analyse STs at day 6 of differentiation (3 days into differentiation + 3 days post-infection) when they should be terminally differentiated. We discuss this point in the results section (Transcriptome-wide profiles of infected STs show an increase in viral responses). Although we did not assess infection in our 2D model beyond day 3 post-infection, we recognize the reviewer’s concern about the effects of the virus at later time points. We have now also included additional data of a co-culture experiment using trophoblast organoids and endometrial epithelial cells infected over 6 days (Fig. 2h-k; Extended Data Fig. 3i,j).

3) It is not clear from the methods whether the TCID₅₀ were calculated using a qRT-PCR method or via a plaque assay method. If only completed by qRT-PCR a plaque assay method should be used for validation.

Response to reviewer: We apologise if this was not clear in our methods. To assess viral replication in these cells we measured infectious virus titres by infectivity assay (TCID₅₀/ml) and viral genomes by RT-PCR (Genomes per ml) in the cell supernatant. The infectivity assay is equivalent to the plaque assay method as these techniques both quantify infectious virus in a susceptible cell line (ie. Vero cells). We have added more information in our methods section (SARS-CoV-2 infection) regarding this analysis.

4) The authors show that ACE2 expression increases significantly soon after initiation of ST differentiation. While this reviewer can see parts of this increase on IF (due to figure obstruction in extended figure 2), western blot do quantitatively validate this would more clearly show these changes over time.

Response to reviewer: We thank the reviewer for this suggestion and apologise again for the figure obstruction. In Figure 2, we show that both ACE2 mRNA expression (RT-PCR) and

protein expression (IF) are observed as early as day 2 of differentiation. To further confirm this result, we have now additionally run a western blot to assess ACE2 protein levels in STs over time. The data shown in Extended Data Fig. 3c indicates that ACE2 expression in STs by western blot is similar to our observations using immunofluorescence. This supports our conclusion that ACE2 expression is rapidly induced during differentiation of iTSCs to STs.

5) It is surprising that the authors see such a prominent impact of SARS-CoV-2 infection of bHCG production into the supernatant given that the uninfected cells are differentiating properly. What percentage of the cells are infected (as the IF images suggest that it is not a high percentage)? How do the authors explain this discrepancy?

Response to reviewer: As per reviewer 1's suggestion we have now analysed the number of dsRNA+ cells to determine the percentage of infected cells. This new data is shown in Extended Data Fig. 1m. Approximately 57% of cells were dsRNA+ following SARS-CoV-2 infection. This correlated with at least a 50% reduction in HCG secretion (Fig. 2, 4). The decrease in HCG secretion is likely due to: 1) infected STs failing to complete their differentiation, 2) a modest increase in the amount of cell death/cell loss and 3) defective production of HCG by the remaining infected cells. Overall, we believe that it is feasible that this proportion of infected cells correlates with a prominent impact on HCG secretion.

6) The authors show increased levels of LDH during SARS-CoV-2 infection of ST cultures and suggest that there are increased levels of cell death but do not demonstrate increased levels of cell loss. Deeper evaluation of these cultures including evidence of cell loss and/or death using markers for apoptosis, necroptosis or necrosis should be completed.

Response to reviewer: We thank the reviewer for this comment. We would like to reiterate that in the original manuscript we showed that following SARS-CoV-2 infection there is a significant increase in extracellular LDH which is indicative of cell death (Fig. 2g). We also identified 14 significantly up-regulated DEGs by RNA-Seq involved in positive regulation of cell death (Fig. 3). However, we noted that these genes were not highly enriched. These results suggested that cell death was likely to be modest after infection.

To confirm this we have now measured cell viability using the Promega CellTiter-Glo kit. This is also a surrogate measure for cell loss as it detects ATP production by metabolically active cells within the culture. This new data is shown in Extended Data Fig. 3g. It indicates that **only** an approximate 25% loss of cell viability is observed by day 3 post infection. We also assessed cell viability at day 5 post-infection in a single experiment and again observed a modest amount of cell death (data not shown). This result is consistent with Argueta et al, who showed only a trend for increased cell death genes linked to apoptosis following SARS-CoV-2 infection of placental clusters (Argueta et al., *iScience* 2022). It also demonstrates that the effect of SARS-CoV-2 infection of STs is predominantly on differentiation potential and not viability. We have now included these points in the discussion.

Although the degree of cell death was modest, we attempted to determine the mechanism of cell death to address the reviewer's suggestion. The new data is shown in Extended Data Fig. 3h. We saw evidence for a trend in increased caspase 3/7 activity suggestive of apoptosis. While these observations were not statistically significant, they are suggestive of a role for apoptosis-mediated cell death in STs following SARS-CoV-2 infection, albeit in a minor proportion of cells.

7) It is not clear if the described studies were completed with one donor TSC cell line with repeated experiments (for the transcriptome) and are generalizable or unique to one line or two lines as referenced in figure 1? This needs to be validated with several different donor lines to demonstrate this generalizability and findings.

Response to reviewer: We apologise if this was not clear in the text. We indeed assessed virus growth in STs, EVT_s and TSC_s in 2 donors (32F and 55F, Fig. 1). We have now also validated infection of STs in an additional primary cell line (FT008). The new data is shown in Extended Data Fig. 2c,d. In addition, the assessment of HCG and LDH levels shown in Fig. 2 was completed in 2 donors (32F and 55F). For the transcriptomics and antibody blockade experiments (Fig. 3 and 4), a single donor (55F) was used as a representative cell line. To ensure transparency, the donor cell line used for each experiment is now given in the figure legend. In addition, we have indicated in the section 'Transcriptome-wide profiles of infected STs show an increase in viral responses' that we have used 55F as a representative cell line for these studies.

8) The authors need to extend their findings in better context of the field and prior reports of placental infection *in vitro* and *in vivo*. Only minimal discussion is present and it primarily is in the context of a prior report that claims there was no infection of trophoblast organoids.

Response to reviewer: We thank the reviewer for highlighting this to us. We have now expanded our introduction on 1) ACE2 and TMPRSS2 expression, 2) SARS-CoV-2 infection and 3) infection outcomes from previous *in vivo* and *in vitro* studies. With these inclusions, we believe that we have adequately canvassed the field prior to our study. In addition, we have now specifically discussed our findings with the limited infection in other studies using *in vitro* placental models to study SARS-CoV-2 infection, compared our findings to other viruses that infect the placenta and discussed how our findings align with *in vivo* studies.

Minor comments

1) Several spelling errors are noted in the figures and text (e.g. Figure 1g Virus).

Response to reviewer: We apologise for this and thank the reviewer for making the time to point out to this. We have reviewed the text and amended spelling and grammar errors.

Reviewer #3:

Remarks to the Author:

RE: Chen et al. An iTSC-derived placental model of SARS-CoV-2 infection reveals ACE2-dependent susceptibility in syncytiotrophoblasts

Major Comments

It is a relatively complete analysis of SARS-CoV-2 infection of an iTSC-derived placental model in 2D cell culture. It is well-written. My major concern with this manuscript is that the study lacks significant novelty as well as updated mechanistic insights into SARS-CoV-2 variant infection of placental cells.

Concerning the SARS-CoV-2 cell tropism and multiorgan infection, this topic has been extensively studied and reviewed in the past three years since the onset of the

pandemic. It is well known that almost any types of cells with the expression of SARS-CoV-2 receptors (such as ACE2 and TMPRSS2) can be infected by the viruses. As far as the placenta is concerned, there were numerous reports using placental cell culture (e.g., Fahmi A et al. Cell Reports Medicine, 2021; Argueta LB et al. iScience 2022; Karvas RM et al. Cell Stem Cell 2022), which confirmed that syncytiotrophoblasts can be efficiently infected by SARS-CoV-2, one major finding in this study.

Response to Reviewer: We thank the reviewer for the comments. On this particular point we would like to clarify that Fahmi et al. show that placental tissue sections can be infected with SARS-CoV-2 relative to ACE2 expression but did not observe any change in cytotoxicity or inflammatory markers. Unlike our study, they did not confirm the requirement for ACE2 binding for infection and they observed infection of several placental cell types in addition to STs.

Argueta et al. showed that STs are a target for SARS-CoV-2 infection *in vivo* using at-term placentas and that STs are similarly infected in primary placental clusters associated with transcriptional responses suggestive of cell death and inflammation. All of their work on viral receptors and variants was done using pseudoviruses. While our data corroborate some of these findings, it is important to note that our model is representative of early placenta, not at-term placenta. This is important because infection during the first trimester is most highly associated with adverse foetal outcomes. Importantly, all our work was done using live SARS-CoV-2 virus and not pseudoviruses.

Karvas et al. showed that both their organoid system and 2D placental cells, which model early placenta, have very little infection with SARS-CoV-2 (most of this work was done with pseudoviruses) and did not show any additional data on cellular response. Therefore, while their 3D system offers exciting potential for infection with Zika virus, their study of SARS-CoV-2 infection is limited and was not the main focus of their manuscript. For these reasons we feel that while these studies also identified STs as the major target for SARS-CoV-2 infection, we have significantly advanced the knowledge beyond their work to include analysis of viral receptors, antivirals, variants, cell function and cell death.

With respect to placental cellular models, physiologically relevant ones should be highly encouraged, which may address the complexity of the placental biology and pathophysiology. Recently, Theunissen and colleagues described such a 3D trophoblast organoid model, derived from naive human pluripotent stem cells and primary TSCs, for studying placental susceptibility to infection of SARS-CoV-2 (Karvas RM et al. Cell Stem Cell 2022). The current 3D model still has many limitations (e.g., an inside-out-villous architecture with an inner syncytial compartment). Nevertheless, the current efforts should continue establishing such organoid type of 3D culture system, possibly including immune cells (as discussed by the authors) and the vasculature, which would faithfully recapitulate the placental architecture (e.g., the blood-placenta barrier). This would provide insights into clinically relevant problems such as stillbirth and reliable platforms for drug discovery.

Response to Reviewer: We thank the reviewer for this comment and agree that highly complex placental models that recapitulate the immune milieu and vasculature of the placenta would be extremely advantageous to the field. We have no doubt that such complex models will be developed over time and indeed we have now made a first step

towards this (see next paragraph). However, we would like to highlight the advantage of our approach that allows us to explore the direct response of specific placental cells to SARS-CoV-2 infection under several experimental conditions.

Nevertheless, following the reviewer's suggestion, we have explored the potential of using our model as a platform to study complex interactions between placental cells during SARS-CoV-2 infection. To do this, we developed a novel co-culture system with endometrial cells and trophoblast organoids during differentiation that models infection during early implantation. The new data is shown in Fig. 2h-k and Extended Data Fig. 2i,j. We show that endometrial cells can be infected with SARS-CoV-2 and they can secondarily infect adjacent trophoblast organoids that are undergoing differentiation into STs, but not non-differentiating organoids. As was seen with direct infection of STs, this leads to a reduced expression of HCG. Overall, we have shown that in a model of infection during implantation, SARS-CoV-2 reduces the differentiation potential of STs similar to our 2D ST infection model.

For example, the first FDA-approved anti-SARS-CoV-2 drug remdesivir (GS-5734), as indicated as a drug candidate that inhibit SARS-CoV-2 infection in 2D cell culture, failed to show clinical benefits such as shortening the recovery time and reducing mortality in the World Health Organization's solidarity trial (Pan H et al. N Engl J Med. 2021). Thus, the remdesivir's inhibitory role in SARS-CoV-2 infection of placental cells

observed from 2D cell culture, another major finding, might also have very limited clinical implications.

Response to reviewer: We appreciate the reviewer's comments. In this study, we chose to study the effects of Remdesivir on ST infection because of its known antiviral activity against SARS-CoV-2 in other cell culture models. Thus, the goal of this experiment was to show that, in addition to anti-ACE2 blockade of infection, antivirals that block viral replication can also restore ST function. We understand that there are conflicting results for the clinical efficacy of Remdesivir, although there is some evidence that Remdesivir treatment may be beneficial to treat severe COVID-19 during pregnancy (Lampejo, *Journal of medical virology* 2021). Remdesivir is also still recommended for use to treat severe COVID-19 by the WHO . Therefore, studying the effects of Remdesivir in our 2D placental cell model is still relevant for clinical practice. We have now addressed these points in the discussion.

Other Comments

The ongoing pandemic raises an imperative question whether SARS-CoV-2 variants may be more likely to infect placental tissues than the original strain (Argueta LB et al. *iScience* 2022). The susceptibility of iTSC-derived placental cells to the original SARS-CoV-2 strain and its major variants (including Delta and Omicron variants) has not been studied in this study.

Response to Reviewer: We appreciate the suggestion. We have now included additional data analysing the growth of Ancestral, Delta and Omicron variants in ST cells derived from 3 separate donor cell lines. The new data is shown in Figure 1f-h and Extended Data Fig. 2. STs can be infected with Delta and Omicron variants in addition to the Ancestral virus, as observed by an increase of virus in the supernatant over time (TCID₅₀ and RT-PCR) and detectable intracellular dsRNA staining at day 3 post infection. Taken together, these data show that STs can be infected by all major SARS-CoV-2 variants.

The reviewer appreciates the authors' efforts to develop new monoclonal antibodies and feel like that the analysis should include the current major SARS-CoV-2 variants. The authors also need to compare the potency of their antibodies with existing ones such as S2E12, REGN10933, AZD1061, CR3022, and 47D11 (Tortorici et al., Science 2020; Baum A et al., Science 2020; Zost SJ et al., Nature 2020; Yuan M et al., Science 2021; Wang C et al., Nat Commun 2020).

Response to Reviewer: We appreciate the comment and have now included additional data exploring the potency of our anti-ACE2 antibodies with existing anti-spike antibodies against the Ancestral and Omicron variants. However, we note that the main aim of this work was to develop an anti-ACE2 antibody as a tool rather than as a therapeutic. Thus, some classical characterisations of potency and pharmacodynamics are missing. Nevertheless, the newly generated data comparing our antibody with existing antibodies is shown in Fig. 4j,k. Ancestral and Omicron infection was inhibited with our cocktail of anti-ACE2 antibodies and with the previously described anti-spike antibody REGN10933. Consistent with what others have reported (Cao et al., Nature 2022; VanBlargan et al., Nature medicine 2022), some anti-spike antibodies (REGN10987, CR3022) have lost the ability to block infection with the Omicron variant.

j

k

Decision Letter, first revision:

Our ref: NCB-A48564A

14th March 2023

Dear Jose,

Thank you for submitting your revised manuscript "An iTSC-derived placental model of SARS-CoV-2 infection reveals ACE2-dependent susceptibility and differentiation impairment in syncytiotrophoblasts" (NCB-A48564A). It has been seen by the original referees and their comments are below. Please note that following our prior communication, during which I shared with you these comments and you provided me with a response, referee #1 further evaluated your response to their comments, but also cross-commented on how you addressed the remaining concerns by referee #2, as we did not manage to obtain the additional report from the latter. The reviewers find that the paper has improved in revision, and therefore we'll be happy in principle to publish it in Nature Cell Biology, pending minor revisions to satisfy the referees' final requests and to comply with our editorial and formatting guidelines.

If the current version of your manuscript is in a PDF format, please email us a copy of the file in an editable format (Microsoft Word or LaTeX)-- we cannot proceed with PDFs at this stage.

Thank you again for your interest in Nature Cell Biology. Please do not hesitate to contact me if you have any questions.

Best regards,
Stelios

Stylianos Lefkopoulos, PhD
He/him/his
Associate Editor
Nature Cell Biology
Springer Nature
Heidelberger Platz 3, 14197 Berlin, Germany

E-mail: stylianos.lefkopoulos@springernature.com
Twitter: @s_lefkopoulos

Reviewer #1 (Remarks to the Author):

Chen et al have provided additional data, confirming that STs can indeed be infected by SARS-CoV-2, but at 10-fold lower rate than lung AT2 cells. The authors have also added other new data, including co-culture of trophoblast organoids with SARS-CoV-2-infected endometrial cells and integration with expression data from placental tissues. Overall, this is a comprehensive analysis of SARS-CoV-2 infection in trophoblast cells that will be of interest to the community. However, the authors need to be more careful when describing their results in relation to other recent studies:

1. The recently published paper from Pentao Liu (Ruan et al., Cell Rep Med, 2022) has several overlapping findings with this study, but is currently only mentioned in passing. Ruan et al. also reported infection of ACE2-positive STs, impaired ST differentiation in SARS-CoV-2-infected trophoblast cells, upregulation of interferon signaling, and suppression by Remdesivir. Given these overlapping findings, it would be better to mention this work explicitly in the Discussion and in the summary of prior literature on p. 4.

2. Karvas et al. did not report an infection rate of 1.6% in 2D STs, as is currently stated on p. 7. That percentage referred to infection of undifferentiated TSCs (see Figure S5C in Karvas), which actually agrees quite well with the data in this manuscript. Karvas et al. did not quantify the proportion of infected STs, but only reported that they saw more robust infection by VSV. Of note, Ruan et al. did quantify the proportion of their infected STs (~10% at 24h post-infection, see Figure S3Q in Ruan). The difference with the 57% of infection in ST cells in this manuscript may be related to timing or the method of detection (dsRNA vs. SARS-CoV-2 N). The dsRNA antibody used here is actually not specific to SARS-CoV-2 and could potentially reflect a broader stress response independent of viral infection.

3. It seems to me that the three abovementioned studies are largely in agreement on key findings: SARS-CoV-2 can productively infect a subset of ST cells, but only has limited capacity to infect trophoblast organoids. The latter was well-documented in the Karvas and Ruan papers and confirmed in Fig. 2j of the present manuscript, in which no significant viral infection was seen in organoids that were maintained in TO medium. While the percentage of infected STs varies between Ruan and this manuscript, this may be explained by technical variables (see point 2 above). I think it would be helpful to emphasize the common points raised by these three independent studies in the Discussion.

Additional comments

The authors have addressed my concerns and now quantify the proportion of SARS-CoV-2 infected STs based on nucleocapsid staining as well as dsRNA. They also added a more detailed discussion of the paper by Ruan et al.

I only noticed one remaining issue, which can be easily addressed. The authors write in their Discussion: "However, contrary to Karvas et al., we confirmed the findings of Ruan et al., that organoids can be infected upon ST differentiation, although the percentage of infected cells varies between studies, as discussed above)."

Actually, neither the Karvas nor Ruan studies performed ST differentiation on trophoblast organoids. Both those studies only performed ST differentiation on 2D-cultured hTSCs, as described in the Okae paper. As far as I'm aware, this manuscript is the first to report successful ST differentiation of trophoblast organoids. Therefore, I suggest modifying this sentence slightly:

"However, in contrast to these two prior studies, we demonstrate that trophoblast organoids can be infected by SARS-CoV-2 upon ST differentiation." This will provide a good introduction to the

description of the secondary infection model in which the authors use their novel ST-cultured trophoblast organoids.

Additional comments on authors' response to Reviewer #2

I've read the authors' response to the remaining comments from referee #2 and am happy to say that they've addressed those concerns adequately. Overall, the data in Fig. 4h support the authors' conclusion about the effect of the ACE2 neutralizing antibody.

Reviewer #2 (Remarks to the Author):

The revised manuscript "An iTSC-derived placental model of SARS-CoV-2 infection reveals ACE2-dependent susceptibility in syncytiotrophoblasts" by Chen et al examines induced fibroblast derived trophoblast stem cell cultures during SARS-CoV-2 infection. This substantially revised manuscript addresses the majority of concerns raised by this and the other reviewers. Only a few minor questions remain.

Major Comments

- 1) It is not clear where the neutralizing ACE2 antibodies and used plasmids identified and generated by the investigators would be available from the investigators or certain public repositories. This should be stated in the methods.
- 2) The transcriptome work strongly supports the investigator hypothesis. No GEO reference or publicly available database is made for where this data will be located.
- 3) In figure 4h the transcriptomic changes are evaluated using GSVA signatures for each cell type and notes the differences in resulting signature after virus/ACE2 neutralizing ab exposure. The ACE2 neutralizing antibody treated mock cells seems to have an altered expression of these signatures as well with a noted increase in the CT2 signature. What is accounting for these changes? Do you see impacts on cellular differentiation status or cellular composition in that control group?

Reviewer #3 (Remarks to the Author):

RE: NCB-A48564A: Chen et al. An iTSC-derived placental model of SARS-CoV-2 infection reveals ACE2-dependent susceptibility in syncytiotrophoblasts

Major Comments

The revised manuscript consists of a significant amount of new data that have appropriately addressed the reviewer's major concerns.

Other Comments

1. Pages 10-12: Under the "SARS-CoV-2 affects differentiation potential and metabolic activity" section: The term "Metabolic activity" is not accurate. The authors only measured LDH levels as a marker for cell death under this section, without examining metabolic activity (e.g., glycolysis), metabolic pathways, and metabolites. So, "metabolic activity" may be deleted throughout the text where no actual metabolic assays were examined.

2. When genes are expressed at the non-protein level, their symbols should be italicized as required by the HUGO nomenclature. Please check this throughout the text (e.g., pages 15 and 16) and revise them accordingly.

3. "UMAP projection" is mentioned in multiple figures without providing its full term for its first appearance in the text or in the legends of figures. Also, what is the "Correspondence analysis (CoA)" and components 1, 2, and 3? Is CoA the principal component analysis (PCA)? Please clarify them and provide some details (e.g., two or three sentences for each analysis) in the methods section.

4. Too much data is presented in four figures. Please consider using 6 to 7 figures as permitted in a research article by this journal.

5. A structural/anatomical cartoon with a major experimental outline (in Figure 1a), including the main placental cell types: cytotrophoblasts, syncytiotrophoblasts (STs), and extravillous cytotrophoblasts (EVTs), etc in early placental development, would enhance the presentation of the data for a broader readership.

Decision Letter, second revision:

Our ref: NCB-A48564A

28th March 2023

Dear Dr. Polo,

Thank you for your patience as we've prepared the guidelines for final submission of your Nature Cell Biology manuscript, "An iTSC-derived placental model of SARS-CoV-2 infection reveals ACE2-dependent susceptibility and differentiation impairment in syncytiotrophoblasts" (NCB-A48564A). Please carefully follow the step-by-step instructions provided in the attached file, and add a response in each row of the table to indicate the changes that you have made. Please also check and comment on any additional marked-up edits we have proposed within the text. Ensuring that each point is addressed will help to ensure that your revised manuscript can be swiftly handed over to our production team.

If you have not done so already, please alert us to any related manuscripts from your group that are under consideration or in press at other journals, or are being written up for submission to other journals (see: <https://www.nature.com/nature-research/editorial-policies/plagiarism#policy-on->

duplicate-publication for details).

In recognition of the time and expertise our reviewers provide to Nature Cell Biology's editorial process, we would like to formally acknowledge their contribution to the external peer review of your manuscript entitled "An iTSC-derived placental model of SARS-CoV-2 infection reveals ACE2-dependent susceptibility and differentiation impairment in syncytiotrophoblasts". For those reviewers who give their assent, we will be publishing their names alongside the published article.

Nature Cell Biology offers a Transparent Peer Review option for new original research manuscripts submitted after December 1st, 2019. As part of this initiative, we encourage our authors to support increased transparency into the peer review process by agreeing to have the reviewer comments, author rebuttal letters, and editorial decision letters published as a Supplementary item. When you submit your final files please clearly state in your cover letter whether or not you would like to participate in this initiative. Please note that failure to state your preference will result in delays in accepting your manuscript for publication.

Cover suggestions

As you prepare your final files we encourage you to consider whether you have any images or illustrations that may be appropriate for use on the cover of Nature Cell Biology.

Nature Cell Biology has now transitioned to a unified Rights Collection system which will allow our Author Services team to quickly and easily collect the rights and permissions required to publish your work. Approximately 10 days after your paper is formally accepted, you will receive an email in providing you with a link to complete the grant of rights. If your paper is eligible for Open Access, our Author Services team will also be in touch regarding any additional information that may be required to arrange payment for your article.

Please note that *Nature Cell Biology* is a Transformative Journal (TJ). Authors may publish their research with us through the traditional subscription access route or make their paper immediately open access through payment of an article-processing charge (APC). Authors will not be required to make a final decision about access to their article until it has been accepted. Find out more about Transformative Journals

Please use the following link for uploading these materials:
[Redacted]

Best regards,

Kendra Donahue
Staff
Nature Cell Biology

On behalf of

Stylios Lefkopoulos, PhD
He/him/his
Associate Editor
Nature Cell Biology
Springer Nature
Heidelberger Platz 3, 14197 Berlin, Germany

E-mail: stylios.lefkopoulos@springernature.com
Twitter: [@s_lefkopoulos](https://twitter.com/s_lefkopoulos)

Reviewer #1:

Remarks to the Author:

Chen et al have provided additional data, confirming that STs can indeed be infected by SARS-CoV-2, but at 10-fold lower rate than lung AT2 cells. The authors have also added other new data, including co-culture of trophoblast organoids with SARS-CoV-2-infected endometrial cells and integration with

expression data from placental tissues. Overall, this is a comprehensive analysis of SARS-CoV-2 infection in trophoblast cells that will be of interest to the community. However, the authors need to be more careful when describing their results in relation to other recent studies:

1. The recently published paper from Pentao Liu (Ruan et al., Cell Rep Med, 2022) has several overlapping findings with this study, but is currently only mentioned in passing. Ruan et al. also reported infection of ACE2-positive STs, impaired ST differentiation in SARS-CoV-2-infected trophoblast cells, upregulation of interferon signaling, and suppression by Remdesivir. Given these overlapping findings, it would be better to mention this work explicitly in the Discussion and in the summary of prior literature on p. 4.

2. Karvas et al. did not report an infection rate of 1.6% in 2D STs, as is currently stated on p. 7. That percentage referred to infection of undifferentiated TSCs (see Figure S5C in Karvas), which actually agrees quite well with the data in this manuscript. Karvas et al. did not quantify the proportion of infected STs, but only reported that they saw more robust infection by VSV. Of note, Ruan et al. did quantify the proportion of their infected STs (~10% at 24h post-infection, see Figure S3Q in Ruan). The difference with the 57% of infection in ST cells in this manuscript may be related to timing or the method of detection (dsRNA vs. SARS-CoV-2 N). The dsRNA antibody used here is actually not specific to SARS-CoV-2 and could potentially reflect a broader stress response independent of viral infection.

3. It seems to me that the three abovementioned studies are largely in agreement on key findings: SARS-CoV-2 can productively infect a subset of ST cells, but only has limited capacity to infect trophoblast organoids. The latter was well-documented in the Karvas and Ruan papers and confirmed in Fig. 2j of the present manuscript, in which no significant viral infection was seen in organoids that were maintained in TO medium. While the percentage of infected STs varies between Ruan and this manuscript, this may be explained by technical variables (see point 2 above). I think it would be helpful to emphasize the common points raised by these three independent studies in the Discussion.

Additional comments

The authors have addressed my concerns and now quantify the proportion of SARS-CoV-2 infected STs based on nucleocapsid staining as well as dsRNA. They also added a more detailed discussion of the paper by Ruan et al.

I only noticed one remaining issue, which can be easily addressed. The authors write in their Discussion: "However, contrary to Karvas et al., we confirmed the findings of Ruan et al., that organoids can be infected upon ST differentiation, although the percentage of infected cells varies between studies, as discussed above)."

Actually, neither the Karvas nor Ruan studies performed ST differentiation on trophoblast organoids. Both those studies only performed ST differentiation on 2D-cultured hTSCs, as described in the Okae paper. As far as I'm aware, this manuscript is the first to report successful ST differentiation of trophoblast organoids. Therefore, I suggest modifying this sentence slightly:

"However, in contrast to these two prior studies, we demonstrate that trophoblast organoids can be infected by SARS-CoV-2 upon ST differentiation." This will provide a good introduction to the description of the secondary infection model in which the authors use their novel ST-cultured trophoblast organoids.

Additional comments on authors' response to Reviewer #2

I've read the authors' response to the remaining comments from referee #2 and am happy to say that they've addressed those concerns adequately. Overall, the data in Fig. 4h support the authors' conclusion about the effect of the ACE2 neutralizing antibody.

Reviewer #2:

Remarks to the Author:

The revised manuscript "An iTSC-derived placental model of SARS-CoV-2 infection reveals ACE2-dependent susceptibility in syncytiotrophoblasts" by Chen et al examines induced fibroblast derived trophoblast stem cell cultures during SARS-CoV-2 infection. This substantially revised manuscript addresses the majority of concerns raised by this and the other reviewers. Only a few minor questions remain.

Major Comments

- 1) It is not clear where the neutralizing ACE2 antibodies and used plasmids identified and generated by the investigators would be available from the investigators or certain public repositories. This should be stated in the methods.
- 2) The transcriptome work strongly supports the investigator hypothesis. No GEO reference or publicly available database is made for where this data will be located.
- 3) In figure 4h the transcriptomic changes are evaluated using GSVA signatures for each cell type and notes the differences in resulting signature after virus/ACE2 neutralizing ab exposure. The ACE2 neutralizing antibody treated mock cells seems to have an altered expression of these signatures as well with a noted increase in the CT2 signature. What is accounting for these changes? Do you see impacts on cellular differentiation status or cellular composition in that control group?

Reviewer #3:

Remarks to the Author:

RE: NCB-A48564A: Chen et al. An iTSC-derived placental model of SARS-CoV-2 infection reveals ACE2-dependent susceptibility in syncytiotrophoblasts

Major Comments

The revised manuscript consists of a significant amount of new data that have appropriately addressed the reviewer's major concerns.

Other Comments

1. Pages 10-12: Under the "SARS-CoV-2 affects differentiation potential and metabolic activity" section: The term "Metabolic activity" is not accurate. The authors only measured LDH levels as a marker for cell death under this section, without examining metabolic activity (e.g., glycolysis), metabolic pathways, and metabolites. So, "metabolic activity" may be deleted throughout the text where no actual metabolic assays were examined.

2. When genes are expressed at the non-protein level, their symbols should be italicized as required by the HUGO nomenclature. Please check this throughout the text (e.g., pages 15 and 16) and revise them accordingly.

3. "UMAP projection" is mentioned in multiple figures without providing its full term for its first appearance in the text or in the legends of figures. Also, what is the "Correspondence analysis (CoA)" and components 1, 2, and 3? Is CoA the principal component analysis (PCA)? Please clarify them and provide some details (e.g., two or three sentences for each analysis) in the methods section.

4. Too much data is presented in four figures. Please consider using 6 to 7 figures as permitted in a research article by this journal.

5. A structural/anatomical cartoon with a major experimental outline (in Figure 1a), including the main placental cell types: cytotrophoblasts, syncytiotrophoblasts (STs), and extravillous cytotrophoblasts (EVTs), etc in early placental development, would enhance the presentation of the data for a broader readership.

Author Rebuttal, first revision:

Reviewer #1

Chen et al have provided additional data, confirming that STs can indeed be infected by SARS-CoV-2, but at 10-fold lower rate than lung AT2 cells. The authors have also added other new data, including co-culture of trophoblast organoids with SARS-CoV-2-infected endometrial cells and integration with expression data from placental tissues. Overall, this is a comprehensive analysis of SARS-CoV-2 infection in trophoblast cells that will be of interest to the community. However, the authors need to be more careful when describing their results in relation to other recent studies:

1. The recently published paper from Pentao Liu (Ruan et al., Cell Rep Med, 2022) has several overlapping findings with this study, but is currently only mentioned in passing. Ruan et al. also reported infection of ACE2-positive STs, impaired ST differentiation in SARS-CoV-2-infected trophoblast cells, upregulation of interferon signaling, and suppression by Remdesivir. Given these overlapping findings, it would be better to mention this work explicitly in the Discussion and in the summary of prior literature on p. 4.

We thank the reviewer for their support, and we apologise and thank the reviewer for pointing out that we need to describe other recent studies in more detail. On this note, we always strive to acknowledge previous work and in the case of Ruan et al, the paper came out when we were wrapping up our resubmission. As such, our intention was to make sure that we cited them and that's why we mentioned Ruan et al, in our result and discussion section.

In order to address this point we will: 1) change the paragraph in the introduction summarising previous work to include Ruan et al work (see below) and 2) Expand our discussion (see below in comment #2 and #3):

“To date only a few studies have investigated SARS-CoV-2 infection of the placenta using *in vitro* models. Fahmi et al. showed that SARS-CoV-2 can replicate to varying degrees in placental explants relative to their ACE2 expression⁵⁴. Argueta et al. confirmed infection of placental clusters and showed an association with a robust inflammatory response²⁸. Although the ability to infect primary placental cells is promising, these models are limited to analysis of at-term placental tissue. In contrast, Karvas et al. utilized a trophoblast organoid approach that models early placental cells; however, they found very limited infection with SARS-CoV-2⁵⁵. In addition, recently Ruan et al. using a model derived from extended pluripotent stem cells (EPSCs) and trophoblast organoids, showed that mononuclear STs exhibited susceptibility to SARS-CoV-2 infection with limited infection observed in mature STs, TSCs and EVT. Therefore, further exploration of susceptible early placental models and the potential for secondary infection is imperative to fully understand the implications of SARS-CoV-2 infection during early implantation.”

2. Karvas et al. did not report an infection rate of 1.6% in 2D STs, as is currently stated on p. 7. That percentage referred to infection of undifferentiated TSCs (see Figure S5C in Karvas), which actually agrees quite well with the data in this manuscript. Karvas et al. did not quantify the proportion of infected STs, but only reported that they saw more robust infection by VSV. Of note, Ruan et al. did quantify the proportion of their infected STs (~10% at 24h post-infection, see Figure S3Q in Ruan). The difference with the 57% of infection in ST cells

in this manuscript may be related to timing or the method of detection (dsRNA vs. SARS-CoV-2 N). The dsRNA antibody used here is actually not specific to SARS-CoV-2 and could potentially reflect a broader stress response independent of viral infection.

We thank the reviewer for pointing out the error with the percentage of infected cells in Karvas et al. We made a mistake in the interpretation of their data and we will of course correct this statement.

In regards to the dsRNA question, while we agree that detection of dsRNA is not specific for SARS-CoV-2, this technique is widely used by virologists to identify virus-infected cells and tissues. We acknowledge that in some circumstances cellular stress can also result in accumulation of intracellular dsRNA. However, in our hands we have previously shown a strong co-localisation between dsRNA and SARS-CoV-2 nucleocapsid staining in SARS-CoV-2 infected cells (Lee et al, 2022, Viruses).

Nevertheless, in order to confirm our results, we have now quantified the proportion of SARS-CoV-2 nucleocapsid positive STs from Extended Figure 1m. As can be seen below, this quantification agrees with the dsRNA staining (58.65±1.12% of cells are infected). We will include this new graph in the subfigure (see below). Together, these observations suggest that the dsRNA staining is a true representation of infected cells and that dsRNA accumulation due to a stress response unrelated to viral infection is unlikely to contribute significantly to this quantification.

Therefore, our data suggest that differences between our study and Ruan et al, is most likely due to timing or models. We will include a sentence in the discussion with respect to this (see below).

Reviewer Figure 1 (Fig. 1m in manuscript), Percentage of uninfected and infected cells in mock and virus-infected ST cultures quantified using dsRNA and SARS-CoV-2 N.

To address these points we will add the following sentence into the discussion:

“In our study, we found that TSCs and EVT_s were not infected by SARS-CoV-2. Ruan et al. reported low infection rates in TSCs (2-3% naive TSCs) and EVT_s (1-2%). Interestingly, we found that SARS-CoV-2 infection in ST cells was ~57 % at day 3 (measured by dsRNA and SARS-CoV-2 N), whereas Ruan et al. reported a lower infection rate of ~10% at day 1 of differentiation. The discrepancy between the two studies might reflect differences in models, differentiation, differentiation stage of the STs, and timing of analysis post infection.”

3. It seems to me that the three abovementioned studies are largely in agreement on key findings: SARS-CoV-2 can productively infect a subset of ST cells, but only has limited capacity to infect trophoblast organoids. The latter was well-documented in the Karvas and Ruan papers and confirmed in Fig. 2j of the present manuscript, in which no significant viral infection was seen in organoids that were maintained in TO medium. While the percentage of infected STs varies between Ruan and this manuscript, this may be explained by technical variables (see point 2 above). I think it would be helpful to emphasize the common points raised by these three independent studies in the Discussion.

Thanks for this suggestion, we agree that the three studies are in agreement on many points, however they also differ and compliment in some key aspects. Following the Reviewer suggestion, and as mentioned in our response to question 1, we will add the following paragraph in our discussion emphasising the common and different points of the three manuscripts:

“Furthermore, this study confirmed the finding of Ruan et al. that STs are susceptible to infection by several virus variants early into differentiation (2 days of differentiation, at mononuclear stage). Our study expands upon this, showing that infection can occur also late in ST differentiation (polynuclear stage), which indicates that infection could occur at different stages of syncytiotrophoblast development. Furthermore, our study indicates a blockage of differentiation upon infection. Interestingly, although as expected, both studies found genes associated with response to viral infection upregulated; our study found genes associated with cellular structure/function were also affected providing a possible molecular explanation for the observed impairment in ST differentiation and morphology upon SARS-CoV-2 infection. In agreement with reports by Karvas et al. and Ruan et al., organoid models are less susceptible to infection when maintained in trophoblast organoid medium. However, contrary to Karvas et al., we confirmed the findings of Ruan et al., that organoids can be infected upon ST differentiation, although the percentage of infected cells varies between studies, as discussed above). In this study, we further expanded these organoids models by generating a secondary infection model with endometrial epithelial cells (Ishikawa cells) and importantly, we showed robust secondary infection when trophoblast organoids are cultured in ST medium together with infected primary Ishikawa cells. Consistently, antiviral treatments such as Remdesivir prevents infection with SARS-CoV-2 similar to Ruan et al. and we demonstrated that anti-ACE2 antibodies prevented infection, in agreement with the finding that an ACE2 knockout line was refractory to SARS-CoV-2 infection (Ruan et al., 2022). . Importantly, we found that the anti-ACE2 antibody therapy restores proper HCG levels and lowers cell death in infected cultures. Importantly, we showed that the combination of different anti-ACE2 and anti-spike antibodies are vital for the prevention of infection with Omicron variants, unlike ancestral SARS-CoV-2. Altogether, our study confirms previous reports on the ability of SARS-CoV-2 to infect ST cells and provides important new insights into the mechanism and possible treatments, highlighting the importance of different methods and models to describe and understand these processes.”

Reviewer #2

The revised manuscript “An iTSC-derived placental model of SARS-CoV-2 infection reveals ACE2-dependent susceptibility in syncytiotrophoblasts” by Chen et al examines induced fibroblast derived trophoblast stem cell cultures during SARS-CoV-2 infection. This substantially revised manuscript addresses the majority of concerns raised by this and the other reviewers. Only a few minor questions remain.

Major Comments

1) It is not clear where the neutralizing ACE2 antibodies and used plasmids identified and generated by the investigators would be available from the investigators or certain public repositories. This should be stated in the methods.

We thank the reviewer for the support and we are pleased to learn that their major concerns have been addressed.

Yes, of course, all plasmids are available upon request and we will include the sequence of the antibody as is customary. We will add the following sentence in Material and methods in the respective sections::

“All plasmids are available upon request”

“Sequences of the ACE2 antibodies are listed in Extended Data Table 2.”

2) The transcriptome work strongly supports the investigator hypothesis. No GEO reference or publicly available database is made for where this data will be located.

We thank the reviewer for pointing this out and we apologise that the GEO number was not there. However, we would like to note that we always deposit our data and indeed we added the GEO reference when we submitted the revised manuscript into the submission form of Nature Cell Biology, and we incorrectly assumed that this information was shared. We will add this information in the manuscript under data availability.

Data Availability:

Repository/DataBank Accession: GEO

Accession ID: GSE185471

Databank URL: <https://www.ncbi.nlm.nih.gov/geo/query/acc.cgi?acc=GSE185471>

3) In figure 4h the transcriptomic changes are evaluated using GSVA signatures for each cell type and notes the differences in resulting signature after virus/ACE2 neutralizing ab exposure. The ACE2 neutralizing antibody treated mock cells seems to have an altered expression of these signatures as well with a noted increase in the CT2 signature. What is accounting for these changes? Do you see impacts on cellular differentiation status or cellular composition in that control group?

This is an interesting point, and we thank the reviewer for noticing that the Mock^{aACE2} has the highest score for the CT2 signature, as it gives us the opportunity to provide an explanation. AS described in the original GSVA paper: “GSVA calculates sample-wise gene set enrichment scores as a function of genes inside and outside the gene set, analogously to a competitive gene set test”. This is achieved by a calculation of sample-wise gene set enrichment scores as a function of genes inside and outside the gene set. To help in the interpretation of the scores, we have now plotted the expression of the genes that make up

each of the signatures within the scores (Reviewer Figure 2a), allowing us to explore the entire expression profile of the gene signature (Reviewer Figure 2b). We observed that the positive enrichment of the $Mock^{aACE2}$ for the CT2 signature is due to subtle differences in a small fraction of the genes within that signature. Meanwhile, the $Virus^{Ctrl}$ sample showed a robust upregulation and downregulation of the CT1 and STp signature scores, respectively, with respect to the $Virus^{aACE2}$, $Mock^{aACE2}$ and $Mock^{Ctrl}$ samples. (Reviewer figure 2b). Furthermore, hierarchical clustering of the samples for each signature, shows that the $Mock^{aACE2}$, $Mock^{Ctrl}$ and $Virus^{aACE2}$ samples are always clustered together, which is in line with our findings.

In regards to the question on the cells, overall, we did not notice an impact on the cellular differentiation or composition at the time of harvest for RNA-sequencing.

These heatmaps will also be included in the manuscript (Extended Data Fig 7) to provide further clarity to any disparity of the scores reflected in the signatures.

Reviewer Figure 2 (b will be in Extended Data Fig. 7 in manuscript), a) GSEA gene signature scores of d3 infected/mock/treated STs samples for the *in vivo* cytotrophoblast and ST progenitor signatures. b) Heatmaps of placental gene signatures CT1, CT2 and STp, with hierarchical clustering of the samples based on the selected gene expression profile.

Reviewer #3

RE: NCB-A48564A: Chen et al. An iTSC-derived placental model of SARS-CoV-2 infection reveals ACE2-dependent susceptibility in syncytiotrophoblasts

Major Comments

The revised manuscript consists of a significant amount of new data that have appropriately addressed the reviewer's major concerns.

We are pleased that our revised manuscript has addressed all the reviewer's major concerns.

Other Comments

1. Pages 10-12: Under the "SARS-CoV-2 affects differentiation potential and metabolic activity" section: The term "Metabolic activity" is not accurate. The authors only measured LDH levels as a marker for cell death under this section, without examining metabolic activity (e.g., glycolysis), metabolic pathways, and metabolites. So, "metabolic activity" may be deleted throughout the text where no actual metabolic assays were examined.

We apologize for the wording of the title in this Results section and our interpretation of "metabolic activity" in the section. Following the reviewer's advice, we will change the title to make it more clear: "SARS-CoV-2 affects differentiation potential, cell death and HCG production".

2. When genes are expressed at the non-protein level, their symbols should be italicized as required by the HUGO nomenclature. Please check this throughout the text (e.g., pages 15 and 16) and revise them accordingly.

We thank the reviewer for indicating this to us and we will make the appropriate changes to italicize gene names when needed.

3. "UMAP projection" is mentioned in multiple figures without providing its full term for its first appearance in the text or in the legends of figures. Also, what is the "Correspondence analysis (CoA)" and components 1, 2, and 3? Is CoA the principal component analysis (PCA)? Please clarify them and provide some details (e.g., two or three sentences for each analysis) in the methods section.

We apologise that we did not clarify the term UMAP or CoA, we will add the following sentences in the "Materials and methods, Gene expression analyses" section of the manuscript:

"Uniform Manifold Approximation and Projection. UMAP is a dimension reduction algorithm used to visualise high-dimensional data (such as single cell RNA-seq), which works by constructing a low-dimensional embedding of the data, where observations (cells) that are similar remain together. It also preserves the global structure of the data. (McInnes et al. 2018)"

"Correspondence Analysis (CoA) is a dimension reduction technique which can, similar to Principal Component Analysis (PCA), display a low dimensional projection of data. However,

one of the key differences between CoA and PCA is that with CoA, two variables of the data may be analysed and visualised to observe the relationship between them (e.g. samples and genes). (Fellenberg et al. 2001; Culhane et al. 2002). “

4. Too much data is presented in four figures. Please consider using 6 to 7 figures as permitted in a research article by this journal.

We appreciate the reviewer’s remarks on figures being too dense and we are happy to move the data into more figures or supplementary. We will do this following the editorial advice.

5. A structural/anatomical cartoon with a major experimental outline (in Figure 1a), including the main placental cell types: cytotrophoblasts, syncytiotrophoblasts (STs), and extravillous cytotrophoblasts (EVTs), etc in early placental development, would enhance the presentation of the data for a broader readership.

We appreciate the great suggestion to include an outline subfigure and we will include the following figure as suggested to enhance the readership’s understanding of our study (any suggestion is welcome)..

Reviewer Figure 3 (Fig. 1a in manuscript), Overview of study of *in vitro* placental model.

Response to additional comments from reviewer 1:

The authors have addressed my concerns and now quantify the proportion of SARS-CoV-2 infected STs based on nucleocapsid staining as well as dsRNA. They also added a more detailed discussion of the paper by Ruan et al.

I only noticed one remaining issue, which can be easily addressed. The authors write in their Discussion: “However, contrary to Karvas et al., we confirmed the findings of Ruan et al., that organoids can be infected upon ST differentiation, although the percentage of infected cells varies between studies, as discussed above).”

Actually, neither the Karvas nor Ruan studies performed ST differentiation on trophoblast organoids. Both those studies only performed ST differentiation on 2D-cultured hTSCs, as described in the Okae paper. As far as I’m aware, this manuscript is the first to report successful ST differentiation of trophoblast organoids.

Therefore, I suggest modifying this sentence slightly:

“However, in contrast to these two prior studies, we demonstrate that trophoblast organoids can be infected by SARS-CoV-2 upon ST differentiation.” This will provide a good introduction to the description of the secondary infection model in which the authors use their novel ST-cultured trophoblast organoids.

We thank this reviewer one more time for all his work and support on this manuscript which has helped us improve it.

We have now added the following sentence to the manuscript as suggested: *“In contrast to these two prior studies, in addition to demonstrating a robust infection of STs in 2D models, we also demonstrated that trophoblast organoids can be differentiated into STs and infected by SARS-CoV-2 upon differentiation”*

Additional comments on authors' response to Reviewer #2

I've read the authors' response to the remaining comments from referee #2 and am happy to say that they've addressed those concerns adequately. Overall, the data in Fig. 4h support the authors' conclusion about the effect of the ACE2 neutralizing antibody.

We thank the reviewer for looking at these responses as well.

Final Decision Letter:

Dear Jose,

I am pleased to inform you that your manuscript, "A placental model of SARS-CoV-2 infection reveals ACE2-dependent susceptibility and differentiation impairment in syncytiotrophoblasts", has now been accepted for publication in Nature Cell Biology. Congratulations to you and the whole team!

Please note that *Nature Cell Biology* is a Transformative Journal (TJ). Authors may publish their research with us through the traditional subscription access route or make their paper immediately open access through payment of an article-processing charge (APC). Authors will not be required to make a final decision about access to their article until it has been accepted. Find out more about Transformative Journals

If you have not already done so, we strongly recommend that you upload the step-by-step protocols used in this manuscript to the Protocol Exchange (www.nature.com/protocolexchange), an open online resource established by Nature Protocols that allows researchers to share their detailed experimental know-how. All uploaded protocols are made freely available, assigned DOIs for ease of citation and are fully searchable through nature.com. Protocols and Nature Portfolio journal papers in which they are used can be linked to one another, and this link is clearly and prominently visible in the online versions of both papers. Authors who performed the specific experiments can act as primary authors for the Protocol as they will be best placed to share the methodology details, but the Corresponding Author of the present research paper should be included as one of the authors. By uploading your Protocols to Protocol Exchange, you are enabling researchers to more readily reproduce or adapt the methodology you use, as well as increasing the visibility of your protocols and papers. You can also establish a dedicated page to collect your lab Protocols. Further information can be found at www.nature.com/protocolexchange/about

With kind regards,
Stelios

Stylios Lefkopoulos, PhD
He/him/his
Associate Editor
Nature Cell Biology
Springer Nature
Heidelberger Platz 3, 14197 Berlin, Germany

E-mail: stylios.lefkopoulos@springernature.com
Twitter: @s_lefkopoulos
